



# Elucidation of the myrcene ozonolysis mechanism from a Criegee Chemistry perspective

Meifang Chen[1,2], Shengrui Tong[1,*], Shanshan Yu[1], Xiaofan Lv[1,2], Yanyong Xu[1,2], Hailiang Zhang[1], Maofa Ge[1,2]

[1]State Key Laboratory for Structural Chemistry of Unstable and Stable Species, Beijing National Laboratory for Molecular Sciences (BNLMS), Institute of Chemistry Chinese Academy of Sciences, Beijing 100190, P.R. China.
[2]University of Chinese Academy of Sciences, Beijing 100049, P.R. China.

*Correspondence to:* Shengrui Tong (tongsr@iccas.ac.cn)

**Abstract.** Criegee intermediates (CIs) are highly reactive species generated during the alkene ozonolysis, which play a critical role in atmospheric chemistry. Myrcene is a typical monoterpene, and its linear structure is significantly different from other cyclic monoterpenes such as α-pinene. This structural distinction consequently leads to different reactions mechanisms. This study employs a combined approach of matrix isolation Fourier transform infrared spectroscopy (MI-FTIR) and smog chamber experiments to elucidate the mechanisms of myrcene ozonolysis from the Criegee chemistry perspective. Two CIs with different molecular sizes, C3-CIs and C7-CIs, are captured at 880 and 905 cm$^{-1}$ by using MI-FTIR. Ordered oligomers with C3-CIs serving as chain units, formed via $RO_2$ + n C3-CIs + $HO_2$/$RO_2$ mechanisms, are detected as significant components in secondary organic aerosol (SOA). C7-CIs are more prone to unimolecular degradation to form C7-$RO_2$ radical, which act as initiators for oligomerization reactions. The mechanisms may also exist in other monoterpenes ozonolysis, which offering new insights into the contribution of CIs to SOA formation. Furthermore, the mechanisms of the synergistic interaction between SCIs oligomerization and $RO_2$ autoxidation are illustrated. The mechanisms facilitate the rapid formation of highly oxygenated species, playing a critical role in particle nucleation. The increase in relative humidity can effectively reduce the formation of higher-order oligomers, thereby suppressing the SOA yields. This study provides a systematic elucidation of myrcene ozonolysis mechanisms, thereby significantly enhancing the understanding of oxidation processes in acyclic monoterpenes.

## 1. Introduction

Criegee Chemistry has long been recognized as playing an important role in tropospheric atmosphere. Multiple model simulations have yielded atmospheric concentrations of stabilized Criegee intermediates (SCIs) of around $10^4$–$10^5$ molecules cm$^{-3}$ (Khan et al., 2018; Chhantyal-Pun et al., 2020). The concentration of OH radical is about $10^6$ molecules cm$^{-3}$ (Ringsdorf et al., 2023; Hofzumahaus et al., 2009; Lelieveld et al., 2016). Typically, if $k_{SCI}$ is ~100 times larger than $k_{OH}$, this is enough to make the rate of loss for reaction with SCIs comparable to that for reaction with OH radical. SCIs can undergo extremely rapid reactions with many common trace molecules in the atmosphere, such as $SO_2$, organic acids, and amides, which are also faster than that reactions with OH radical (Long et al., 2025; Welz. O. et al., 2014; Chhantyal-Pun et al., 2015; Mauldin et al.,



2012). Recent study has shown that SCIs are significantly more efficient at removing greenhouse gases ($(CF_3)_2CFCN$) than OH radical (Jiang et al., 2025). The reaction with water vapor is the main removal pathway for SCIs due to the extremely high abundance of water vapor in the atmosphere. The reaction rates of SCIs of different configuration with water vapor are found to be $10^{-11}$-$10^{-20}$ cm$^3$ molecule$^{-1}$ s$^{-1}$. Among them, the reaction rate of syn-$CH_3CHOO$ with $H_2O$ is approximately $10^{-19}$-$10^{-20}$ cm$^3$ molecule$^{-1}$ s$^{-1}$, while the reaction rate of anti-$CH_3CHOO$ with $H_2O$ is faster, at approximately $10^{-14}$ cm$^3$ molecule$^{-1}$ s$^{-1}$ (Vereecken et al., 2022; Yin and Takahashi, 2018; Lin et al., 2016; Anglada and Sole, 2016; Taatjes et al., 2013). The reaction has a pronounced conformational dependence, leading to large differences in reaction rates. Similarly, the reactivity of SCIs unimolecular degradation is also conformation-dependent. Compared to anti-CIs, syn-CIs are more prone to undergo unimolecular degradation reactions (Long et al., 2018; Vereecken et al., 2017).

The unimolecular degradation process of SCIs, as well as bimolecular reactions involving organic acids, $RO_2$, etc. contribute significantly to the formation and growth of atmospheric secondary organic aerosol (SOA) (Boy et al., 2013; Chen et al., 2022; Zhao et al., 2015). Both field observations and laboratory studies have revealed that smaller-sized SCIs undergo sequential oligomerization reactions with $RO_2$ and organic acids, leading to the formation of low volatility organic compounds (LVOCs) or extremely low volatility organic compounds (ELVOCs) (Luo et al., 2025). Oligomers with $CH_3CH_2CHOO$ (SCIs) as chain units were observed during the trans-3-hexene ozonolysis process (Zhao et al., 2015). Early studies on ethylene ozonolysis also identified oligomers formed from the reaction of multiple $CH_2OO$ with organic acids (Sakamoto et al., 2013; Chen et al., 2021). Caravan et al. observed the oligomer sequence obtained through the continuous reaction of formic acid and six $CH_2OO$ in the Central Amazon region (Caravan et al., 2024). These direct evidences indicate that SCIs contribute to SOA formation by oligomerization. These oligomerization mechanisms are now widely observed in the ozonolysis of alkenes that generate small-sized SCIs. Relative humidity (RH) may affect ozonolysis systems through the direct reactions between water vapor and SCIs. Zhang et al. found that limonene yield gradually increased with increasing RH, while SOA yield from $\Delta^3$-carene ozonolysis decreased slightly. They attributed this phenomenon to the differing volatility of the products resulting from the reaction of structurally distinct CIs generated from the ozonolysis of these two monoterpenes with $H_2O$ (Zhang et al., 2023). Quantum chemical calculations have revealed that multiple SCIs may undergo oligomerization reactions with water vapor to form oligomers with lower volatility (Chen et al., 2019). Increasing RH has been found to result in lower SOA yields and reduced oligomerization during the cis-3-hexenol ozonolysis (Harvey et al., 2016). The addition of water alters the reaction mechanism of SCIs in the ozonolysis system, leading to changes in the particle composition, which in turn affects the SOA yield. This indicates that RH is a key factor influencing the ozonolysis mechanism.

Monoterpenes ($C_{10}H_{16}$) emit into the atmosphere at a mean global rate of $95 \times 10^6$ metric tons of carbon per year, and contribute significantly to the global SOA budget. The yields of SCIs obtained from monoterpene ozonolysis ranges from approximately 0.2 to 0.60 (Sipila et al., 2014; Zhang and Zhang, 2005; Gong and Chen, 2021; Cox et al., 2020;



Newland et al., 2018). The high yield leads to the occurrence of numerous SCIs-related reactions. Products derived from SCIs are also significant components of the SOA generated from monoterpene ozonolysis. Both in the α-pinene and limonene ozonolysis, SCIs-derived products contribute to both monomers and dimers formation of SOA (Zhao et al., 2021; Zhang et al., 2023). The emissions of myrcene account for 2% to 10% of total biogenic monoterpene emissions (Sindelarova et al., 2014; Helmig et al., 2013). In addition to biogenic emissions, myrcene can also be present in indoor air (Kostiainen, 1995). The reaction with $O_3$ is one of the main removal pathways in myrcene, and plays an important role in the particle formation in the atmosphere. Myrcene is a straight-chain alkene containing both a conjugated π system, similar to that of isoprene ($CH_2=CH-C(=CH_2)$-moiety), and a second part, structurally analogous to 2-methyl-2-butene ($(CH_3)_2C=CH$-moiety), which is different from most of the widely studied cyclic monoterpenes (e.g., α-pinene, limonene). Such structural differences may give rise to variations in the reaction mechanisms and kinetics. The reaction rate of myrcene with $O_3$ is $3.8\times10^{-16}$ $cm^3$ $molecule^{-1}$ $s^{-1}$, approximately 50 times faster than that of α-pinene ozonolysis and nearly 2 times faster than that of limonene ozonolysis (Deng et al., 2018; Atkinson et al., 1990; Cox et al., 2020; Munshi et al., 1989). About 99% of the $O_3$ addition reactions with myrcene occur at the isolated double bond ($(CH_3)_2C=CH-CH_2-$ moiety) (Deng et al., 2018). Therefore, the myrcene ozonolysis can lead to the formation of two CIs, $C_7H_{10}O_2$ (C7-CIs) and $C_3H_6O_2$ (C3-CIs). For cyclic monoterpenes such as α-pinene and limonene, their ozonolysis processes can only produce larger CIs containing ten carbon atoms (C10-CIs). Current studies have not confirmed that the C10-CIs generated from monoterpene ozonolysis can contribute to SOA formation through oligomerization. This suggests that the contribution mechanisms of CIs of different sizes produced during monoterpene ozonolysis to SOA formation may differ. Further studies are needed to elucidate the distinct roles of these CIs with different molecular sizes in SOA formation mechanisms, as well as the influence of their interactions on particle formation.

In this work, matrix isolation technology combined with vacuum Fourier transform infrared spectroscopy (MI-FTIR) and a smog chamber were employed to investigate the roles of both C3-CIs and C7-CIs in the myrcene ozonolysis. MI-FTIR was used to characterize the key intermediates produced in the early stage of myrcene ozonolysis, which to confirm the formation of stable C3-CIs and C7-CIs. To get the contribution mechanism of C3-CIs and C7-CIs to SOA, we also performed the smog chamber experiments in various conditions during myrcene ozonolysis. RH was also considered as a significant factor influencing the ozonolysis mechanism and taken as an experimental variable. By analyzing the yields and molecular compositions of SOA, we elucidated distinct formation mechanisms through which varying molecular sizes of CIs contribute to SOA formation. Integrating these advanced techniques effectively improved the understanding of myrcene-derived SOA formation mechanism from the Criegee chemistry perspective.



## 2. Materials and methods

### 2.1 Matrix isolation experiment

Myrcene (97%, Aladdin) was further purified through three freeze-pump-thaw cycles before being stored as a liquid in a

storage tube. The $O_2/O_3$ mixture was produced by a high-voltage discharge type ozone generator (Beijing Tonglin Technology

Co., LTD). The collected $O_2/O_3$ mixture was frozen in liquid nitrogen and undergoes several cycles of freezing pump - thawing

to remove the residual impurity gas. $O_3$ or myrcene, in a 1:100 ratio with Ar, was mixed and stored in a spherical glass container.

The matrix was maintained at 6±1 K within a closed-cycle helium refrigerator (Physike Technology Co., LTD, Qcryo-Scryo-

S-300). The characteristic infrared peaks of initial ozonolysis products were detected by using a Fourier transform infrared

spectrometer (Bruker Vertex 70v) equipped with a liquid nitrogen-cooled mercury-cadmium-telluride (MCT) detector.

Measurements spanned 600-4000 $cm^{-1}$, employing 32-scan averaging and 0.5 $cm^{-1}$ resolution. The experimental chamber was

maintained under a vacuum of $10^{-5}$ Pa. The deposition of myrcene/Ar and $O_3$/Ar onto the 6±1 K cold window was facilitated

by two angled and independent tubes, thereby ensuring a brief mixing period for the reactants before deposition. This

deposition was known as the twin-jet co-deposition mode. To allow limit the diffusion and/or reaction of reactants, these

matrices were heated or annealed to 35 K and held for 0.5 h, and then cooled to 6±1 K after which the spectra were recorded.

To promote the further occurrence of the reaction and further soften and diffuse the matrix, the matrix was further heated to

45 and 55 K. To prevent matrix loss, it was imperative to immediately cool down to 6±1 K after reaching the target temperature

and to record the spectra.

### 2.2 Particles generation and collection

The myrcene ozonolysis experiments were performed in a 1.2 $m^3$ atmospheric simulation Teflon chamber maintained at room

temperature (24±1 °C) under different RH conditions. Myrcene was introduced into the chamber by passing zero air through

a heated three-way U-shaped tube. The addition of formic acid and n-hexane was conducted in a consistent manner. The

addition of approximately 367 ppm of n-hexane resulted in the removal of approximately 99% of OH radical. Formic acid was

added about 0.65 ppm. Subsequently, the $O_2/O_3$ mixture was injected into the chamber through a syringe. The $O_3$ concentration

was monitored by an $O_3$ analyzer (Thermo Scientific model 49i) throughout myrcene ozonolysis. The measurement of the

myrcene concentration was conducted by means of a gas chromatograph with a quadrupole mass spectrometer (GC-MS,

Agilent, 7890, 5977B) equipped thermal desorption instrument (TD). The maintenance of different RH levels was achieved

by the implementation of a 10 L/min flow of zero air through the water bubbler. The indoor temperature and RH were measured

using a hygrometer. Throughout each experiment, size distributions and volume concentrations of particles were continuously

recorded using a scanning mobility particle sizer (SMPS), which consisted a differential mobility analyzer (DMA, TSI, Model

3081) and a condensation particle counter (CPC, TSI, Model 3776).



The yield of SOA was obtained by the ratio of the maximum mass concentration of the corrected particles to the mass concentration of myrcene consumed. The average effective density of SOA obtained from the myrcene ozonolysis is 1.25 g cm$^{-3}$ (Boge et al., 2013). The SOA particles generated within the chamber were captured on a 25 mm PTFE filter (Sartorius, 0.45 μm pore size) and subsequently analyzed using an ultra-high performance liquid chromatography with a Quadrupole-Orbitrap mass spectrometer equipped with an electrospray ionization source (UHPLC/ESI-MS, Thermo Scientific). The collected particle sample was eluted with 0.5 ml of methanol (Optima™ LC/MS Grade, Fisher Chemical) into a sample bottle. Mass spectrometric analysis utilized positive ion mode, scanning a molecular weight range of m/z 50–750 Da. In positive ion mode, three ionic forms of particulate components were identified, specifically [M+H]$^+$, [M+Na]$^+$ and [M+NH$_4$]$^+$. Tandem mass spectrometry (MS/MS) was employed to elucidate component structures within the SOA.

**2.3 Quantum chemistry calculations**

The geometries of the SCIs were optimized using the hybrid density functional theory B3LYP-D3(BJ) with the aug-cc-pVTZ basis set. Harmonic and anharmonic vibrational frequencies were calculated at the same calculation level for the comparison with the experimental infrared peak. This method has been proved to be applicable to the relevant calculations of the SCIs system (Chen et al., 2025; Lin et al., 2018; Yu et al., 2025). The above-mentioned related calculations were all performed by using Gaussian 16 software package (Frisch et al., 2016). Molclus 1.1.2 in conjunction with the xtb software package was used to perform a systematic conformational search for myrcene, POZs and SCIs (Lu, 2023). The single-point energy was further calculated at the DLPNO-CCSD(T)/CBS level by using ORCA 5.0 software to obtain the Boltzmann distribution of each conformation more accurately (Neese, 2022).

**3. Results and Discussion**

**3.1 Criegee chemistry in myrcene ozonolysis**

C$_7$H$_{10}$O$_2$ (C7-CIs) and C$_3$H$_6$O$_2$ (C3-CIs) was produced from myrcene ozonolysis. The initial ozonolysis mechanism of myrcene had been established as shown in Figure 1 based on the current studies. The products generated along with C7-CIs and C3-CIs were acetone and 4-vinyl-4-pentental with yields of 0.27 and 0.73, respectively (Deng et al., 2018). The nascent C7-CIs and C3-CIs, possessing high internal energy, rapidly degraded in part. This process could contribute to the formation of OH radical (R2 and R3). This process could contribute to the formation of OH radical. The unimolecular isomerization pathways of CIs were strongly configuration-dependent. Syn-C7-CIs were mainly isomerized by 1,4-H transfer to form vinyl hydroperoxide (VHP), which decomposed to form OH radical and a vinoxy or β-oxo alkyl radical (R2 and R4). Anti-C7-CIs mainly proceed an initial rearrangement (1,3 ring-closure) to form a dioxirane intermediate. Then the dioxirane intermediates could isomerize to form organic acid (R3 and R5). C$_3$H$_6$O$_2$ isomerized mainly through the formation of VHP. Another portion was collisional



stabilized, forming SCIs (R1) (Criegee, 1975; Hassan et al., 2021; Jr-Min Lin and Chao, 2017; Khan et al., 2018). A portion

of the SCIs might undergo unimolecular degradation reactions, while another portion primarily engaged in bimolecular

reactions with trace gases in the atmosphere (Jr-Min Lin and Chao, 2017; Kidwell et al., 2016; Su et al., 2014; Vereecken et

al., 2012). The yields of OH radical and SCIs obtained from the myrcene ozonolysis were $0.63 \pm 0.09$ and $0.30$, respectively

(Cox et al., 2020; Newland et al., 2018).

**Figure 1 Proposed the key pathway in the initial ozonolysis of myrcene. The values in parentheses represent the yields of the corresponding products.**

Although previous studies had provided some references to the initial ozonolysis mechanism of myrcene, the presence of CIs

had only been demonstrated deductively. In this study, MI-FTIR was employed to capture the key intermediates formed during

the initial ozonolysis of myrcene. As shown in Figure 2, some new bands were observed in the spectra obtained by the twin-

jet co-deposition. Most of the absorption peaks of the products grew with rising temperature within the 35-55 K interval. This

observation indicated that the myrcene initial ozonolysis occurred when the Ar matrix softened and diffused. The new bands

were located at 765, 880, 905, 1074, 1177, 1370 and 1720 cm$^{-1}$. The spectra of a single precursor (Myrcene/Ar or O$_3$/Ar)

obtained after annealing at different temperatures as comparison was shown in Figure S1.

The characteristic IR vibrational bands of the POZs, formed via initial cycloaddition between myrcene and O$_3$, were localized

on its 1,2,3-trioxolane ring. These included asymmetric O-O-O stretching and C-O stretching vibrations within the five-

membered ring structure (Wang et al., 2020). The presence of two different configurations in POZs was attributable to the

varying orientations of the central O atom on the five-membered ring. The peaks calculated from the asymmetric stretching

vibration in both POZs configurations overlapped at 772 cm$^{-1}$. Moreover, O-O-O asymmetric stretching vibrations of POZs

were commonly observed within the 700-800 cm$^{-1}$ spectral region as documented in prior literature (Yang et al., 2020; Wang,

2020). Thus, the band at 765 cm$^{-1}$ which obtained from the twin-jet experiment was assigned to the O-O-O asymmetric

stretching vibration of POZs. The calculation results indicated that the C-O stretching vibrations of the two POZs were located



at positions 1068 and 1174 cm$^{-1}$. The newly emerged peaks at positions 1074 and 1177 cm$^{-1}$ corresponded to the C-O stretching
vibration of POZs in the twin-jet spectra. Moreover, these POZ-related infrared peaks emerged at 35 K, and their intensity
increased with rising annealing temperature until distinct infrared peaks became apparent at 55 K.

The main conformers and the infrared vibration frequencies of C3- and C7-CIs were obtained through quantum chemical
calculations. The strongest characteristic infrared vibration peak of CIs was caused by O-O stretching vibration (Su et al., 2013;
Lin et al., 2015). The extant research results indicated that the characteristic infrared peak caused by the O-O stretching
vibration of C3-CIs in the gas phase located at 887 cm$^{-1}$ (Wang et al., 2016). In the twin-jet spectra, a distinct new peak was
observed at position 880 cm$^{-1}$, which was likely to be attributed to the C3-CIs. The resulting ~7 cm$^{-1}$ deviation might be
attributable to the Ar matrix effect (Bach, 1999). The calculated O-O stretching vibrations of C7-CIs were located at 892 (syn-
C7-CIs) and 923 (anti-C7-CIs) cm$^{-1}$, which were covered by the main characteristic infrared vibration peaks of myrcene.
Consequently, it was challenging to directly observe the strongest characteristic infrared vibration peaks belonging to C7-CIs.
In addition to the -COO group, C7-CIs also possessed conjugated double bonds as characteristic functional groups. In both
syn- and anti-C7-CIs, the second most intense infrared vibration band consistently corresponded to the wagging vibration of
the =CH$_2$ group on the conjugated moiety. The calculated position of this vibration agreed with that of myrcene. Position 905
cm$^{-1}$ was the infrared characteristic peak generated by the wagging vibration of myrcene =CH$_2$, as obtained by twin-jet method.
In the spectra, a significant relative increase at the 905 cm$^{-1}$ position was observed after annealing to 55 K. This increase might
be attributable to the generation of C7-CIs. The characteristic infrared peaks of C3- and C7-CIs were both generated after 55
K annealing.

In addition to the early intermediates, acetone was also identified as a major product co-produced with C7-CIs in myrcene
ozonolysis. The spectral bands of 1370 and 1720 cm$^{-1}$ appeared in the twin-jet spectra belong to acetone (Han and Kim, 1996).
The generation of acetone also indirectly demonstrated the formation of C7-CIs. MI-FTIR experiments unequivocally verified
that myrcene ozonolysis proceeded via the Criegee mechanism. Furthermore, both C3- and C7- CIs were generated during this
process.

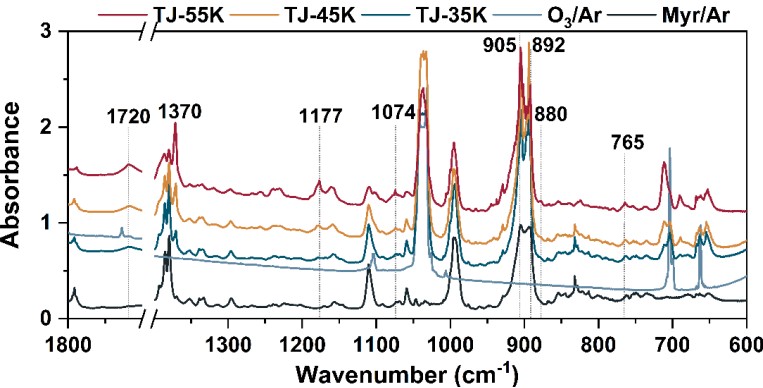

**Figure 2 The twin-jet spectra of myrcene ozonolysis reaction in a low temperature and Ar matrix after annealing to 35 K, 45 K and**





**55 K. T-J means the spectra obtained by twin-jet method.**

**Table 2 Identification and assignments of experimental absorption bands in the initial ozonolysis of myrcene.**

| Experimental bands/(cm⁻¹) | Calculated Band/(cm⁻¹) | Reference | Belonger | Assignment |
|---|---|---|---|---|
| 765 | \ | 772 | POZs | O-O-O str. |
| 880 | \ | 887 [a] | C3-CIs | O-O str. |
| 905 | \ | \ | C7-CIs | $=CH_2$ wag. |
| 1074 | 1068 | \ | POZs | C-O str. |
| 1177 | 1171 | \ | POZs | C-O str. |
| 1370 | \ | 1370 [b] | Acetone | $\delta\ CH_3$ |
| 1720 | \ | 1721.6 [b] | Acetone | C=O str. |

Note: [a] The characteristic infrared peaks of gas-phase C3-CIs measured by step-scanning Fourier transform infrared spectroscopy (Wang et al., 2016).
[b] Measure in the low- temperature Ar matrices (Han and Kim, 1996).
[c] str. stands for stretching vibration; wag. stands for wagging vibration; δ stands for the scissoring vibration.

### 3.2 Impact of CIs on SOA generation

To explore the contribution of CIs to the SOA generation, the myrcene ozonolysis experiments were conducted under different conditions in a 1.2 m³ smog chamber. The unimolecular decay of both excited CIs and SCIs could produce OH radical. Therefore, SOA formation included contributions from both SCIs and OH radical derived in myrcene ozonolysis.

As shown in Table 2, Exp. 1-2 added n-hexane (HA) as an OH radical scavenger to highlight the contribution of SCIs-derived pathways. Exp. 3-5 were conducted with progressively increasing RH to modulate SOA formation by introducing direct reactions between water and SCIs. The volume concentration of the particles rose sharply within approximately 30 minutes and reached its maximum value within 180 minutes (Figure S2). The dominant size range of SOA expanded from 50-250 nm during myrcene ozonolysis (Figure 3). The suppression of particle formation induced by the OH scavenger and elevated RH was conspicuously demonstrated. Following the 367 ppm HA addition, the particle yield from myrcene ozonolysis decreased by approximately 72%. As RH increased from dry conditions to ~20% and subsequently to 50%, the particle yield gradually declined by 17% to 40% (Table 2). As shown in Fig. 2(a), after the addition of HA, no significant reduction in the particle size distribution was observed within the first 30 min. This might be attributed to the rapid reactions of SCIs, indicating that gas-phase reactions of SCIs played an important role in particle nucleation. Between 90 and 180 min of reaction, the particle size distribution decreased markedly. This reduction was due to the scavenging of OH radicals, which inhibited the formation of OH-RO₂ pathway. Bimolecular reactions and the autoxidation of OH-RO₂ played a crucial role in the growth of SOA (Baker et al., 2024). As the RH gradually increased from ~ 20% to 50%, the growth of particle size was inhibited across different time periods. This phenomenon occurred because reactions between water and SCIs might prevent these compounds from converting into larger, less volatile products. Therefore, OH radical generated from CIs and SCIs also contributed to the growth of particle size distribution and enhanced SOA yield.



**Table 2 Summary of experimental conditions.**

| Exp. | [Myrcene]/ppb | [O$_3$]/ppb | RH/% | Scavenger | M$_{SOA}$(μg/cm³) | Y$_{SOA}$ |
|------|---------------|-------------|------|-----------|-------------------|-----------|
| 1 | 171.5 | ~ 200 | < 0.5 | / | 346 | 0.36 |
| 2 | 170 | ~ 200 | < 0.5 | ~ 367 ppm HA | 91 | 0.10 |
| 3 | 222 | ~ 200 | < 0.5 | / | 509 | 0.41 |
| 4 | 231 | ~ 200 | ~20 | / | 432 | 0.34 |
| 5 | 230 | ~ 200 | ~50 | / | 311 | 0.24 |

Note: n-hexane is abbreviated as HA.

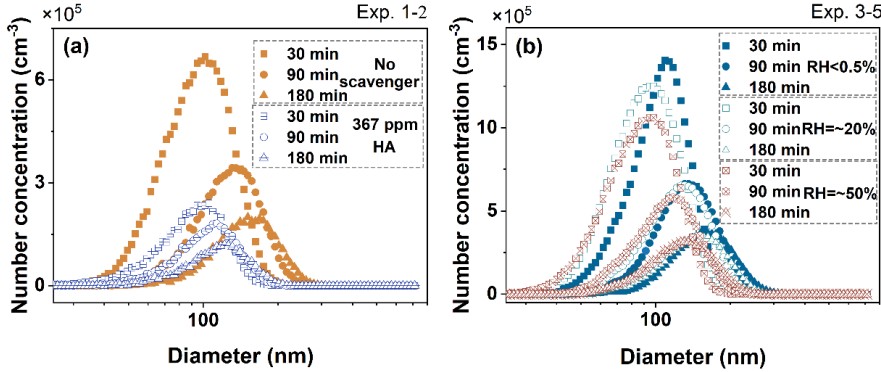

Figure 3 Time series of size distributions of aerosol particles formed at 30, 90, and 180 min after the initiation of myrcene ozonolysis under different conditions. (a) depicted particle formation processes corresponding Exp. 1-2 from Table 2, while (b) illustrated particle formation processes in Exp. 3-5 from Table 2.

### 3.3 Mechanism of SCIs contribution to SOA

UHPLC-ESI-MS was employed to analyze the chemical compositions of SOA collected offline. CIs were important SOA precursors during the ozonolysis process. In this study, the contribution mechanisms of CIs to SOA formation were proposed in myrcene ozonolysis.

$C_{10}H_{18}O_5$ and $C_{10}H_{18}O_4$, with relative abundances (RA) of 76% and 73% respectively, ranked among the top 3 in the absence of any scavengers at RH< 0.5% as shown in Figure 4(a). The formation of these two intense peaks was belonged to the contribution of OH radical oxidation processes. The OH radical was almost entirely originated from the CIs unimolecular degradation reactions. The OH radical and $O_2$ reacted sequentially with myrcene to form $C_{10}H_{17}O_3$ radical (C10-RO$_2$, as shown in Scheme 1, R6). $C_{10}H_{17}O_3$ radical then underwent autoxidation channel which was highly likely to occur to produce $C_{10}H_{17}O_5$ radical (C10-R'O$_2$) (Jokinen et al., 2014). The reaction of $C_{10}H_{17}O_5$ radical with HO$_2$ yielded $C_{10}H_{18}O_5$ (C10-R'OOH) which was detected as the main product. Meanwhile, $C_{10}H_{17}O_5$ radical could also react with any RO$_2$ radical to yield $C_{10}H_{18}O_4$ (C10-R'OH). The relative peak intensity of $C_7H_{10}O_2$ was slightly lower than that of $C_{10}H_{18}O_5$ and $C_{10}H_{18}O_4$, and its molecular formula was consistent with C7-CIs. It was inferred that it might originate from C7-OH produced during the unimolecular degradation of C7-CIs. C7-CIs underwent unimolecular degradation via the VHP pathway, producing OH radical while simultaneously generating R radical. The R radical subsequently reacted with $O_2$ to form $C_7H_9O_3$ (C7-RO$_2$). $C_7H_9O_3$ could react with any RO$_2$ radical to form $C_7H_{10}O_2$ (C7-OH). The OH radical yield from myrcene ozonolysis was generally high,



which also confirmed that the larger CIs generated during this process tend to react via unimolecular decay pathways (Cox et al., 2020).

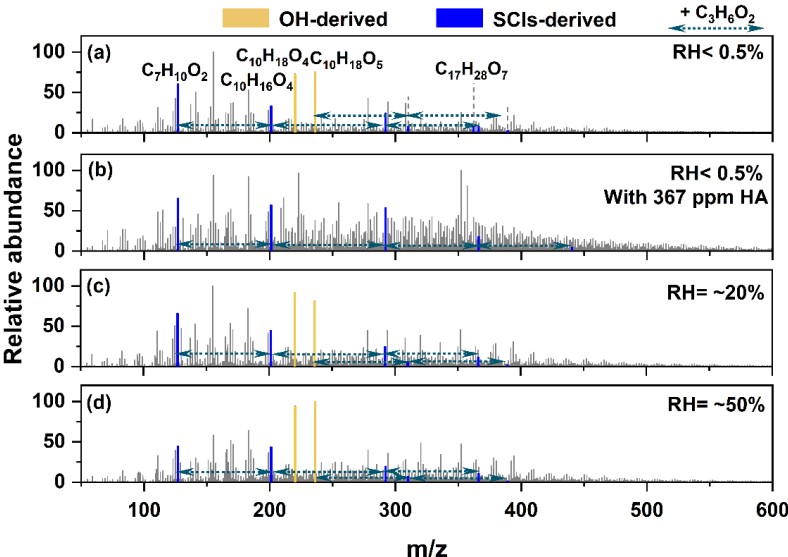

**Figure 4** UHPLC/ESI-MS of SOA from myrcene ozonolysis in different conditions. (a) No scavenger was added. (b) 367 ppm n-hexane (HA) was added. (c) RH=~20% condition. (d) RH=~50% condition. The substances marked by the yellow lines originated from the OH-derived channel, while those marked by the blue lines mainly originated from the SCIs-derives channel. The oligomer sequence containing C3-CIs as the chain unit was marked by the green dotted line.

$C_{10}H_{17}O_5$ radical as the main $RO_2$ radical could undergo sequential oligomerization. Several peaks representing ordered oligomers with $C_3H_6O_2$ as chain units were found. The chemical formula of $C_3H_6O_2$ was identical to C3-CIs and confirmed to existent by the matrix isolation experiment. The proposed pathway was shown in R7. $C_{10}H_{17}O_5$ radical reacted with one or two C3-CIs, followed by reaction with $HO_2$ radical resulting in chain termination ($C_{10}H_{17}O_5$ + n C3-CIs + $HO_2$, n= 0-2, Sequence 1). However, the relative intensities of this set of oligomer signals were relatively low. This was likely because $C_{10}H_{17}O_5$ radical more readily reacted with $HO_2$ or $RO_2$ to form $C_{10}H_{18}O_5$ and $C_{10}H_{18}O_4$, thereby reducing the likelihood of its reaction with C3-CIs in the gas phase. These oligomer sequences exhibited similar distribution patterns of major fragment ion peaks in MS/MS spectra in Figure S3(a). Therefore, these oligomers with C3-CIs as the chain units had similar structural units in their chemical composition. Moreover, the corresponding structural assignments for its major fragment peaks were provided in Figure S3(b). The fragments $C_5H_7^+$ (m/z 67.06), $C_6H_9O_2^+$ (m/z 113.06), and $C_9H_{13}O^+$ (m/z 137.10) revealed that the oligomers in Sequence 2 all feature a carbon chain skeleton characterized by conjugated double bonds. The $C_3H_7O^+$ (m/z 59.05)/$C_3H_7O_2^+$ (m/z 75.05) fragment ions originated from cleavage of terminal -OH or -OOH functionalized tertiary carbon moieties in the oligomers. For C7-CIs, no significant oligomers with C7-CIs as chain units were observed when reacting with $C_{10}H_{17}O_5$ radical. Although the mass spectral peak corresponding to $C_{17}H_{28}O_7$ was observed and its molecular formula aligned with the expected product of $C_{10}H_{17}O_5$ + C7-CIs. But $C_{17}H_{28}O_7$ could also originate from $C_{10}H_{17}O_5$ + C7-$RO_2$ ($C_7H_{11}O_4$, the formation pathway as shown in Scheme S1). Moreover, $C_{17}H_{28}O_7$ did not appear as a dominant product in the mass spectra. Consequently, the





contribution of C7-CIs oligomerization to myrcene-derived SOA formation might be considered negligible. In addition, an

ordered oligomer sequence formed by $C_7H_{10}O_2$ + n C3-CIs was also identified. The corresponding compounds in the sequence

are $C_7H_{10}O_2$ (m/z 127.075), $C_{10}H_{16}O_4$ (m/z 201.112), $C_{13}H_{22}O_6$ (m/z 292.175) and $C_{16}H_{28}O_8$ (m/z 371.168), namely Sequence

2 (The peaks with relative abundance of less than 1% were ignored). The proposed pathway of Sequence 2 was as shown in

the R8. Sequence 2 had a greater degree of oligomerization and contributed more to particle formation than Sequence 1. This

indicated that $C_7H_9O_3$, which produced by C7-CIs unimolecular reaction, more readily reacted with C3-CIs in the gas phase.

The oligomerization reactions within the Sequence 2 all terminated with the reaction of RO$_2$ radical ($C_7H_9O_3$ + n C3-CIs +

RO$_2$). Consistently, no discernible oligomerization signals were detected for reactions using $C_7H_9O_3$ as the starting reactant

with C7-CIs.

$$\text{(R7)}$$

As shown in Figure 4(b), the peaks corresponding to $C_{10}H_{18}O_4$ and $C_{10}H_{18}O_5$ disappeared after the addition of the OH radical

scavenger, as expected. The $C_7H_{10}O_2$ peak remained, which further demonstrated that $C_7H_{10}O_2$ originated from SCIs-derived

products. Compared with the mass spectrum without the scavenger, the contribution of the oligomers in Sequence 2 markedly

increased. Water served as the dominant removal pathway for SCIs in the atmosphere. Correspondingly, the contribution of

Sequence 2 to SOA formation decreased progressively with increasing RH as shown in Figure 4(c) and (d). So, the Sequence

2 was primarily contributed by SCIs-derived.

$$\text{(R8)}$$

Meanwhile, some compounds were found to have the same number of C and H atoms, differing only in the number of O atoms

in parallel oligomer sequences, namely, $C_7H_{10}O_{2-5}$, $C_{10}H_{16}O_{4-7}$, $C_{13}H_{22}O_{6-9}$ and $C_{16}H_{28}O_{8-11}$ (Sequence 2N) as shown in Figure

5(a). Compounds with identical carbon and hydrogen atom counts exhibited closely resembling major fragment ions in their

MS/MS spectra as shown in Figure 5(b). This suggested that compounds with identical carbon and hydrogen atom counts

exhibited highly consistent formation pathways. The process leading to progressive oxygenation (i.e., increasing oxygen atom

content) was the RO$_2$ autoxidation mechanism (Jokinen et al., 2014; Liu et al., 2023). Simultaneously, Sequence 2N was the

oligomer series formed by oligomerization with C3-CIs as chain units. This indicated that SCIs oligomerization and RO$_2$



autoxidation mechanistically interplayed in the formation of Sequence 2N. The proposed formation mechanism of Sequence 2N was illustrated in Figure 5(c). $RO_2$ underwent oligomerization with n SCIs to form the oligomer $RO_2$-n SCIs, while concurrently undergoing autoxidation to generate $R'O_2$. $R'O_2$ might subsequently underwent analogous cycles of SCIs oligomerization and autoxidation. This radical propagation cascade ultimately terminated via reactions with $RO_2$ or $HO_2$ radical.

SCIs inherently contained at least two oxygen atoms in their structure due to the presence of terminal -COO groups. The $RO_2$ autoxidation inherently involved reaction with $O_2$, which rapidly elevated the oxygen content in products. Consequently, the synergistic interplay between SCIs oligomerization and $RO_2$ autoxidation facilitated rapid formation of high-molecular-weight species and highly oxygenated molecules (HOMs), which substantially contributed to particle nucleation and growth.

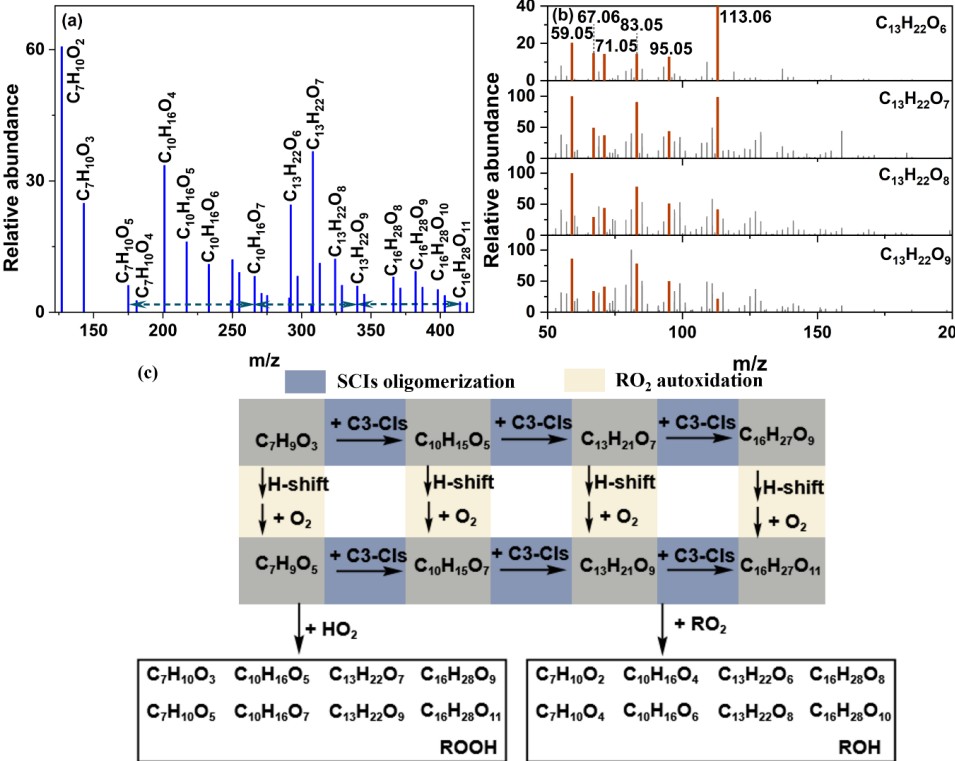

**Figure 5 The relative abundance and the proposed mechanism of oligomer sequences derived from the synergistic SCIs oligomerization and $RO_2$ autoxidation. (a) Mass spectrometric distribution of oligomeric sequences under dry conditions. (b) The MS/MS spectra of $C_{13}H_{22}O_{6-9}$ and the major fragment ions were highlighted in orange in the mass spectra. (c) The proposed mechanism of this oligomer sequences. The blue area denoted the SCIs oligomerization pathway. The yellow area represented the $RO_2$ autoxidation pathway. The black solid box indicated the particulate composition detected by mass spectrometry (MS).**

The addition of different scavengers also affected the formation of Sequences 2N. Figure 6 illustrated the proportions of compounds with various oligomerization degrees of oligomerization in Sequence 2N under different scavenger conditions. The dominant compounds in Sequences 2N were $C_{13}H_{22}O_{6-9}$ which contained two C3-CIs as chain units under dry conditions, with or without added OH radical (Figure 6(a) and (b)). Furthermore, following the addition of OH radical, the proportions of both $C_{13}H_{22}O_{6-9}$ and $C_{16}H_{28}O_{8-11}$ in Sequences 2N gradually increased (Figure 6(b)). This indicated that the SCIs



oligomerization reaction significantly contributed to the particle formation after the addition of the OH scavenger. This was

consistent with the result observed in Sequences 2. As the RH increased, the main compound in Sequences 2N became

$C_{10}H_{16}O_{4-7}$, which contained only a single C3-CIs as a chain unit (Figure 6(c) and (d)). The proportions of both $C_{13}H_{22}O_{6-9}$ and

$C_{16}H_{28}O_{8-11}$ in Sequence 2N decreased with increasing RH. This indicated that the reaction of water with C3-CIs directly

affected the generation of oligomers with higher degrees of oligomerization. Additionally, the fraction of $C_7H_{10}O_{2-5}$ decreased

significantly with increasing RH. This indicated that besides reacting with C3-CIs, water also scavenged C7-CIs to some extent.

Therefore, the contribution of the SCIs oligomerization mechanism to SOA formation were suppressed under elevated RH

conditions.

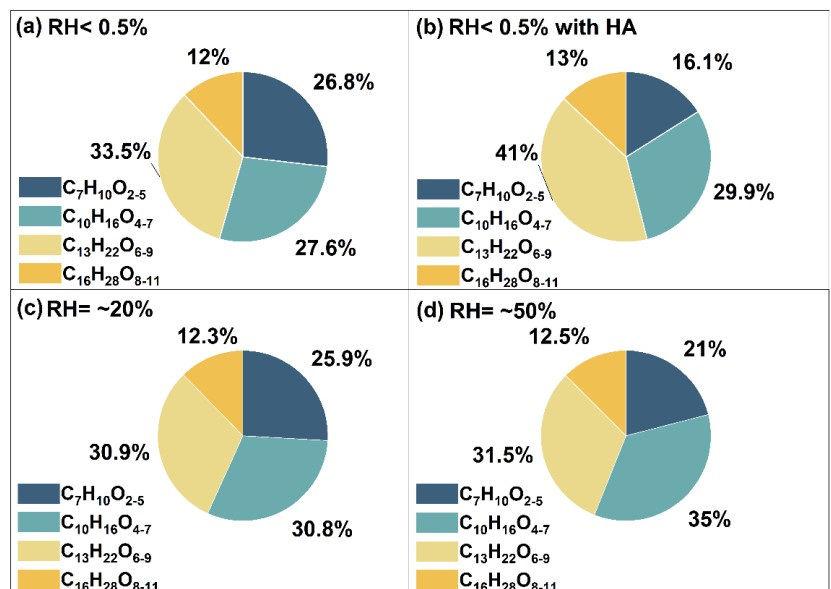

**Figure 6 The proportion of compounds in sequence 2N generated during the myrcene ozonolysis under different conditions.**

Sequence 1, 2 and 2N demonstrated ordered oligomers formed with C3-CIs as chain units. However, ordered oligomeric

sequences (RO$_2$ + n C7-CIs + HO$_2$/RO$_2$, n≥2) with C7-CIs as the chain unit were not discovered. Based on the current formation

mechanism of oligomers, C3-CIs and C7-CIs differed in their contribution to particle formation in myrcene ozonolysis. For

smaller molecular-sized CIs (C3-CIs), the primary pathway for partitioning into the particle involved oligomerization with

RO$_2$ radical and other species. The C7-CIs had higher molecular masses than C3-CIs. The products from unimolecular and

bimolecular reaction of C7-CIs partitioned more readily to particles which reduced the viability to function as chain units in

oligomerization processes (Donahue et al., 2012; Donahue et al., 2011). Furthermore, C7-RO$_2$ generated from unimolecular

decomposition of C7-CIs acts as a chain-initiating precursor that reacts with C3-CIs, thereby contributing to particle nucleation.

These findings implied divergent dominant pathways for SOA formation mediated by SCIs across varying molecular

dimensions in VOCs ozonolysis. The C10-CIs with high molecular weights generated during the ozonolysis of α-pinene and

limonene might also be more likely to incorporate into the particle via unimolecular reaction mechanisms rather than through oligomerization. Our findings further demonstrated that when evaluating SCIs contribution pathways to SOA, the molecular size of SCIs must be prioritized especially during monoterpenes ozonolysis. In the myrcene ozonolysis system, the branching ratio might favor the formation of C3-CIs over C7-CIs, which further enhanced the propensity of C3-CIs toward oligomerization.

## 4. Conclusions

This study employed the MI-FTIR method to directly determine the presence of C3-CIs and C7-CIs during the myrcene ozonolysis. The O-O stretching vibration peak of C3-CIs was located at 880 cm$^{-1}$. The characteristic infrared vibration peak of C7-CIs, caused by the $=CH_2$ wagging vibration on the conjugated double bonds, was located at 905 cm$^{-1}$.

Furthermore, combined with smog chamber experiments, it was verified that these two distinct CIs of different molecular sizes differ in their primary contribution mechanisms during SOA formation. For C3-CIs, they primarily underwent oligomerization reactions (e.g., $RO_2$ + n C3-CIs + $RO_2$/$HO_2$) to form lower-volatility oligomers that incorporate into the particle. Meanwhile, the SCIs oligomerization and $RO_2$ autoxidation synergistically occurred during the myrcene ozonolysis, leading to the formation of HOMs-$RO_2$ with higher oxygen content. To our knowledge, this was the first time this synergistic mechanism has been proposed. Larger C7-CIs primarily underwent unimolecular decomposition, producing C7-ROH and C7-$RO_2$. C7-ROH could be detected as a stable product in particle. The C7-$RO_2$ subsequently oligomerized with n C3-CIs to partition into the particle phase. The coexistence of these CIs of different molecular sizes led to a distinctly different ozonolysis mechanism for myrcene compared to that of cyclic monoterpenes (e.g., α-pinene, limonene). An increase in RH caused some SCIs to react directly with water, thereby reducing the fraction of SCIs that form oligomers and partition into the particle. This mechanism also significantly contributed to the decrease in SOA yield and the shift toward a smaller particle size distribution. Our study further deepened the understanding of monoterpene ozonolysis mechanisms, particularly from the perspective of Criegee chemistry.

### Data availability

All raw data can be provided by the corresponding authors upon request.

### Supplement

The supplement related to this article is available online at:



**Author contribution**

MFC and SRT planned the campaign; MFC, XFL and YYX performed the measurements; MFC and SSY analyzed the data; MFC wrote the manuscript draft; MFC, SRT, SST, HLZ and MFG reviewed and edited the manuscript.

**Competing interests**

The authors declare that they have no conflict of interest.

**Acknowledge**

This work was supported by the National Key Research and Development Program of China (No. 2022YFC3700200), National Natural Science Foundation of China (Contract No. 42130606, 22321004).

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
