# Peer review of "Elucidation of the myrcene ozonolysis mechanism from a Criegee Chemistry perspective"

_EGUsphere, 2025_

## Referee Comment (RC3)

Review of *Elucidation of the myrcene ozonolysis mechanism from a Criegee Chemistry perspective* by Chen et al.

**Significance**

Formation of condensable chemicals initiated by gas-phase ozonolysis of volatile organic compounds (VOCs) contributes significantly to atmospheric secondary organic aerosol (SOA) budgets. The molecular level mechanism is generally understood to evolve through the well-known Criegee mechanism which produces two distinct zwitterion / diradical species called Criegee intermediates (CIs). These CIs are known to have a rich uni- and bimolecular chemistry, commonly dominated by unimolecular decomposition and reaction with water dimer under atmospheric conditions. The current study focuses on myrcene, an often-overlooked acyclic monoterpene with three double-bonds, and reports the apparent importance of its two CIs, which dominate the initial oxidation product distribution, on atmospheric SOA formation. The work utilizes complementary investigation methods from matrix isolation and chamber investigations to quantum chemical computations and attempts to understand myrcene oxidation chemistry by synthesizing the output from these distinct methodologies.

While the topic of the work is certainly of interest to the readers of ACP, in the current form it is difficult to assess what has actually been accomplished here. Specifically, the current level of documentation does not allow to fully assess the reliability of the results as the methods and results appear only partly described. Also, to me it seems that the type of oligomerization reaction described here would be seriously kinetically limited in the atmosphere, and thus I do suspect there could be some easier explanation for the observed product signals. I'll detail my concerns below.

**Major comments**

First the kinetic limitation: I suspect that most of the gas- and particle-phase results could be explained by the more common peroxy radical (RO2) chemistry without the need to invoke exotic Criegee intermediate (CI) oligomerization reaction which is generally limited by the availability of the very reactive CIs in almost any conceivable atmospheric environments. That is, the CIs simply cannot find an "already dimerized" reaction product to form a "further trimerized" product, because they react away in multitude of reactions with several of the co-produced oxidation products (e.g., any products with carbonyl groups). It is simply difficult to see how the CI concentration could ever be so high to permit sequential reactions with the same products under such reaction time and oxidizing conditions. To me it seems far more likely that recently uncovered pathways in RO2 + R'O2 reactions, where RO (and R'O) radical rearrangement occurs after the initial peroxy radical cross-combination reaction and leads to products with various amounts of carbon and oxygen in the "dimeric" structures, is the explanation here. Please have a

look at Peräkylä et. al. (https://pubs.acs.org/doi/full/10.1021/jacs.2c10398) and Frandsen et al. (https://pubs.acs.org/doi/full/10.1021/acsearthspacechem.4c00355) for the mechanism and topical examples. Could they explain what is seen here? Again, it is hard to see how the numbers would match and allow the oligomerization to happen, and that's why I urge the Authors to back up the current sequential oligomerization conclusions by a gas-phase kinetic modelling of the relevant reaction system using some prototypical reaction rate coefficients.

Second is the lack in documentation: Many important details are missing, and the current story appears to choose the results across the very specific investigation methodologies without clearly referencing on what part of the study the results have been obtained. Please remember that the minimum amount of documentation is always such that the work can be repeated and hence the results verified in a replicate study. Now, I am not sure how I could repeat the matrix isolation work, which is likely the best described of the experimental procedures, though still appears to miss the temperature of the mixing jet and the details of mixing the reactants, the volume of the chamber, the timescale of deposition and reaction, the purity of the O3 mixture, for example. The aerosol formation study seems to be missing more details including the timescale, the used reactant and sampling flows, the details of aerosol particle and gas compound detection, the details of the LC-MS technique and so on. The computations seem to be missing almost all details, and it is not clear what has been computed. You should give the computational details and the resulting molecular geometries in the SI. The current level of documentation in the SI is inadequate, and you must add all the relevant details to be able to replicate the study.

**More specific comments**

Note that in this reaction system ozonolysis initiates all the observed oxidation chemistry and based on your results in Table 2 the co-produced OH also makes a big, apparently dominating impact on SOA mass. Now, when you add water, you decrease the whole oxidation sequence – also the important OH that would be generated from the CI isomerization through the VHP decompositions. Thus, it is very unclear if the reduction in SOA occurs specifically through scavenging of the Criegee intermediate and preventing its oligomerization or because the added water reduces also the further sequence of reactions contributing to SOA in the system.

Moreover, when zooming on the SOA yields in Table 2, they indicate that OH chemistry played a major role in forming the observed SOA (i.e., 91 vs 346 ug/cm^3). How was the OH scavenging determined to be 99% completed? Were there any repeated experiments using the hexane scavenger?

So, you say that increasing RH decreased the SOA yield. But there's no apparent change between 0.5% and 20% humidity, and the humidity only plays an apparent role at 50% RH. This leads to a question that how was the particle size measured, and subsequently the SOA mass determined?

I suspect the particles were dried before sizing, correct? (=not documented here). I would expect the particles to collect considerably more water at 50% than at 20% RH, so maybe the "missing SOA" mass at higher humidity is simply evaporating water. Could this be the case? Again, hard to say with the missing documentation. As a related result the data in Figure 6 are hardly conclusive as the results at 0.5% and 20% RH look very similar and the HA addition seems to modulate mainly the C13 product, which makes sense if it was affected by a RO2 + RO2 → RO + RO +O2 step. Also, contrary to the text, the C7H10Ox products appear to be decreasing with OH scavenging as well. Is this the case when looking at the details?

From the figures it is unclear what species were detected and what are just assumed based on mechanistic principles. This is especially true for figure 5: Is the Figure 5a a measured spectrum or just a visualization of the identified peaks? Note that you will obtain the C10H15Ox radical by hydrogen abstraction reaction too, and under such a high loading conditions RO2 + RO2 also surely occurs generating the odd oxygen product species through alkoxy radical isomerization reaction.

**Comments about the methodology**

These are specialized techniques that must be explained carefully. The reader probably does not know that the signals are not exactly comparable across Ar matrix at 35K and particle-phase at room temperature. These are very different physical worlds, but now it sounds like it is just okay to equate chemical observations from the matrix to gas- and particle-phases. Why would you expect so? The minimum is that you explain to the reader why you think you can equate these worlds.

I am also a bit worried about the experimental conditions, but due to lack of documentation it is hard to be sure. So, CIs react with many of the present oxidation products with rapid rates, and if you really are observing CI related oligomerization, then it implies that you are using very high concentrations. Otherwise, it is very hard to see how you could see such reactive species oligomerizing in the gas-phase. However, from the Table 2 it seems that the highest primary oxidation rate is around 0.002 s$^{-1}$ (corresponding to $k$ x [O3]) which is rather low in comparison to atmospheric oxidation rates but still appears to rapidly result in the very high particle loads obtained without seed particles. Please explain what does not add up in these results.

Generally, it is very unrewarding to read "experiments were conducted under different conditions". From the quite vague results given on the particles it seems that the experiment had very high oxidizing conditions, which seems surprising indeed in absence of seed particles. For example, it is said ". The dominant size range of SOA expanded from 50-250 nm during myrcene ozonolysis (Figure 3)." How can you reach so high particle loads without seed particles? What was the history of the used chamber setup? Could that have affected the results?

More about the chamber experiments:

- What is the timescale of the chamber experiments: It seems your growth rates are very high to obtain so many particles at so little time.
- A representative figure of the experiment as a function of time showing the O3, myrcene and some product time profiles would help to put the results in context.
- How was the chamber experiments performed? Based on adding O2/O3 mixture with a syringe it sounds like you were doing batch-mode experiments, right? With the current documentation it is unclear.
- Spark generators do generally produce a lot of NOx. Did you measure how much NOx is in your reaction gas?
- The spectra given in Figure 4 are hard to compare. Especially the spectrum with hexane scavenger is very noisy and it is pretty much impossible to compare it to the peaks in the others. Also, it seems to contain more peaks than the other experiments, but the measured SOA yield is less. Does this mean that these specta are not relevant to the SOA observations?
- What is the cut-off size and the maximum size detected by the aerosol instrumentation?
- How stabile was the O3 syringe injection?
- What are the details of the LC-MS measurements? Please explain why you would get Na+ and NH4+ clusters from a normal H3O+ source? Or what was the utilized ion source? What settings were used in the MS/MS analysis?

Why did you study the system by computations? Were they performed only to get the corresponding IR absorptions? If yes, then this should be explained clearly. Also, the accuracy of the predictions should be discussed. Currently the details of the computations are apparently missing.

What was the temperature in the twin-jet mixing stage. I'm trying to understand at what conditions the MT+O3 reaction occurred as it is difficult to see how you could form the Criegee in the cold matrix through POZ isomerization.

Some of the new peaks resemble mor like noise, for example the 880 and 1074 cm^-1. How do you define a new peak exactly?

You said that "(The peaks with relative abundance of less than 1% were ignored)." But the Spectra shown in Figure 4 contain 100s (or 1000s?) of peaks and you have only labelled 5 and given a handful of others in the text. Thus, how many of the peaks were neglected? You should ideally provide the peaklist for signals (above some threshold) observed during the experiment.

Clearly more details are needed to understand the experiments done, and thereby also the proposed chemistry, which is discussed next.

**Comments about the proposed mechanisms**

The manuscript makes several claims about the potential mechanism of oligomer formation by Criegee intermediate reactions and their relevance for SOA formation from myrcene.

First of all, I would like to take the time to explain that in chemistry the word "mechanism" has a very strong meaning and is reserved to explain how the molecules transform. What is meant here by the same word in several instances is hardly a mechanism. Thus, mentions like "The mechanisms may also exist in other monoterpenes ozonolysis, which offering new insights into the contribution of CIs to SOA formation." do not make much sense without detailing the molecular steps.

Related: "The coexistence of these CIs of different molecular sizes led to a distinctly different ozonolysis mechanism for myrcene compared to that of cyclic monoterpenes (e.g., α-pinene, limonene)". No. It is the same mechanism for both but with acyclic species the bond breaking leads to two species, whereas with cyclic species only one product is generated.

And further: I can't seem to make sense of the following statement: "Our findings further demonstrated that when evaluating SCIs contribution pathways to SOA, the molecular size of SCIs must be prioritized especially during monoterpenes ozonolysis."

And further: You say that "To our knowledge, this was the first time this synergistic mechanism has been proposed." – this seems like an awkward statement as there is no actual mechanism presented.

And further: "MI-FTIR experiments unequivocally verified that myrcene ozonolysis proceeded via the Criegee mechanism." – this is confusing as it is completely unclear what would be the "other mechanism" the Authors are referring to? Ozonolysis is commonly expected to proceed through the Criegee mechanism.

The detected compounds are C7, C10, C13 and C16 species, which all appear to have also alternative production paths, especially through RO2 chemistry. What is noteworthy is that in the current chamber experiments very high growth rates are obtained even in apparent absence of seed particles which testifies the very high oxidation conditions used in the experiments. Under such conditions many sorts of radical recombination can occur – potentially even the sCI + RO2, which I still find much more unlikely than the RO2 + R'O2 processes.

"Current studies have not confirmed that the C10-CIs generated from monoterpene ozonolysis can contribute to SOA formation through oligomerization." This makes sense in considering the

reactivities of the CIs and sCIs discussed in the above comments and again appears to point out that it is more likely you are observing RO2 chemistry. Perhaps this is possible in high concentrations in a laboratory setting, but even then, it is not so easy to make sCIs oligomerize due to abundance of other potential sCI reaction partners generated during oxidation (e.g., any species containing carbonyl functionality).

**Comments**

I would strongly recommend language editing by a native speaker as the text contains several apparent ambiguities that hinder understanding the work. Some examples below:

When you talk about unimolecular degradation of sCI contributing to SOA it seems odd. Note that "degradation" seems to imply the molecule breaking into small pieces whereas you probably just mean the OH loss through a VHP that kickstarts the autoxidation process. I can only assume you mean this as nothing like that has been explained in the article (e.g., this is omitted from Figure 1, for example).

Figure captions should be expanded to explain clearly what is shown in the figures. Some examples: It took me a while to realize that the myrcene + O3 spectrum is included in Figure 2 as it is currently poorly labelled and not mentioned in the caption. Figure 1 is messy and hard to follow and will require a long explanation of the steps shown (e.g., the all-important VHP decomposition is not marked).

I can't understand the following: "The OH radical yield from myrcene ozonolysis was generally high, which also confirmed that the larger CIs generated during this process tend to react via unimolecular decay pathways (Cox et al., 2020)". Why "generally"? Why "confirm" here?

The following seems to contradict itself: "The initial ozonolysis mechanism of myrcene had been established as shown in Figure 1 based on the current studies."

"As shown in Figure 4(b), the peaks corresponding to $C_{10}H_{18}O_4$ and $C_{10}H_{18}O_5$ disappeared after the addition of the OH radical scavenger, as expected. The $C_7H_{10}O_2$ peak remained, which further demonstrated that $C_7H_{10}O_2$ originated from SCIs-derived products. Compared with the mass spectrum without the scavenger, the contribution of the oligomers in Sequence 2 markedly increased." With this low-resolution figure, it is hard to say what peaks decreased and what increased. To me it looks like the sequence 2 peaks actually decreased. Also, the next claim "Correspondingly, the contribution of Sequence 2 to SOA formation decreased progressively with increasing RH as shown in Figure 4(c) and (d)." I really can't read from the current figure.

As this is ACP the page content is not limited. Thus, you should put the SI material directly into the main text to improve readability.

**More minor comments:**

- Please refrain from using "extreme" when it is not needed. There is definitely no "extremely high abundance of water vapor" and Criegee does not react at extremely fast rates.
- What is a "a heated three-way U-shaped tube."?
- There is no Table 1.
- "Zhang et al. found that limonene yield gradually increased with increasing RH" – I don't think you mean limonene concentration was increasing.
- Please explain in the text why you added formic acid to the mixture?
- Mark the peaks with the corresponding assignments in Figure S1.
- What is the difference in sequence 2 and sequence 2N?
- Note that it is generally not possible to label the products to -OOH and -OH species simply by their measured composition. It is plausible these could be the products but with the current techniques and experimental conditions it appears that you have no way to be sure about them. Reword accordingly.
- The following is misleading "Quantum chemical calculations have revealed that multiple SCIs may **undergo oligomerization reactions with water vapor to form oligomers** with lower volatility (Chen et al., 2019)." – this is not what the cited article says or what you mean here.
- This is ambiguous "Both in the α-pinene and limonene ozonolysis, SCIs-derived products contribute to both monomers and dimers formation of SOA" – of course the reaction initiating the whole oxidation systems contributes. Please be clearer what you mean.
- Open all abbreviations. For example, POZ does not appear to be explained.

---

## Author Comment (AC1)

**Response to reviewers' comments on "Elucidation of the myrcene ozonolysis mechanism from a Criegee Chemistry perspective"**

**Response to Reviewer #1**

This study employs a combined approach of matrix isolation Fourier transform infrared spectroscopy (MI-FTIR) and smog chamber experiments to elucidate the mechanisms of myrcene ozonolysis from the perspective of Criegee chemistry, thus significantly enhancing our understanding of oxidation processes in acyclic monoterpenes.

The manuscript presents research of high quality, with sound methodology and clearly described experimental processes. The findings are of considerable significance and are generally well-articulated and convincing. The methods applied are appropriate, and the reaction mechanisms are described in a detailed and clear manner. This work merits publication in Atmospheric Chemistry and Physics after the authors address the minor revisions outlined below.

The authors greatly thank the reviewer for the careful review of our manuscript and the valuable feedback. All the comments are addressed point by point, with our responses in blue, and the corresponding revisions to the manuscript in red *italics*. All updates are marked in the revised manuscript. According to the comments, we have added new analysis to strengthen our work.

**Specific comments:**

1.  It is recommended to provide additional clarification on the calculation method of SOA yield in Table 2, such as whether corrections were made for particle wall loss.

**The author's answer:** Thank you for your comments. In our study, the SOA yield was calculated according to the following formula.

$$Y_{SOA} = \frac{M_{SOA}}{\Delta M_{VOC}}$$

Here, $Y_{SOA}$ represented the SOA yield, $M_{SOA}$ denoted the maximum mass concentration of particle after wall-loss correction during the reaction process, and $\Delta M_{VOC}$ referred to the total consumption mass concentration of VOCs throughout the reaction.

In this study, wall-loss corrections for SOA were applied. Specifically, ammonium sulfate particles were introduced smog chamber under identical environmental conditions (e.g., relative humidity (RH)

and temperature), and their wall-loss rate was used to correct the SOA mass concentration. Table R1 presented the wall-loss rates of ammonium sulfate particles measured under different RH conditions. And in supplement Figure S2 presented the corrected SOA volume concentration.

**Table R1 Wall-loss rate of ammonium sulfate as a function of relative humidity.**

| | RH | Rate/(min$^{-1}$) |
|---|---|---|
| 1 | RH<0.5 % | $0.00165 \pm 7.83 \times 10^{-5}$ |
| 2 | RH= ~20% | $0.0019 \pm 7.47 \times 10^{-5}$ |
| 3 | RH= ~50% | $0.00301 \pm 8.97 \times 10^{-5}$ |

[Figure]

**Figure S2 Time series of particle volume concentration in myrcene ozonolysis under different condition. (a) depict particle formation processes in Exp. 1-2 from Table 2, while (b) illustrate particle formation processes in Exp. 3-5 of Table 2.**

**Lines 123-126**, we added "*The specific equation was as follows.*

$$Y_{SOA} = \frac{M_{SOA}}{\Delta M_{VOC}}$$

*Here, $Y_{SOA}$ represented the SOA yield, $M_{SOA}$ denoted the maximum mass concentration of particle after wall-loss correction during the reaction process, and $\Delta M_{VOC}$ referred to the total consumption mass concentration of VOCs throughout the reaction.*".

2. In the MI-FTIR experiment, the characteristic peak of C7-CIs (905 cm$^{-1}$) overlaps with that of myrcene itself. How was interference from the parent compound ruled out in this case?

**The author's answer:** Thank you for your comments. The spectra of a single precursor (myrcene/Ar or O$_3$/Ar) were conducted at different temperatures. The results showed that temperature variation did not alter the intensity of the infrared characteristic peak of myrcene or O$_3$ at 905 cm$^{-1}$ as shown in Figure S1(a). To assess and minimize computational errors, we compared the accuracy of several commonly calculational levels used for calculating POZs and CIs infrared spectra in VOC ozonolysis. This

comparison identified B3LYP/6-311G++(d,2p) as the optimal method for CIs, which we accordingly used to recalculate all infrared spectra. Based on a re-examination of the characteristic IR peaks of C7-CIs, we concluded that no such peaks were observed in our experiments. However, in the spectrum from the twin-jet experiment at 55 K, the relative peak intensity at 905 $cm^{-1}$ increased significantly. This indicated the formation of new species exhibiting a characteristic vibration at this location. The characteristic IR bands of C3-CIs could only be detected at 55 K, and the intensity change of the peak at 905 $cm^{-1}$ followed this consistent behavior. Therefore, the 905 $cm^{-1}$ position might be attributed to the 4-vinyl-4-pentenal (C7-aldehyde).

**Lines 187-192**, we modified "The calculated O-O stretching vibrations of C7-CIs were located at 892 (syn-C7-CIs) and 923 (anti-C7-CIs) $cm^{-1}$, which were covered by the main characteristic infrared vibration peaks of myrcene. Consequently, it was challenging to directly observe the strongest characteristic infrared vibration peaks belonging to C7-CIs. In addition to the -COO group, C7-CIs also possessed conjugated double bonds as characteristic functional groups. In both syn- and anti-C7-CIs, the second most intense infrared vibration band consistently corresponded to the wagging vibration of the =$CH_2$ group on the conjugated moiety. The calculated position of this vibration agreed with that of myrcene. Position 905 $cm^{-1}$ was the infrared characteristic peak generated by the wagging vibration of myrcene =$CH_2$, as obtained by twin-jet method. In the spectra, a significant relative increase at the 905 $cm^{-1}$ position was observed after annealing to 55 K. This increase might be attributable to the generation of C7-CIs. The characteristic infrared peaks of C3- and C7-CIs were both generated after 55 K annealing." to "*The calculated O-O stretching vibrations of C7-CIs were located at 882 (syn-C7-CIs) and 910 (anti-C7-CIs) $cm^{-1}$. According to Table S1, the calculated IR peaks of CIs were consistently overestimated by over 10 $cm^{-1}$ relative to experimental values, with the deviation being most pronounced for anti-CIs. Notably, no distinct new peaks appeared below 882 $cm^{-1}$ in the experiment. This absence suggested that very few stabilized C7-CIs were likely generated during the ozonolysis of myrcene. Instead, most C7-CIs were consumed via unimolecular decay pathways, this conclusion which was also supported by our subsequent analysis.*".

**Lines 148-149**, we changed "The products generated along with C7-CIs and C3-CIs were acetone and 4-vinyl-4-pentental with yields of 0.27 and 0.73, respectively." to "*The products generated along with C7-CIs and C3-CIs were acetone and 4-vinyl-4-pentental (C7-aldehyde) with yields of 0.27 and 0.73, respectively.*".

**Lines 195-196**, we added "*The peak at 905 cm$^{-1}$ coincided in temperature with the C3-CIs peaks and suggested its assignment to the =CH$_2$ wagging vibration of the C7-aldehyde.*".

3. In this study, the conclusion regarding the role of C7-CIs in SOA formation primarily relies on their degradation product C7-RO$_2$. Could this potentially underestimate the direct involvement of C7-CIs themselves in other bimolecular reactions, such as those with organic acids?

**The author's answer:** Thank you for your comments. For CIs, the unimolecular and bimolecular reactions were competing processes. In mass spectrometric analysis, the signals corresponding to the products of C7-CIs (C$_7$H$_9$O$_3$) unimolecular degradation exhibited significant intensity (Top 4), both in the presence and absence of OH scavengers. Furthermore, anti-C7-CIs could react via a unimolecular degradation pathway to produce C7-acid (C$_7$H$_{10}$O$_2$). No observable products (C$_{14}$H$_{20}$O$_4$, m/z= 253.142, relative abundance=1.73%) from the reaction between C7-CIs and C7-acid were detected in the system. Thus, we considered unimolecular degradation to be the primary loss process for C7-CIs in the myrcene ozonolysis system.

Both excited-state and stabilized Criegee intermediates (SCIs) were susceptible to unimolecular degradation. However, only the SCIs could participate in rapid bimolecular reactions with organic acids. Our analysis detected no oligomeric sequences with C7-CIs as repeating units in the particles. This suggested that C7-CIs were predominantly removed via unimolecular degradation in their excited state, with only a minor fraction surviving to the stabilized form. Consequently, bimolecular reactions made a negligible contribution to particle formation for C7-CIs in myrcene ozonolysis.

4. In the Results and Discussion section, it is mentioned that for C10-CIs produced by cyclic monoterpenes such as α-pinene and limonene, they may more closely resemble C7-CIs in being incorporated into the particle phase through monomeric reactions rather than oligomerization. This inference tends to rely more on molecular size analogy. Are there any literature sources that provide direct experimental observations or quantum chemical calculations to support this view?

**The author's answer:** Thank you for your comments. Currently, no literature has elucidated the contribution mechanisms of CIs of different sizes to SOA formation from the perspective of molecular size. This study presented this concept for the first time. However, indirect evidence supporting our hypothesis existed in the literature. First, the unimolecular reactions of CIs could generate OH radicals. As shown in Table R2, the main monoterpene ozonolysis reactions generally exhibited higher OH radical

yields ($\leq 0.97$), which was consistent with the inference that most CIs undergo unimolecular reactions. Second, existing studies on the limonene ozonolysis mechanism have identified the monomer as the dominant contributor to SOA (Zhang et al., 2023).

**Table R2 Summary of OH yields for reactions of $O_3$ with monoterpene at 298 K and 1 bar.**

| Monoterpene | OH yield | References [a] |
|---|---|---|
| α-pinene | 0.80±0.10 | (Cox et al., 2020) |
| β-pinene | 0.30±0.06 | |
| limonene | 0.66±0.04 | |
| camphene | ≤0.18 | |
| 2-carene | 0.81±0.11 | |
| 3-carene | 0.86±0.11 | |
| myrcene | 0.63±0.09 | |
| β-ocimene | 0.55±0.09 | |
| β-phellandrene | 0.29±0.05 | |
| sabinene | 0.33±0.05 | |
| α-terpinene | 0.32±0.06 | |
| γ-terpinene | 0.81±0.11 | |
| terpinolene | 0.70±0.08 | |

Note: [a] This table is sourced from a review on CIs.

**Lines 342-343**, we added "*Currently, no literature has elucidated the contribution mechanisms of CIs of different sizes to SOA formation from the perspective of molecular size. This study presented this concept for the first time.*".

5. This study mentions that "the contribution of C7-CIs oligomerization is negligible." Is this due to their low reactivity or the high volatility of the reaction products? It is recommended to briefly explain this in the discussion.

**The author's answer:** Thank you for your comments. As shown in Figure 4(c), oligomers derived from the addition of two or more C7-CIs were not observed. Notably, even the monomeric adduct $C_{17}H_{28}O_7$, might resulting from the reaction of C10-R'$O_2$ with a single C7-CIs, displayed only weak signal intensity. $C_{17}H_{28}O_7$ was classified as a low-volatility organic compound (LVOCs, as calculated by the formula below), which readily partitioned into the particle phase. But its low abundance implied that oligomerization via C7-CIs is not a major process. The primary sink for C7-CIs was their competing unimolecular degradation, as evidenced by the high-intensity signals of their decomposition products ($C_7H_{10}O_2$) detected in experiments both with and without an OH scavenger as shown in Figure 4(a) and

(d).

The saturation mass concentration ($C^0$, µg m$^{-3}$) of $C_{17}H_{28}O_7$ was also calculated based on its elemental composition using the following expression:

$$\log_{10} C_i^0 = \left(n_C^0 - n_C^i\right)b_C - n_O^i b_O - 2\frac{n_C^i n_O^i}{n_C^i + n_O^i}b_{CO}$$

Where $n_C^0$ was the reference carbon number; $n_C^i$ and $n_O^i$, represented the numbers of carbon and oxygen atoms, respectively; $b_C$ and $b_O$ denoted the contribution of each carbon and oxygen atom; and $b_{CO}$ was the carbon–oxygen nonideality. LVOCs defined by saturation mass concentrations of $0.3\text{-}3\times10^{-4}$ µg m$^{-3}$.

**Lines 272-274**, we changed "$C_{17}H_{28}O_7$ did not appear as a dominant product in the mass spectra." to "*$C_{17}H_{28}O_7$ was classified as a low-volatility organic compound (LVOCs) (Donahue et al., 2012; Donahue et al., 2011), which readily partitioned into the particle phase. But its low abundance implied $C_{17}H_{28}O_7$ did not appear as a dominant product.*".

6. Some sentences in the text are quite lengthy; it is suggested to appropriately break them down to enhance the readability of the article.

**The author's answer:** Thank you for your comments. Following the reviewers' suggestions, we have made some revisions to some of the long sentences in the manuscripts. The details are as follows:

**Lines 14-15**, we changed "Ordered oligomers with C3-CIs serving as chain units, formed via $RO_2$ + n C3-CIs + $HO_2/RO_2$ mechanisms, are detected as significant components in secondary organic aerosol (SOA)." to "*Ordered oligomers, which contain C3-CIs as chain units, are detected as significant components in secondary organic aerosol (SOA). These oligomers are formed via $RO_2$ + n C3-CIs + $HO_2/RO_2$ mechanisms.*".

7. Is the synergistic mechanism discovered in this study universal in other monoterpene or alkene systems beyond the myrcene system?

**The author's answer:** Thank you for your comments. Our findings established that the synergistic mechanism emerged from the coexistence of smaller (C3-CIs) and larger-sized CIs (C7-CIs) during monoterpene ozonolysis as shown in Figure 5(c). Such a coexistence of CIs of varying molecular sizes

might also occur in other monoterpenes besides myrcene, including ocimene (C3-CIs and C7-CIs). Although the results and analysis within the manuscript suggested the probable occurrence of this synergy in these systems, its definitive verification remained a subject for future experimental investigation.

**Lines 357-358**, we added "*The structural similarity of ocimene to myrcene meant its ozonolysis might also lead to CIs of varying sizes, potentially allowing for this synergistic mechanism.*".

**References**

Cox, R. A., Ammann, M., Crowley, J. N., Herrmann, H., Jenkin, M. E., McNeill, V. F., Mellouki, A., Troe, J., and Wallington, T. J.: Evaluated kinetic and photochemical data for atmospheric chemistry: Volume VII - Criegee intermediates, Atmos. Chem. Phys., 2020, 13497-13519, 10.5194/acp-2020-472, 2020.

Donahue, N. M., Epstein, S. A., Pandis, S. N., and Robinson, A. L.: A two-dimensional volatility basis set: 1. organic-aerosol mixing thermodynamics, Atmos. Chem. Phys., 11, 3303-3318, 10.5194/acp-11-3303-2011, 2011.

Donahue, N. M., Kroll, J. H., Pandis, S. N., and Robinson, A. L.: A two-dimensional volatility basis set - Part 2: Diagnostics of organic-aerosol evolution, Atmos. Chem. Phys., 12, 615-634, 10.5194/acp-12-615-2012, 2012.

Zhang, S., Du, L., Yang, Z., Tchinda, N. T., Li, J., and Li, K.: Contrasting impacts of humidity on the ozonolysis of monoterpenes: insights into the multi-generation chemical mechanism, Atmos. Chem. Phys., 23, 10809-10822, 10.5194/acp-23-10809-2023, 2023.

---

## Author Comment (AC2)

**Response to reviewers' comments on "Elucidation of the myrcene ozonolysis mechanism from a Criegee Chemistry perspective"**

**Response to Reviewer #2**

This manuscript presents a comprehensive investigation of myrcene ozonolysis from the perspective of Criegee intermediate (CI) chemistry. Using a combination of MI-FTIR, smog-chamber experiments, UHPLC-Orbitrap MS, and quantum chemical calculations, the authors identify both C3-CIs and C7-CIs, and evaluate their distinct contributions to secondary organic aerosol (SOA) formation. The work proposes a potentially important synergistic mechanism between SCIs oligomerization and $RO_2$ autoxidation, offering new insights into particle nucleation in monoterpene systems.

Thank you for giving us the opportunity to revise our manuscript. We appreciate reviewer for your time and constructive comments, which have helped us significantly improve the quality of our work. We have carefully considered all points raised and have made extensive revisions to the manuscript accordingly. Below, we provide a point-by-point response to each comment. Our detailed responses are provided below (in blue). The manuscript has been thoroughly revised accordingly (in red), with all changes highlighted in *italics*, including new analyses added to address the specific comments.

Major issues:

1. Criegee intermediates are produced in the ozonolysis of myrcene as proposed in Figure 1. It is noted that the formed Criegee intermediates contain C=C group and COO group, which could lead to the C=C group reaction with COO group similar to the previous investigation in the literature (Nat. Commun. 2019, 10, 2003.).

**The author's answer:** Thank you for your comments. We are grateful for your highly innovative suggestion. Indeed, we were aware of this literature (Nat. Commun. 2019, 10, 2003) at the inception of our study. As shown in Figure R1(1), the unimolecular degradation pathways of CIs with a C=O group from the literature were provided. It was plausible that this pathway was accessible to C7-CIs, as shown in Figure R1(2). However, after a detailed examination of the structural features of the C7-CIs, we concluded that the mechanism reported in the literature did not represent the dominant pathway for the unimolecular degradation of the C7-CIs produced in myrcene ozonolysis. A detailed explanation was provided below.

[Figure]

**Figure R1 The unimolecular decomposition of CIs.**

(1) The C=C group of the C7-CIs was part of a conjugated system, which conferred significant stability and resists being broken.

(2) Weak interaction analysis based on the interaction region indicator (IRI) method was conducted for the most stable configuration of C7-CIs as shown in Figure R2 (Lu and Chen, 2021). The blue, green, and red colors in Figure R2 represented the bond interactions, Van der Waals interactions, and steric effects, respectively. The results indicated no interaction between the terminal -COO group and the C=C moiety (Figure R2(a) and (b)). In contrast, weak interactions were identified between the -COO group and the β- and γ-H atoms (Figure R2(a)), implying a potential preference of syn-C7-CIs for unimolecular degradation through an H-transfer mechanism. Analysis of anti-C7-CIs reveals that no weak interactions were present between the -COO group and the C=C moiety (Figure R2(b)). Structurally, the simultaneous addition of the -COO group to C1 and C4 in C7-CIs required considerable molecular distortion, suggesting a potentially high energy barrier for this pathway.

[Figure]

**Figure R2 The isosurface maps of syn-C7-CIs (a) and anti-C7-CIs (b) were obtained via the IRI method. Standard coloring method and chemical explanation of *sign(λ₂)ρ* on IRI isosurfaces (c).**

(3) Considering the new unimolecular pathway noted by the reviewer, the resulting product c-$C_7H_{10}O_2$, which contained a double bond, could react further to form $C_7H_{12}O_5$ and $C_7H_{10}O_4$ as potential

subsequent products (Figure R3). In the absence of a scavenger, the mass spectral intensities for $C_7H_{12}O_5$ (m/z 177.076) and $C_7H_{10}O_4$ (m/z 159.065) were 4% and 10%, respectively. Both compounds were minor constituents in particle. And this degradation pathway did not generate $RO_2$ radicals or the corresponding $C_7H_{10}O_X$ (X=2-5) products. Our analysis revealed the presence of $C_7H_{10}O_X$ (X=2-5) products, as shown in Figure 5 of the manuscript. Therefore, we determined it was not a major degradation (Nat. Commun. 2019, 10, 2003) route for C7-CIs.

**Figure R3 The reaction pathway of $C_7H_{10}O_2$.**

**Lines 39**, we added this references to "*Compared to anti-CIs, syn-CIs are more prone to undergo unimolecular degradation reactions (Long et al., 2018, 2019; Vereecken et al., 2017).*".

In the References part, **lines 483-484**, we added "*Long, B., Bao, J. L., and Truhlar, D. G.: Rapid unimolecular reaction of stabilized Criegee intermediates and implications for atmospheric chemistry, Nat. Commun., 10, 2003, 10.1038/s41467-019-09948-7, 2019.*".

2. Although I think that the calculated results can be used to help explain the experimental measurements, it is better to explain the error bars due to the computational methods. This is very helpful for potential readers to understand the challenges in the calculations.

**The author's answer:** Thank you for your comments. As the reviewer noted, discrepancies could exist between calculated and experimental values for infrared spectra. Calculational levels commonly were employed to calculate POZs and CIs in VOCs ozonolysis systems include: B3LYP-D3(BJ)/6−311++G(d,p) (Yu et al., 2025), B3LYP-D3(BJ)/aug-cc-pVTZ (Lv et al., 2017), B3LYP/6-311G++(d,2p) (Hoops and Ault, 2009; Coleman and Ault, 2010, 2013; Pinelo et al., 2013), B3LYP/6-311++G(2d,2p) (Yang et al., 2020; Deng et al., 2012). To address the discrepancies among the computational methods, we calculated the infrared spectra of carbonyl oxides (CIs) containing 1 to 3

carbon atoms as well as those with 7 carbon atoms mentioned in the manuscripts using the method described above.

**Table R1 The characteristic infrared peaks of CIs were obtained using different computational methods.**

| CIs | Calculated band/(cm$^{-1}$) | | | | Experimental band/ (cm$^{-1}$) [a] | Assignment |
|---|---|---|---|---|---|---|
| | B3LYP-D3(BJ)/ 6−311++G(d,p) | B3LYP-D3(BJ)/ aug-cc-pVTZ | **B3LYP/6-311G++(d,2p)** | B3LYP/6-311++G(2d,2p) | | |
| Anti-CH$_3$CHOO | 961 | 968 | **961** | 971 | **884** | O-O stretch |
| Syn-CH$_3$CHOO | 884 | 897 | **882** | 908.17 | **871.2** | O-O stretch |
| (CH$_3$)$_2$COO | 912 | 925 | **909** | 928 | **887.4** | O-O stretch |
| CH$_2$OO | 899 | 912 | **901** | 923 | **909** | O-O stretch |
| Syn-C7-CIs | 883 | 893 | **882** | 900 | \ | O-O stretch |
| Anti-C7-CIs | 912 | 923 | **910** | 928 | \ | O-O stretch |

Note: [a] Data from references (Chhantyal-Pun et al., 2020a).

As shown in Table R1, the table correspondingly list only the peak positions associated with this vibration because of the manuscript focused solely on the most intense O-O stretching vibration of CIs. Among the four computational methods, the B3LYP/6-311G++(d,2p) level demonstrated superior performance in predicting the O-O stretching vibrational frequencies of CIs. The results from this method were also in close agreement with the computational values reported previously by Li et al (Lin et al., 2015; Su et al., 2013; Wang et al., 2016). Thus, we have recalculated the infrared spectra of all relevant configurations in our manuscript using the B3LYP/6-311G++(d,2p) method and incorporated the necessary revisions. It could be observed that the calculated peak positions for alkyl-substituted CIs were systematically overestimated compared to the available experimental values. For example, a significant discrepancy of 77 cm$^{-1}$ was noted for anti-CH$_3$CHOO. In contrast, the deviation was much smaller for syn-CH$_3$CHOO being only 10.8 cm$^{-1}$. This was consistent with the trend previously observed by Li et al (Lin et al., 2015). Therefore, the computational results provided strong support for the identification of syn-C7-CIs.

Due to the discrepancies between the calculated and experimental infrared spectra, the computational results were not the sole basis for our peak assignments. For example, we observed that POZs and CIs appear sequentially. The characteristic peaks of POZs emerged first at lower temperatures. As the temperature increased, the infrared signals of CIs, along with those of the concurrently formed aldehydes, then became detectable. In addition, the characteristic IR peak ranged for CIs and POZs have been summarized from the literature, thereby providing a key basis for our spectral assignments.

**Lines 14**, we changed "Two CIs with different molecular sizes, C3-CIs and C7-CIs, are captured at 880

and 905 cm$^{-1}$ by using MI-FTIR." to "*C3-CIs are captured at 880 cm$^{-1}$ by using MI-FTIR.*".

**Lines 136-139**, we modified "The geometries of the SCIs were optimized using the hybrid density functional theory B3LYP-D3(BJ) with the aug-cc-pVTZ basis set. Harmonic vibrational frequencies were calculated at B3LYP/6-311G++(d,2p) calculation level for the comparison with the experimental infrared peak. Various computational levels for CIs were compared, and the one with superior performance was selected accordingly. Please refer to Table S1 for the specific comparison." to "*The geometries of the myrcene, POZs and SCIs were optimized using the hybrid density functional theory B3LYP-D3(BJ) with the aug-cc-pVTZ basis set. Harmonic vibrational frequencies were calculated at B3LYP/6-311G++(d,2p) calculation level for the comparison with the experimental infrared peak. Various computational levels for CIs were compared, and the one with superior performance was selected accordingly. Please refer to Table S1 for the specific comparison.*".

Correspondingly, Table R1 has been added to the supporting information (Table S1). The following explanation was added beneath the Table S1: "*As shown in Table R1, the table correspondingly list only the peak positions associated with this vibration because of the manuscript focused solely on the most intense O-O stretching vibration of CIs. Among the four computational methods, the B3LYP/6-311G++(d,2p) level demonstrated superior performance in predicting the O-O stretching vibrational frequencies of CIs. The results from this method were also in close agreement with the computational values reported previously by Li et al (Lin et al., 2015; Su et al., 2013; Wang et al., 2016). Thus, we have recalculated the infrared spectra of all relevant configurations in our manuscript using the B3LYP/6-311G++(d,2p) method and incorporated the necessary revisions. It could be observed that the calculated peak positions for alkyl-substituted CIs were systematically overestimated compared to the available experimental values. For example, a significant discrepancy of 77 cm$^{-1}$ was noted for anti-CH$_3$CHOO. In contrast, the deviation was much smaller for syn-CH$_3$CHOO being only 10.8 cm$^{-1}$. This was consistent with the trend previously observed by Li et al (Lin et al., 2015). Therefore, the computational results provided strong support for the identification of syn-C7-CIs.*".

**Lines 187-192**, we modified "The calculated O-O stretching vibrations of C7-CIs were located at 892 (syn-C7-CIs) and 923 (anti-C7-CIs) cm$^{-1}$, which were covered by the main characteristic infrared vibration peaks of myrcene. Consequently, it was challenging to directly observe the strongest characteristic infrared vibration peaks belonging to C7-CIs. In addition to the -COO group, C7-CIs also possessed conjugated double bonds as characteristic functional groups. In both syn- and anti-C7-CIs, the

second most intense infrared vibration band consistently corresponded to the wagging vibration of the =CH$_2$ group on the conjugated moiety. The calculated position of this vibration agreed with that of myrcene. Position 905 cm$^{-1}$ was the infrared characteristic peak generated by the wagging vibration of myrcene =CH$_2$, as obtained by twin-jet method. In the spectra, a significant relative increase at the 905 cm$^{-1}$ position was observed after annealing to 55 K. This increase might be attributable to the generation of C7-CIs. The characteristic infrared peaks of C3- and C7-CIs were both generated after 55 K annealing." to "*The calculated O-O stretching vibrations of C7-CIs were located at 882 (syn-C7-CIs) and 910 (anti-C7-CIs) cm$^{-1}$. According to Table S1, the calculated IR peaks of CIs were consistently overestimated by over 10 cm$^{-1}$ relative to experimental values, with the deviation being most pronounced for anti-CIs. Notably, no distinct new peaks appeared below 882 cm$^{-1}$ in the experiment. This absence suggested that very few stabilized C7-CIs were likely generated during the ozonolysis of myrcene. Instead, most C7-CIs were consumed via unimolecular decay pathways, a conclusion which was also supported by our subsequent analysis.*".

**Lines 178-179**, we modified "The calculation results indicated that the C-O stretching vibrations of the two POZs were located at positions 1068 and 1174 cm$^{-1}$." to "*The calculation results indicated that the C-O stretching vibrations of the two POZs were located at positions 1059 and 1165 cm$^{-1}$.*".

**Lines 202**, the Table 1 was changed as shown below.

*Table 1 Identification and assignments of experimental absorption bands in the initial ozonolysis of myrcene.*

| Experimental bands/(cm$^{-1}$) | Calculated Band/(cm$^{-1}$) | Reference | Belonger | Assignment |
|---|---|---|---|---|
| 765 | \ | 772 | POZs | O-O-O str. |
| 880 | \ | 887 [a] | C3-CIs | O-O str. |
| 905 | | | 4-vinyl-4-pentenal | =CH$_2$ wag. |
| 1074 | 1054 | \ | POZs | C-O str. |
| 1177 | 1165 | \ | POZs | C-O str. |
| 1370 | \ | 1370 [b] | Acetone | $\delta$ CH$_3$ |
| 1720 | \ | 1721.6 [b] | Acetone | C=O str. |

**Lines 345-348**, in the Conclusion part, we changed "This study employed the MI-FTIR method to directly determine the presence of C3-CIs and C7-CIs during the myrcene ozonolysis. The characteristic infrared vibration peak of C7-CIs, caused by the =CH$_2$ wagging vibration on the conjugated double bonds, was located at 905 cm$^{-1}$." to "*This study employed the MI-FTIR method to determine the presence of C3-CIs*

*and C7-CIs during the myrcene ozonolysis. The O-O stretching vibration peak of C3-CIs was located at 880 cm$^{-1}$. The absence of characteristic infrared peaks belonged to C7-CIs, likely due to their low steady-state yield or rapid unimolecular decay. The characteristic IR peaks of acetone indirectly confirmed the production of C7-CIs.".*

Minor issues

1. Figure 1: Make the font of this figure consistent with those of the other figures in the entire text.

**The author's answer:** Thank you for your comments. We have modified Figure 1. The revised Figure 1 was as follows:

**Lines 161**,

*Figure 1 Proposed the key pathway in the initial ozonolysis of myrcene. The values in parentheses represent the yields of the corresponding products.*

2. Figure 2: How to confirm that the 905 cm$^{-1}$ peak is the characteristic IR peak for C7-CIs, given the overlap with the parent peak and the presence of other products with conjugated double bonds, such as POZ.

**The author's answer:** Thank you for your comments. First, the spectra of a single precursor (myrcene/Ar or O$_3$/Ar) were conducted at different temperatures. The results showed that temperature variation did not alter the intensity of the infrared characteristic peak of myrcene or O$_3$ at 905 cm$^{-1}$ as shown in Figure S1(a). Based on a re-examination of the characteristic IR peaks of C7-CIs, we concluded that no such

peaks were observed in our experiments. However, in the spectrum from the twin-jet experiment at 55 K, the relative peak intensity at this wavenumber increased significantly. This indicated the formation of new species exhibiting a characteristic vibration at this location. The characteristic IR bands of C3-CIs could only be detected at 55 K, and the intensity change of the peak at 905 cm$^{-1}$ followed this consistent behavior. Therefore, the 905 cm$^{-1}$ position might be attributed to the 4-vinyl-4-pentenal (C7-aldehyde) which also contained conjugated double bonds.

The characteristic IR peaks at 892 and 905 cm$^{-1}$ for myrcene were due to the wagging vibrations of the =CH$_2$ groups terminating its conjugated diene system. The characteristic IR peaks for POZs emerged at a lower temperature (35 K) compared to CIs (55 K). Within the 35-45 K range, the peak at 892 cm$^{-1}$ remained more intense than that at 905 cm$^{-1}$ (Figure R4), suggesting the former was more likely due to the =CH$_2$ wagging of POZs. Consequently, the 905 cm$^{-1}$ feature was assigned to C7- aldehyde based on its distinct temperature-dependent behavior.

[Figure]

**Figure R4 The twin-jet spectra of myrcene ozonolysis reaction in a low temperature and Ar matrix after annealing to 35 K, 45 K and 55 K. T-J means the spectra obtained by twin-jet method.**

**Lines 148-149**, we changed "The products generated along with C7-CIs and C3-CIs were acetone and 4-vinyl-4-pentenal with yields of 0.27 and 0.73, respectively." to "*The products generated along with C7-CIs and C3-CIs were acetone and 4-vinyl-4-pentenal (C7-aldehyde) with yields of 0.27 and 0.73, respectively.*".

**Lines 195-196**, we added "*The peak at 905 cm$^{-1}$ coincided in temperature with the C3-CIs peaks and suggested its assignment to the =CH$_2$ wagging vibration of the C7-aldehyde.*".

3. Regarding the matrix isolation experiment, what are the scientific considerations governing the choice of the specific temperature window, namely 35-55 K?

**The author's answer:** Thank you for your comments. For traditional matrix isolation experiments that used the inert gas argon as the matrix, the commonly used temperature range was $\leq$ 35 K. In our myrcene ozonolysis experiment, we advanced beyond the conventional matrix isolation scheme. By elevating the matrix temperature, the precursors overcame the confinement of the matrix "cage". This allowed the precursors to make more sufficient contact and react more thoroughly, thereby leading to a significant enhancement of the infrared characteristic peaks of the intermediates and products. However, gradual heating under the high vacuum conditions of matrix isolation led to precursor loss, prompting us to cap the maximum temperature at 55 K.

This approach was echoed by existing studies. For instance, Yu et al., while investigating the ozonolysis of styrene using the same methodology, employed a temperature range of 6 K to 57 K. They observed the most intense characteristic peaks of the intermediates and products precisely at 57 K (Yu et al., 2025). In studying the ozonolysis mechanism of tetramethylethylene, Yang et al. adopted a higher experimental temperature to detect final products (Yang et al., 2020).

**Lines 107**, in the 2.1 Matrix isolation experiment part, we cited the two above-mentioned references in the following sentence, "*To promote the further occurrence of the reaction and further soften and diffuse the matrix, the matrix was further heated to 45 and 55 K (Yu et al., 2025; Yang et al., 2020).*".

4. Section 3.2, How to confirm that SCIs first react with $RO_2$ and then with $HO_2$, rather than $RO_2$ first reacting with $HO_2$ to form ROOH, which then reacts with SCIs.

**The author's answer:** Thank you for your comments. Current evidence showed that the reaction between SCIs and $RO_2$ was considerably faster than that between SCIs and ROOH. Zhao et al. used quantum chemical calculations to compare the free energy barriers and found that the formation of ROO-SCIs-H prefers to follow the $RO_2$ + SCIs + $HO_2$ reaction rather than the ROOH + SCIs reaction (Zhao et al., 2017). The reaction rate between $CH_2OO$ and $CH_3O_2$ was determined by UV-vis spectroscopy to be $(1.7 \pm 0.5)\times10^{-11}$ $cm^3$ $s^{-1}$ at 294 K and 10 Torr (Chao et al., 2024). Chhantyal-Pun et al. measured the reaction rate of $CH_2OO$ with $CH_3O_2/CH_3C(O)O_2$ using cavity ring-down spectroscopy, obtaining a value of $(2.4 \pm 1.2)\times10^{-11}$ $cm^3$ $molecule^{-1}$ $s^{-1}$ (Chhantyal-Pun et al., 2020b). However, the reaction rates of $CH_2OO$ with

$H_2O_2$ and $(CH_3)_3COOH$ (peroxides) were determined to be $(2.2\pm0.9) \times 10^{-13}$ and $(1.2^{+1.2}_{-0.6}) \times 10^{-13}$ $cm^3$ molecule$^{-1}$ s$^{-1}$, respectively (Caravan et al., 2024). And existing studies indicated that oligomers were produced via the $RO_2 + SCIs + HO_2$ mechanism, as opposed to the $RO_2 + HO_2 + SCIs$ mechanism (Yu et al., 2025; Zhao et al., 2015). Therefore, we concluded that the predominant formation pathway for oligomers proceeded via $RO_2 + SCIs + HO_2$ rather than $RO_2 + HO_2 + SCIs$.

**Lines 258-260**, we added "*The rate constants for the reaction of $RO_2$ radicals with SCIs were much higher than that for ROOH (Zhao et al., 2017; Chao et al., 2024; Chhantyal-Pun et al., 2020; Caravan et al., 2024).*".

5. The authors characterized the particulate phase composition and detected oligomers within it. How can it be demonstrated that these oligomers were formed in the gas phase rather than in the particle phase? For instance, oligomers can also be formed by hemiacetal reactions within the particle phase. How is this particle-phase pathway accounted for?

**The author's answer:** Thank you for your comments. The rate constant for reactions between CIs and organic acids approached the collision limit, approximately $10^{-10}$ $cm^3$ molecule$^{-1}$ s$^{-1}$, while the rate constant for reactions with $RO_2$ radicals was on the order of $10^{-11}$ $cm^3$ molecule$^{-1}$ s$^{-1}$. These exceptionally high gas-phase rate constants resulted in the near-complete consumption of CIs in the gas phase, leading to their conversion into less volatile products that subsequently partition into the particle phase. Several studies have identified CIs oligomerization as a contributor to particle formation, and categorizing this reaction pathway as a gas-phase process (Yu et al., 2025; Zhao et al., 2015; Caravan et al., 2024). The mass spectral signals of the oligomers exhibited an enhancing trend following the introduction of the OH scavenger. This demonstrated that the oligomers predominantly originated from CIs-derived channels.

A substantial contribution from the hemiacetal reaction to the particle phase typically necessitated an acid-catalyzed system, involving the introduction of acids (Zhao et al., 2015). As this was not the case in our study, the oligomers were consequently not attributed to acid catalysis.

6. Equation R8 contains an error in its chemical formula.

**The author's answer:** Thank you for your comments. We have modified Formula R8. The revised R8 is as follows.

**Lines 291,**

$C_7H_9O_3$ + C3-CIs → $C_{10}H_{15}O_5$ + C3-CIs → $C_{13}H_{21}O_7$ + C3-CIs → $C_{16}H_{27}O_9$

↓ + $RO_2$    ↓ + $RO_2$    ↓ + $RO_2$    ↓ + $RO_2$

$C_7H_{10}O_2$ + C3-CIs → $C_{10}H_{16}O_4$ + C3-CIs → $C_{13}H_{22}O_6$ + C3-CIs → $C_{16}H_{28}O_8$

(R8)

**References**

Caravan, R. L., Bannan, T. J., Winiberg, F. A. F., Khan, M. A. H., Rousso, A. C., Jasper, A. W., Worrall, S. D., Bacak, A., Artaxo, P., Brito, J., Priestley, M., Allan, J. D., Coe, H., Ju, Y., Osborn, D. L., Hansen, N., Klippenstein, S. J., Shallcross, D. E., Taatjes, C. A., and Percival, C. J.: Observational evidence for Criegee intermediate oligomerization reactions relevant to aerosol formation in the troposphere, Nat. Geosci., 17, 219–226, 10.1038/s41561-023-01361-6, 2024.

Chao, W., Markus, C. R., Okumura, M., Winiberg, F. A. F., and Percival, C. J.: Chemical Kinetic Study of the Reaction of $CH_2OO$ with $CH_3O_2$, J. Phys. Chem. Lett., 15, 3690-3697, 10.1021/acs.jpclett.4c00159, 2024.

Chhantyal-Pun, R., Khan, M. A. H., Taatjes, C. A., Percival, C. J., Orr-Ewing, A. J., and Shallcross, D. E.: Criegee intermediates: production, detection and reactivity, Int. Rev. Phys. Chem., 39, 383-422, 10.1080/0144235x.2020.1792104, 2020a.

Chhantyal-Pun, R., Khan, M. A. H., Zachhuber, N., Percival, C. J., Shallcross, D. E., and Orr-Ewing, A. J.: Impact of Criegee Intermediate Reactions with Peroxy Radicals on Tropospheric Organic Aerosol, Acs Earth Space Chem., 4, 1743-1755, 10.1021/acsearthspacechem.0c00147, 2020b.

Coleman, B. E. and Ault, B. S.: Matrix isolation investigation of the ozonolysis of propene, J. Mol. Struct., 976, 249-254, https://doi.org/10.1016/j.molstruc.2010.03.050, 2010.

Coleman, B. E. and Ault, B. S.: Investigation of the mechanism of ozonolysis of (Z)-3-methyl-2-pentene using matrix isolation infrared spectroscopy, J. Mol. Struct., 1031, 138-143, https://doi.org/10.1016/j.molstruc.2012.07.046, 2013.

Deng, J., Chen, J., Liu, H., Wang, W., and Wang, X.: Matrix Isolation FT-IR Study on the Reaction Mechanisms between Ozone and Ethene, Res. Environ. Sci., 25, 1-9, 2012.

Hoops, M. D. and Ault, B. S.: Matrix Isolation Study of the Early Intermediates in the Ozonolysis of Cyclopentene and Cyclopentadiene: Observation of Two Criegee Intermediates, J. Am. Chem. Soc., 131, 2853-2863, 10.1021/ja8065286, 2009.

Lin, H.-Y., Huang, Y.-H., Wang, X., Bowman, J. M., Nishimura, Y., Witek, H. A., and Lee, Y.-P.: Infrared identification of the Criegee intermediates syn- and anti-$CH_3CHOO$, and their distinct conformation-dependent reactivity, Nat. Commun., 6, 7012, 10.1038/ncomms8012, 2015.

Lu, T. and Chen, Q.: Interaction Region Indicator: A Simple Real Space Function Clearly Revealing Both Chemical Bonds and Weak Interactions, Chem. Method, 1, 231-239, 10.1002/cmtd.202100007, 2021.

Lv, C., Du, L., Tang, S. S., Tsona, N. T., Liu, S. J., Zhao, H. L., and Wang, W. X.: Matrix isolation study of the early intermediates in the ozonolysis of selected vinyl ethers, Rsc Adv., 7, 19162-19168, 10.1039/c7ra01011g, 2017.

Pinelo, L., Gudmundsdottir, A. D., and Ault, B. S.: Matrix Isolation Study of the Ozonolysis of 1,3-and 1,4-Cyclohexadiene: Identification of Novel Reaction Pathways, J. Phys. Chem. A, 117, 4174-4182, 10.1021/jp402981n, 2013.

Su, Y.-T., Huang, Y.-H., Witek, H. A., and Lee, Y.-P.: Infrared Absorption Spectrum of the Simplest Criegee Intermediate $CH_2OO$, Science, 340, 174-176, 10.1126/science.1234369, 2013.

Wang, Y.-Y., Chung, C.-Y., and Lee, Y.-P.: Infrared spectral identification of the Criegee intermediate $(CH_3)_2COO$, J. Chem. Phys., 145, 154303, 10.1063/1.4964658, 2016.

Yang, X., Deng, J., Li, D., Chen, J., Xu, Y., Zhang, K., Shang, X., and Cao, Q.: Transient species in the ozonolysis of tetramethylethene, J. Environ. Sci., 95, 210-216, 10.1016/j.jes.2020.03.027, 2020.

Yu, S. S., Tong, S. R., Chen, M. F., Zhang, H. L., Xu, Y. Y., Guo, Y. C., and Ge, M. F.: Characterization of Key Intermediates and Products from the Ozonolysis of Styrene-Like Compounds, Environ. Sci. Technol., 10.1021/acs.est.5c00769, 2025.

Zhao, Q., Wang, W., Liu, F., Lu, J., and Wang, W.: Oligomerization reactions for precursors to secondary organic aerosol:

Comparison between two formation mechanisms for the oligomeric hydroxyalkyl hydroperoxides, Atmos. Environ., 166, 1-8, 10.1016/j.atmosenv.2017.07.008, 2017.

Zhao, Y., Wingen, L. M., Perraud, V., Greaves, J., and Finlayson-Pitts, B. J.: Role of the reaction of stabilized Criegee intermediates with peroxy radicals in particle formation and growth in air, Phys. Chem. Chem. Phys., 17, 12500-12514, 10.1039/c5cp01171j, 2015.

---

## Author Comment (AC3)

**Response to reviewers' comments on "Elucidation of the myrcene ozonolysis mechanism from a Criegee Chemistry perspective"**

**Response to Reviewer #3**

**Significant**

Formation of condensable chemicals initiated by gas-phase ozonolysis of volatile organic compounds (VOCs) contributes significantly to atmospheric secondary organic aerosol (SOA) budgets. The molecular level mechanism is generally understood to evolve through the well-known Criegee mechanism which produces two distinct zwitterion / diradical species called Criegee intermediates (CIs). These CIs are known to have a rich uni- and bimolecular chemistry, commonly dominated by unimolecular decomposition and reaction with water dimer under atmospheric conditions. The current study focuses on myrcene, an often-overlooked acyclic monoterpene with three double-bonds, and reports the apparent importance of its two CIs, which dominate the initial oxidation product distribution, on atmospheric SOA formation. The work utilizes complementary investigation methods from matrix isolation and chamber investigations to quantum chemical computations and attempts to understand myrcene oxidation chemistry by synthesizing the output from these distinct methodologies.

While the topic of the work is certainly of interest to the readers of ACP, in the current form it is difficult to assess what has actually been accomplished here. Specifically, the current level of documentation does not allow to fully assess the reliability of the results as the methods and results appear only partly described. Also, to me it seems that the type of oligomerization reaction described here would be seriously kinetically limited in the atmosphere, and thus I do suspect there could be some easier explanation for the observed product signals. I'll detail my concerns below.

Thank you for your positive assessment of the significance of our work and for your constructive comments. We appreciate the opportunity to clarify and strengthen the manuscript. In response to your concerns regarding the documentation of methods and results, we have expanded the description of experimental procedures to ensure reproducibility and transparency. Regarding the kinetic feasibility of the oligomerization pathways under atmospheric conditions, these points have been addressed in detail in the responses to the specific questions below. In addition, the proposed reactions are further supported by the presentation of additional MS/MS evidence. All the comments are addressed point by point, with our responses in blue, and the corresponding revisions to the manuscript in red. We believe these revisions

address your concerns while preserving the core findings of the study. Thank you again for the thoughtful review, which has helped us improve the clarity and rigor of the manuscript.

**Major comments**

First the kinetic limitation: I suspect that most of the gas- and particle-phase results could be explained by the more common peroxy radical ($RO_2$) chemistry without the need to invoke exotic Criegee intermediate (CI) oligomerization reaction which is generally limited by the availability of the very reactive CIs in almost any conceivable Atmos. Environs. That is, the CIs simply cannot find an "already dimerized" reaction product to form a "further trimerized" product, because they react away in multitude of reactions with several of the co-produced oxidation products (e.g., any products with carbonyl groups). It is simply difficult to see how the CI concentration could ever be so high to permit sequential reactions with the same products under such reaction time and oxidizing conditions. To me it seems far more likely that recently uncovered pathways in $RO_2+R'O_2$ reactions, where RO (and R'O) radical rearrangement occurs after the initial peroxy radical cross-combination reaction and leads to products with various amounts of carbon and oxygen in the "dimeric" structures, is the explanation here. Please have a look at Peräkylä et. al. (https://pubs.acs.org/doi/full/10.1021/jacs.2c10398) and Frandsen et al. (https://pubs.acs.org/doi/full/10.1021/acsearthspacechem.4c00355) for the mechanism and topical examples. Could they explain what is seen here? Again, it is hard to see how the numbers would match and allow the oligomerization to happen, and that's why I urge the Authors to back up the current sequential oligomerization conclusions by a gas-phase kinetic modelling of the relevant reaction system using some prototypical reaction rate coefficients.

**The author's answer:** Thank you for this thorough and critical assessment, which highlights the central kinetic challenges and offers a compelling alternative perspective based on established $RO_2$ chemistry. Our response below integrates your points and aims to provide evidence for the occurrence of CIs oligomerization.

Based on current research, the reaction between $RO_2$ and CIs is kinetically feasible. The reaction rate constant between $CH_2OO$ and $CH_3OO$ was $(1.7 \pm 0.5) \times 10^{-11}$ $cm^3$ $s^{-1}$ (https://doi.org/10.1021/acs.jpclett.4c00159) or $(2.4 \pm 1.2) \times 10^{-11}$ $cm^3$ $molecule^{-1}$ $s^{-1}$ (https://dx.doi.org/10.1021/acsearthspacechem.0c00147). Meanwhile, the rate constant for the $CH_3OO +$ $CH_3OO$ reaction was $(2.0 \pm 0.9) \times 10^{-13}$ $cm^3$ $s^{-1}$. The reaction rate between $CH_2OO$ and $RO_2$ exceeded that of the $RO_2 + RO_2$ reaction by almost two orders of magnitude (Chao et al., 2024; Chhantyal-Pun et

al., 2020b). Chhantyal-Pun et al. simulated the contribution of CIs + $RO_2$ reactions to regional SOA formation using the STOCHEM-CRI model. The model adopted a rate constant of $2.4 \times 10^{-11}$ $cm^3$ $s^{-1}$ for the CIs + $RO_2$ reaction and $9.2 \times 10^{-14} \times 0.7$ $cm^3$ $s^{-1}$ for the $RO_2$ + $RO_2$ reaction. The combination of the direct kinetic measurements and atmospheric modeling suggested that CIs reactions with peroxy radicals made non-negligible (up to a percent) contributions to SOA production in the forested regions of the world. (https://dx.doi.org/10.1021/acsearthspacechem.0c00147) (Chhantyal-Pun et al., 2020b). Moreover, this study considered the CIs generated from monoterpene ozonolysis without those produced from the ozonolysis of compounds such as isoprene.

The kinetic feasibility of additional CIs oligomerization steps was supported by available experimental data. $CH_2OO$ could react with water to form $HO-CH_2OO-H$ (step 1 in the oligomerization). Chen et al. reported a rate constant of $5.4 \times 10^{-12}$ $cm^3$ $molecule^{-1}$ $s^{-1}$ for the reaction between $CH_2OO$ and $HO-CH_2OO-H$ at 298 K (step 2 in the oligomerization) (Chen et al., 2019). $CH_2OO$ could react with formic acid to form $HC(O)O-CH_2OO-H$ (step 1 in the oligomerization). Caravan et al. (https://doi.org/10.1038/s41561-023-01361-6) reported a rate constant of $3.2 \times 10^{-12}$ $cm^3$ $molecule^{-1}$ $s^{-1}$ for the reaction between $CH_2OO$ and $HC(O)O-CH_2OO-H$ (step 2 in the oligomerization) (Caravan et al., 2024). These rate values suggested that oligomerization reactions involving two Criegee intermediates were kinetically competitive with the $RO_2$ + $RO_2$ reaction.

Oligomers containing CIs structural units have also been observed in laboratory studies of VOC ozonolysis. The ozonolysis of ethylene could simultaneously produce $CH_2OO$ and formic acid. In the laboratory study by Yosuke Sakamoto et al. (https://pubs.acs.org/doi/10.1021/jp408672m), the formation of oligomers using $CH_2OO$ as the repeating unit was also observed during the ozonolysis of ethylene (Sakamoto et al., 2013). Moreover, the generation of oligomers was significantly suppressed after adding the scavenger of CIs. This also demonstrated that the formation of these substances originated from the oligomerization of CIs. Rousso and colleagues also identified multiple sequences of $CH_2OO$ additions in laboratory studies of ethene ozonolysis using a jet-stirred reactor (JSR) (https://pubs.rsc.org/en/content/articlelanding/2019/cp/c9cp00473d) (Rousso et al., 2019). Zhao et al. have observed oligomers formed from the reaction of the $CH_3CH_2CHOO$ with $RO_2$ radicals during the ozonolysis of *trans*-3-hexene (https://pubs.rsc.org/en/content/articlelanding/2014/cp/c4cp02747g) (Zhao et al., 2015).

Furthermore, the results from field observations constitute direct evidence. A study by Caravan et al. from Argonne National Laboratory, published in *Nature Geoscience*, reported the detection of oligomers formed from $CH_2OO$ and formic acid in the Amazon rainforest using FIGAERO-CIMS, with the degree

of oligomerization reaching up to 6. Furthermore, this study detected oligomers with $CH_2OO$ as the chain unit in both the gas and particle phases. The degree of oligomerization measured in the gas phase ranged from 2 to 6, while that in the aerosols ranged from 3 to 5. The corresponding modeling simulation further estimated that the global concentration of oligomers formed from the reaction of two $CH_2OO$ with formic acid is in the range of 0.005–0.01 ppt. This underestimate of the modeled value (0.005–0.01 ppt) compared to the field-observed concentration (with a peak of ~2.0 ppt) was attributed to the current gaps in Criegee chemical mechanisms and missing reaction rate constants (Caravan et al., 2024). The model underestimate provided direct evidence for the importance of CIs oligomerization in atmospheric chemistry, challenging the reviewer's notion that such reactions were highly unlikely to occur.

In addition, Luo et al. also detected products at the Station for Measuring Ecosystem-Atmosphere Relations (SMEAR II) in Hyytiälä, southern Finland, which likely originated from the reactions of SCIs (generated from monoterpene ozonolysis) with organic acids (https://doi.org/10.5194/acp-25-4655-2025) (Luo et al., 2025).

There is no doubt that $RO_2$ chemistry plays a significant role in SOA formation. However, it is not the focus of our study. In analyzing the experimental results, the $RO_2$-related chemistry was thoroughly considered, however, it did not adequately account for the formation of the oligomers observed in our experiments. Regarding the $RO_2 + RO_2 \rightarrow ROOR + O_2$ mechanism cited by the reviewer from Peräkylä et al. (https://pubs.acs.org/doi/full/10.1021/jacs.2c10398), its potential role in the formation of dimers in the system was undeniable. The initial ozonolysis of myrcene involved cleavage of the carbon chain, which led to the formation of C7-$RO_2$ and C3-$RO_2$ radicals through subsequent transformation. According to the mechanism proposed by the reviewers, the reaction between C7-$RO_2$ and C7-$RO_2$ can only form C14 compounds at most. Thus, the $RO_2 + RO_2$ reaction mentioned by the reviewer could potentially explain the formation of compounds with ≤C14 in the myrcene ozonolysis, especially after the addition of an OH scavenger. Regarding the formation of larger molecular-size compounds such as C16 species, we were currently unable to provide a definitive explanation. Moreover, for the oligomer sequence (C7, C10, C13, and C16) proposed in our manuscripts, we have presented MS/MS evidence that supports our conclusions.

We have added Figure S4 to the Supporting Information (SI). Figure S4 presented the MS/MS spectra of $C_{10}H_{16}O_4$, $C_{13}H_{22}O_6$, and $C_{16}H_{28}O_8$. The major fragment peaks of $C_{13}H_{22}O_6$ and $C_{16}H_{28}O_8$ were highly similar. The presence of fragments such as $C_5H_7^+$ (m/z 67.06), $C_7H_7O^+$ (m/z 95.05), and $C_6H_9O_2^+$ (m/z 113.06) indicated that the oligomers in Sequence 2 possessed a conjugated double-bond skeleton. This suggested that this sequence was initiated by C7 compounds. The fragment at m/z 71.05 ($C_4H_7O^+$)

was generated via α-cleavage adjacent to a carbonyl group in the oligomer backbone. This fragmentation was typical of compounds containing conjugated carbonyl motifs and further supported the presence of a C7-based conjugated skeleton in the sequence. The $C_3H_7O^+$ (m/z 59.05) fragment ions originated from cleavage of terminal -OH functionalized tertiary carbon moieties in the oligomers. The formation of this ion fragment was similar to that in Sequence 1. In the MS/MS spectra of $C_{10}H_{16}O_4$, we observed that besides the fragment peaks annotated in Figure S4, there existed fragment ion peaks with higher intensities. This suggested that $C_{10}H_{16}O_4$ was formed through multiple pathways. (such as the $RO_2 + RO_2$ reaction mentioned by the reviewer).

[Figure]

**Figure S4 The MS/MS of $C_7H_{10}O_2$ + n-C3-SCIs + $RO_2$ sequence (a) and chemical structures of ions corresponding to major fragment peaks in MS/MS spectra (b).**

**Lines 302-312**, we added "Figure S4 presented the MS/MS spectrum of Sequence 2 along with the proposed structures corresponding to the major fragment peaks. The presence of fragments such as $C_5H_7^+$ (m/z 67.06), $C_7H_7O^+$ (m/z 95.05), and $C_6H_9O_2^+$ (m/z 113.06) indicated that the oligomers in Sequence 2 possessed a conjugated double-bond skeleton. This suggested that this sequence was initiated by C7 compounds. The fragment at m/z 71.05 ($C_4H_7O^+$) was generated via α-cleavage adjacent to a carbonyl group in the oligomer backbone. This fragmentation was typical of compounds containing conjugated carbonyl motifs and further supported the presence of a C7-based conjugated skeleton in the sequence. The $C_3H_7O^+$ (m/z 59.05) fragment ions originated from cleavage of terminal -OH functionalized tertiary carbon moieties in the oligomers. The formation of this ion fragment was similar to that in Sequence 1. In the MS/MS spectra of $C_{10}H_{16}O_4$, we observed that besides the fragment peaks annotated in Figure S4, there existed fragment ion peaks with higher intensities. This suggested that $C_{10}H_{16}O_4$ was formed through multiple pathways (such as the $RO_2 + RO_2$ reaction) (Peräkylä et al., 2023; Frandsen et al., 2025).".

Regarding the mechanism mentioned in the work by Frandsen et al. (https://pubs.acs.org/doi/full/10.1021/acsearthspacechem.4c00355), which involved the rearrangement of RO (and R'O) radicals following initial peroxy radical cross-combination reactions, leading to the formation of "dimer" structures with varying carbon and oxygen content in the products. While acknowledging the mechanism described, it did not fully explain the formation of compounds larger than C14. Furthermore, we also accounted for the variation in the oxygen content of myrcene-derived products by incorporating $RO_2$ chemistry, including $RO_2$ autoxidation and the $RO_2 + RO_2 \rightarrow RO + RO + O_2$ reaction (Jokinen et al., 2014; Zhao et al., 2023; Liu et al., 2023; Benoit et al., 2023) as shown in Equation (R8) and Figure 5 of the manuscript.

We have also incorporated these two references into the manuscript. **Lines 311-312**, "This suggested that $C_{10}H_{16}O_4$ was formed through multiple pathways (such as the $RO_2 + RO_2$ reaction) (Peräkylä et al., 2023; Frandsen et al., 2025).".

**References**

**Lines 467-469**, "Frandsen, B. N., Franzon, L., Meder, M., Pasik, D., Ahongshangbam, E., Vinkvist, N., Myllys, N., Iyer, S., Rissanen, M. P., Ehn, M., and Kurtén, T. C.: Detailed Investigation of 2,3-Dimethyl-2-butene Ozonolysis-Derived Hydroxyl, Peroxy, and Alkoxy Radical Chemistry, ACS Earth Space Chem., 9, 1322-1337, 10.1021/acsearthspacechem.4c00355, 2025.".

**Lines 541-543**, "Peräkylä, O., Berndt, T., Franzon, L., Hasan, G., Meder, M., Valiev, R. R., Daub, C. D., Varelas, J. G., Geiger, F. M., Thomson, R. J., Rissanen, M., Kurtén, T., and Ehn, M.: Large Gas-Phase Source of Esters and Other Accretion Products in the Atmosphere, J. Am. Chem. Soc., 145, 7780-7790, 10.1021/jacs.2c10398, 2023.".

Second is the lack in documenta on: Many important details are missing, and the current story appears to choose the results across the very specific investigation methodologies without clearly referencing on what part of the study the results have been obtained. Please remember that the minimum amount of documenta on is always such that the work can be repeated and hence the results verified in a replicate study. Now, I am not sure how I could repeat the matrix isolation on work, which is likely the best described of the experimental procedures, though still appears to miss the temperature of the mixing jet and the details of mixing the reactants, the volume of the chamber, the timescale of deposition and reaction, the purity of the $O_3$ mixture, for example. The aerosol forma on study seems to be missing more details

including the timescale, the used reactant and sampling flows, the details of aerosol particle and gas compound detection, the details of the LC-MS technique and so on. The computations seem to be missing almost all details, and it is not clear what has been computed. You should give the computational details and the resulting molecular geometries in the SI. The current level of documenta on in the SI is inadequate, and you must add all the relevant details to be able to replicate the study.

**The author's answer:** Thank you for your thorough review and for highlighting the need for more detailed documentation across the experimental and computational methodologies. We appreciate the opportunity to improve the clarity and reproducibility of our work.

Every experimental procedure in this work, including matrix isolation and chamber studies, was repeated multiple times to confirm data robustness and reproducibility. We have also included a comparison of calculated IR spectra of CIs using different methods to justify our chosen level of theory. In response, we have added more experimental details to the manuscript and SI. We were confident that the level of detail now provided in the revised manuscript and SI was complete and will allow for the replication of our work.

Regarding the experimental details of matrix isolation, we have been clearly presented in the manuscript. First, matrix isolation was not merely a specific investigation methodology but a well-established technique that had been routinely employed to characterize reactive species such as free radicals (Hearne et al., 2019; Ryazantsev et al., 2017; Saraswat et al., 2025; Zasimov et al., 2022; Zhu et al., 2016) and POZs, CIs. The earliest application of matrix isolation coupled with vacuum-ultraviolet Fourier transform infrared spectroscopy was reported by Michael D. Hoops and Bruce S. Ault at the University of Cincinnati, who successfully detected key intermediates (POZs and CIs) during the ozonolysis of cyclopentene and cyclopentadiene (Hoops and Ault, 2009). Follow-up work by the group further employed this technique to detect intermediates in the ozonolysis of multiple VOCs (Coleman and Ault, 2010; Pinelo et al., 2013; Kugel and Ault, 2015, 2019). However, it should be noted that matrix isolation systems were not commercially available as fully integrated instruments. The system was custom-built according to experimental needs and was subsequently validated through actual experiments. Following the construction of our apparatus, the setup was validated through the ozonolysis of 2,3-dimethyl-2-butene (Yang et al., 2020). The infrared characteristic peaks of the products obtained showed perfect agreement with those reported in the literature. A comparison between our experimental spectrum and the corresponding literature spectrum was illustrated in Figure R1.

[Figure]

**Figure R1 Infrared spectra of a matrix formed by twin jet deposition of tetramethylethene and ozone. (a) was obtained by us. (b) came from the references (Yang et al., 2020). TME refers to 2,3-dimethyl-2-butene, and T-J denotes the twin-jet deposition method.**

Secondary, the details regarding the temperature of the mixing jet, the procedure for mixing the reactants, and the reaction timescale of deposition and reaction were all comprehensively addressed in the manuscript. **Lines 103-109**, "The deposition of myrcene/Ar and $O_3$/Ar onto the 6±1 K cold window was facilitated by two angled and independent tubes at a rate of 5 ml/min. This deposition was known as the twin-jet co-deposition mode. The deposition duration was approximately 120 min. To allow limit the diffusion and/or reaction of reactants, these matrices were heated or annealed to 35 K and held for 0.5 h,

and then cooled to 6±1 K after which the spectra were recorded. To promote the further occurrence of the reaction and further soften and diffuse the matrix, the matrix was further heated to 45 and 55 K (Yu et al., 2025; Yang et al., 2020). To prevent matrix loss, it was imperative to immediately cool down to 6±1 K after reaching the target temperature and to record the spectra.".

Compared to the descriptions in the existing literature, our description was essentially equally detailed (Yang et al., 2020; Lv et al., 2017; Coleman and Ault, 2013). The specific descriptions from these references were presented below.

The descriptions of Yang et al., "Ozone and tetramethylethene were mixed separately with Ar to the desired ratio (Mixed–gas/Ar = 1:100) in a 5 L Pyrex bulb (Fig. 2). The peek tube and the cold window were evacuated (about $10^{-5}$ Pa) with the temperature gradually dropping to 15 K and kept constant. The ozone/Ar and alkene/Ar were co-deposited (in a ratio of 2 mmol/h) onto a cold window (15 K) from two separate lines. After 2h of deposition, the infrared spectra of the sample were scanned. Meanwhile, the matrix was annealed to 35 K and held at this temperature for 1 h. During the process, the matrix solids softened and the Ar atoms began to diffuse. Then it was further heated to 40 K, and Ar started to evaporate rapidly. All Ar matrix on the cold window evaporated at 55 K. Although some of the reactants were carried away by the Ar gas, most of the reactants remained on the cold window to form a neat film due to the high melting and boiling point of ozone and TME. Finally, we turned off helium refrigerator and slowly warmed up the salt plate. The infrared spectrum of the obtained neat film was recorded for every 10 K until the cold window was slowly annealed to room temperature."

The descriptions of Chen, et al., "Twin jet mode was used in the depositing process, in which the $Ar/O_3$ and Ar/samples gas mixtures were fed to the cold window through separate ports in the cold head. A temperature programmed method was used to control reaction time. The two gas samples were deposited from separate jets onto the 14 K cold window at the rate of 2 mmol h$^{-1}$, allowing for only a very brief mixing time prior to matrix deposition. The matrices were subsequently warmed to 25, 30 and then to 35 K to permit limited diffusion and/or reaction. These matrices were then recooled to 14 K and additional spectra were recorded.".

The descriptions of Bridgett E. Coleman, et al., "Matrix samples were deposited in two different modes, twin jet and merged jet. In the first, the two gas samples were deposited from separate nozzles onto the 14 K window, allowing for only a brief mixing time prior to matrix deposition. The distance from the tip of each nozzle to the cold window was approximately 3 cm. Several of these matrices were subsequently warmed to 33–35 K to permit limited diffusion and then recooled to 14 K and additional spectra recorded.".

We would like to clarify for the reviewer that our setup employed a twin-jet method, not a mixing jet. Nowhere in the manuscript or the SI was the term "mixing jet" mentioned. In matrix isolation, twin-jet and merged-jet represented two entirely distinct sampling configurations. The work by Bruce S. Ault's group on matrix isolation of key ozonolysis intermediates from cyclopentene and cyclopentadiene included a discussion of the two separate sampling configurations employed (J. Am. Chem. Soc. 2009, 131, 2853–2863) (Hoops and Ault, 2009). The detailed description was as follows. "Matrix samples were deposited in both the twin jet and merged jet modes. In the former, the two gas samples were deposited from separate nozzles onto the 14 K window, allowing for only a very brief mixing time prior to matrix deposition. Several of these matrices were subsequently warmed or annealed to 25 and then to 35 K to permit limited diffusion and/or reaction. These matrices were then recooled to 14 K and additional spectra recorded. Many experiments were conducted in the merged jet mode, in which the two deposition lines were joined with an Ultra Torr tee at a predetermined distance from the cryogenic surface, and the flowing gas samples were permitted to mix and react during passage through the merged region. The length of this region can be varied and was ~50 cm in length for this study.". In our experiments, only the twin-jet deposition method was employed. A schematic diagram of this sampling configuration was provided in Figure R2.

[Figure]

**Figure R2 Schematic diagram of twin-jet method.**

For the volume of the chamber, the chamber volume was typically not reported in the matrix isolation literature for CIs characterization. The reaction chamber was purchased from JANIS (Model ST-100). Figure R3 also provided a physical diagram of this chamber.

[Figure]

**Figure R3 The matrix chamber.**

Regarding the purity of the $O_3$ mixture, we noted that it was subjected to multiple purification steps as mentioned in the manuscript. **Lines 97-98**, "The collected $O_2/O_3$ mixture was frozen in liquid nitrogen and undergoes several cycles of freezing pump - thawing to remove the residual impurity gas. $O_3$ or myrcene.". This purification method has been widely adopted in matrix studies (Wang, 2020; Tang et al., 2017; Hoops and Ault, 2009). Furthermore, our $O_2/O_3$ mixture was generated via the discharge of high-purity oxygen (≥99.999%), ensuring it was free from NOx interference. Furthermore, the pure $O_3/Ar$ reference spectrum provided in the SI showed no detectable impurity peaks that could interfere with the experiment.

**Lines 114-115**, we added "An $O_2/O_3$ mixture was generated by passing high-purity $O_2$ (≥99.999%) at a flow rate of 200 mL/min through an ozone generator (Beijing Tonglin Technology Co., Ltd.).".

In response to the reviewer's comments regarding the lack of chamber experimental details, we have provided additional clarifications in the revised manuscript.

**Lines 118-150**, The measurement of the myrcene concentration was conducted by means of a gas chromatograph with a quadrupole mass spectrometer (GC-MS, Agilent, 7890, 5977B) equipped thermal desorption instrument (TD). The GC-MS is on an HP-5MS column (30 m × 0.25 × 5 mm) with helium as carrier gas and a flow rate of 1.2 mL/min. The temperature of the chromatographic column was set as follows: the initial temperature was 40 ℃ and held for 3 minutes. The temperature was then increased to 140 ℃ by maintaining a rate of 20 ℃/min; the program was then finished by increasing the temperature to 200 ℃ at a rate of 25 ℃/min. Organics were quantified with mode selection SIM and 41 m/z, 69 m/z

and 93 m/z were selected as the characteristic ions detected by mass spectrometry for myrcene. The maintenance of different RH levels was achieved by the implementation of a 10 L/min flow of zero air through the water bubbler. The indoor temperature and RH were measured using a hygrometer (Vaisala, HMP3). Throughout each experiment, size distributions and volume concentrations of particles were continuously recorded using a scanning mobility particle sizer (SMPS), which consisted a differential mobility analyzer (DMA, TSI, Model 3081) and a condensation particle counter (CPC, TSI, Model 3776). The SMPS measured particles every 3 minutes across a size range of 14.3 to 723.4 nm. A sampling flow of 0.3 L min$^{-1}$ and a sheath flow of 3.0 L min$^{-1}$ were used. The yield of SOA was obtained by the ratio of the maximum mass concentration of the corrected particles to the mass concentration of myrcene consumed (Liu et al., 2024; Chen et al., 2022). The specific equation was as follows.

$$Y_{SOA} = \frac{M_{SOA}}{\Delta M_{VOC}}$$

Here, $Y_{SOA}$ represented the SOA yield, $M_{SOA}$ denoted the maximum mass concentration of particle after wall-loss correction during the reaction process, and $\Delta M_{VOC}$ referred to the total consumption mass concentration of VOCs throughout the reaction. The average effective density of SOA obtained from the myrcene ozonolysis is 1.25 g cm$^{-3}$ (Boge et al., 2013). The SOA was sampled after 3 hours of reaction, with a sampling duration of approximately 60 minutes. The SOA particles generated within the chamber were captured on a 25 mm PTFE filter (Sartorius, 0.45 μm pore size) and subsequently analyzed using an ultra-high performance liquid chromatography with a Quadrupole-Orbitrap mass spectrometer equipped with an electrospray ionization source (UHPLC/ESI-MS, UPLC, UltiMate 3000, Thermo Scientific, ESI-MS, Q-Exactive, Thermo Scientific) equipped with a Hypersil GOLD C18 column (2.1 × 100 mm, 1.9 μm packing size). The collected particle sample was eluted with 0.5 ml of methanol (Optima™ LC/MS Grade, Fisher Chemical) into a sample bottle. Mass spectrometric analysis utilized positive ion mode, scanning a molecular weight range of m/z 50–750 Da. The elution flow rate was set at 0.2 mL/min with a total run time of 4.0 min. An injection volume of 10 μL was used. In positive ion mode, three ionic forms of particulate components were identified, specifically [M+H]$^+$, [M+Na]$^+$ and [M+NH$_4$]$^+$. Tandem mass spectrometry (MS/MS) was employed to elucidate component structures within the SOA. The isolation width of 1.2 m/z units was applied. The electrospray ionization source was operated at a spray voltage of 3.0 kV and a capillary temperature of 300 °C, with sheath and auxiliary gas flows set to 35 and 10 Arb units, respectively. Both scans were performed at a resolution setting defined at m/z 200, with values of R = 70,000 for the MS scan and R = 17,500 for the MS/MS scan. Data acquisition and processing were conducted using Xcalibur software (version 3.0).

Corresponding revisions to the quantum chemical calculation methods and the Cartesian coordinates of the obtained molecular structures have been made in both the main text and the SI.

**Lines 152-161**, Additional quantum chemical calculations were performed to compare with the experimentally obtained infrared spectra. The geometries of the myrcene, POZs and SCIs were optimized using the hybrid density functional theory B3LYP-D3(BJ) with the aug-cc-pVTZ basis set. Harmonic vibrational frequencies were calculated at B3LYP/6-311G++(d,2p) calculation level for the comparison with the experimental infrared peak. Various computational levels for CIs were compared, and the one with superior performance was selected accordingly. Please refer to Table S1 for the specific comparison. This method has been proved to be applicable to the relevant calculations of the SCIs system (Chen et al., 2025; Lin et al., 2018; Yu et al., 2025). The above-mentioned related calculations were all performed by using Gaussian 16 software package (Frisch et al., 2016). Molclus 1.1.2 in conjunction with the xtb software package was used to perform a systematic conformational search for myrcene, POZs and SCIs (Lu, 2023). The single-point energy was further calculated at the DLPNO-CCSD(T)/CBS level by using ORCA 5.0 software to obtain the Boltzmann distribution of each conformation more accurately (Neese, 2022).

In the SI, we have added content to validate the reliability of our selected quantum chemical calculation methods.

**Table S1 The characteristic infrared peaks of CIs were obtained using different computational methods.**

| CIs | Calculated band/($cm^{-1}$) | | | | Experimental band/ ($cm^{-1}$) [a] | Assignment |
|---|---|---|---|---|---|---|
| | B3LYP-D3(BJ)/ 6−311++G(d,p) | B3LYP-D3(BJ)/ aug-cc-pVTZ | **B3LYP/6-311G++(d,2p)** | B3LYP/6-311++G(2d,2p) | | |
| Anti-$CH_3CHOO$ | 961 | 968 | **961** | 971 | **884** | O-O stretch |
| Syn-$CH_3CHOO$ | 884 | 897 | **882** | 908.17 | **871.2** | O-O stretch |
| $(CH_3)_2COO$ | 912 | 925 | **909** | 928 | **887.4** | O-O stretch |
| $CH_2OO$ | 899 | 912 | **901** | 923 | **909** | O-O stretch |
| Syn-C7-CIs | 883 | 893 | **882** | 900 | \ | O-O stretch |
| Anti-C7-CIs | 912 | 923 | **910** | 928 | \ | O-O stretch |

Note: [a] Data from references (Chhantyal-Pun et al., 2020a).

As shown in Table R1, the table correspondingly list only the peak positions associated with this vibration because of the manuscript focused on the most intense O-O stretching vibration of CIs. Among the four computational methods, the B3LYP/6-311G++(d,2p) level demonstrated superior performance in predicting the O-O stretching vibrational frequencies of CIs. The results from this method were also in

close agreement with the computational values reported previously by Li et al (Lin et al., 2015; Su et al., 2013; Wang et al., 2016). Thus, we have recalculated the infrared spectra of all relevant configurations in our manuscript using the B3LYP/6-311G++(d,2p) method and incorporated the necessary revisions. It could be observed that the calculated peak positions for alkyl-substituted CIs were systematically overestimated compared to the available experimental values. For example, a significant discrepancy of 77 cm$^{-1}$ was noted for anti-CH$_3$CHOO. In contrast, the deviation was much smaller for syn-CH$_3$CHOO being only 10.8 cm$^{-1}$. This was consistent with the trend previously observed by Li et al (Lin et al., 2015). Therefore, the computational results provided strong support for the identification of syn-C7-CIs.

The Cartesian coordinates for relevant molecules have been provided in Table S4 of the Supporting Information (SI).

**More specific comments**

Note that in this reaction system ozonolysis initiates all the observed oxidation chemistry and based on your results in Table 2 the co-produced OH also makes a big, apparently dominating impact on SOA mass. Now, when you add water, you decrease the whole oxidation sequence – also the important OH that would be generated from the CI isomerization through the VHP decompositions. Thus, it is very unclear if the reduction in SOA occurs specifically through scavenging of the Criegee intermediate and preventing its oligomerization or because the added water reduces also the further sequence of reactions contributing to SOA in the system. Moreover, when zooming on the SOA yields in Table 2, they indicate that OH chemistry played a major role in forming the observed SOA (i.e., 91 vs 346 ug/cm$^{-3}$). How was the OH scavenging determined to be 99% completed? Were there any repeated experiments using the hexane scavenger? So, you say that increasing RH decreased the SOA yield. But there's no apparent change between 0.5% and 20% humidity, and the humidity only plays an apparent role at 50% RH. This leads to a question that how was the particle size measured, and subsequently the SOA mass determined?

**The author's answer:** Thank you for your comments. The addition of an OH radical scavenger is a common approach in studies of VOC ozonolysis mechanisms (Docherty and Ziemann, 2003; Gong et al., 2024; Deng et al., 2021). The addition of an OH scavenger also influences the contribution of SCIs to SOA formation. By affecting the production of RO$_2$ radicals, the scavenger thereby reduces the total amount of RO$_2$ available to react with SCIs. Consequently, the observed effects cannot be attributed solely to the removal of OH radicals. In our study, the addition of the OH scavenger was implemented as a comparative experiment specifically designed to validate the proposed mechanism. Furthermore, in

ozonolysis systems, OH radicals are predominantly generated via the unimolecular decay of CIs.

Regarding the reviewer's point that $H_2O$ addition affects only the OH produced via the VHP channel of SCIs, this can be intuitively explained using the currently established reaction rate constants. Under conditions of 0.5% to 50%, the concentration of water monomers ranges from $4.0\times10^{15}$ to $3.8\times10^{17}$ molecule $cm^{-3}$. And the concentration of the water dimer ranges from $3.3\times10^{10}$ to $3.0\times10^{14}$ molecule $cm^{-3}$ (Jr-Min Lin and Chao, 2017). The unimolecular decay rate of $(CH_3)_2COO$ is $(305 \pm 70)$ $s^{-1}$ (https://pubs.acs.org/doi/10.1021/acs.jpca.6b07810) (Chhantyal-Pun et al., 2017). Its rate constant with the water monomer is $<1.5 \times 10^{-16}$ $cm^3$ $molecule^{-1}$ $s^{-1}$ and with the water dimer is $< 1.3 \times 10^{-13}$ $cm^3$ $molecule^{-1}$ $s^{-1}$ (https://www.pnas.org/doi/full/10.1073/pnas.1513149112) (Huang et al., 2015). Therefore, based on the current kinetic data, even at 50% RH, the addition of water cannot significantly affect the contribution of SCIs to OH radical production.

The scavenging efficiency of OH radicals was evaluated based on the following reaction rate constants. The relevant reactions and their rate constants are presented in the table below.

**Table R1 The rate constants for the reactions of myrcene and n-hexane with the OH radical.**

| Reactions | $k/ (cm^3\ molecule^{-1}\ s^{-1})$ |
|---|---|
| MYR + OH | $2.30\times10^{-10}$ (https://acp.copernicus.org/articles/21/16067/2021/) (Tan et al., 2021a) |
| HA + OH | $5.20\times10^{-12}$ (https://acp.copernicus.org/articles/3/2233/2003/) (Atkinson, 2003) |

$$\frac{k_{HA + OH}\times[HA]}{k_{MYR + OH}\times[MYR]}=\sim50$$

Therefore, approximately 98% of the OH radicals were eliminated.

**Lines 113**, we changed "The addition of approximately 367 ppm of n-hexane resulted in the removal of approximately 99% of OH radical." to "The addition of approximately 367 ppm of n-hexane resulted in the removal of approximately 98% of OH radical.".

We have conducted repeated experiments for all the experiments. As shown in Table R2 below, Exp. 1 presented the data included in the manuscript, while Exp. 2 was an additional replicated experiment provided here for reference but not included in the main text. The yields from these two replicate experiments showed almost no deviation. Additionally, Figure R4 depicted the temporal evolution (over time) of the particle volume concentration for Exp. 2.

**Table R2 Comparison of two different HA addition experiments.**

| Exp. | [Myrcene]/ppb | $[O_3]$/ppb | RH/% | HA | $M_{SOA}(\mu g/cm^3)$ | $Y_{SOA}$ |
|---|---|---|---|---|---|---|
| 1 | 170 | ~ 200 | < 0.5 | ~ 367 ppm | 91 | 0.10 |
| 2 | 127 | ~200 | < 0.5 | ~ 367 ppm | 69 | 0.10 |

[Figure]

**Figure R4 Particle volume concentration over time for Exp. 2 (Table R2).**

The particle size and SOA volume concentration mentioned by the reviewer were measured using a Scanning Mobility Particle Sizer (SMPS), with the instrument model specified in the manuscript. The formula for calculating mass concentration has been added to the manuscript.

**Lines 126-129**, "Throughout each experiment, size distributions and volume concentrations of particles were continuously recorded using a scanning mobility particle sizer (SMPS), which consisted a differential mobility analyzer (DMA, TSI, Model 3081) and a condensation particle counter (CPC, TSI, Model 3776).".

**Lines 130-135**, we added "The yield of SOA was obtained by the ratio of the maximum mass concentration of the corrected particles to the mass concentration of myrcene consumed (Liu et al., 2024; Chen et al., 2022). The specific equation was as follows.

$$Y_{SOA} = \frac{M_{SOA}}{\Delta M_{VOC}}$$

Here, $Y_{SOA}$ represented the SOA yield, $M_{SOA}$ denoted the maximum mass concentration of particle after wall-loss correction during the reaction process, and $\Delta M_{VOC}$ referred to the total consumption mass concentration of VOCs throughout the reaction.".

The aerosol particles were not dried during the measurement process. Existing research has demonstrated that for aerosols formed from VOC oxidation, when the relative humidity (RH) was below

50%, their water content was insufficient to alter the aerosol yield. Virkkula et al. (1999) measured the hygroscopic growth factor of aerosol formed from α-pinene ozonolysis at 84% RH to be 1.07 ± 0.01 (https://agupubs.onlinelibrary.wiley.com/doi/10.1029/1998JD100017) (Virkkula et al., 1999). Anthony J. Prenni et al. determined the hygroscopic growth factor of toluene-derived aerosol at a relative humidity (RH) of 85 ± 1% to be 1.01 to 1.07 ± 0.02 (https://agupubs.onlinelibrary.wiley.com/doi/10.1029/2006JD007963) (Prenni et al., 2007). Our experiments were conducted at even lower RH conditions, where the influence of water on aerosol volume growth was less pronounced. Hence, drying was not applied during the aerosol measurements.

I suspect the particles were dried before sizing, correct? (=not documented here). I would expect the particles to collect considerably more water at 50% than at 20% RH, so maybe the "missing SOA" mass at higher humidity is simply evaporating water. Could this be the case? Again, hard to say with the missing documenta on. As a related result the data in Figure 6 are hardly conclusive as the results at 0.5% and 20% RH look very similar and the HA addition seems to modulate mainly the C13 product, which makes sense if it was affected by a $RO_2$ + $RO_2$—RO + RO +$O_2$ step. Also, contrary to the text, the $C_7H_{10}O_x$ products appear to be decreasing with OH scavenging as well. Is this the case when looking at the details?

**The author's answer:** Thank you for your comments. As detailed in the previous response, the particles were not dried in our experiments. Therefore, the decrease in particle yield was not attributable to the evaporation of water.

As depicted in Figure 6, a very regular variation was observed at RH of 0.5%, 20%, and 50%. Furthermore, a marked difference was observed between RH 0.5% and 20%. For instance, the abundances of both $C_{13}H_{22}O_{6-9}$ and $C_{16}H_{28}O_{8-11}$ differed by nearly 3%.

As indicated above, $RO_2$ chemistry could not explain the formation of compounds larger than C14. Since the primary fragment peaks in the MS/MS spectra of the C13 compound aligned with those of C16, we concluded that C13 originated from the oligomerization of CIs. Additional MS/MS evidence supporting this was provided in the manuscript.

We have added Figure S4 to the SI.

[Figure]

**Figure S4 The MS/MS of $C_7H_{10}O_2$ + n-C3-SCIs + $RO_2$ sequence (a) and chemical structures of ions corresponding to major fragment peaks in MS/MS spectra (b).**

**Lines 302-312**, we added "Figure S4 presented the MS/MS spectrum of Sequence 2 along with the proposed structures corresponding to the major fragment peaks. The presence of fragments such as $C_5H_7^+$ (m/z 67.06), $C_7H_7O^+$ (m/z 95.05), and $C_6H_9O_2^+$ (m/z 113.06) indicated that the oligomers in Sequence 2 possessed a conjugated double-bond skeleton. This suggested that this sequence was initiated by C7 compounds. The fragment at m/z 71.05 ($C_4H_7O^+$) was generated via α-cleavage adjacent to a carbonyl group in the oligomer backbone. This fragmentation was typical of compounds containing conjugated carbonyl motifs and further supported the presence of a C7-based conjugated skeleton in the sequence. The $C_3H_7O^+$ (m/z 59.05) fragment ions originated from cleavage of terminal -OH functionalized tertiary carbon moieties in the oligomers. The formation of this ion fragment was similar to that in Sequence 1. In the MS/MS spectra of $C_{10}H_{16}O_4$, we observed that besides the fragment peaks annotated in Figure S4, there existed fragment ion peaks with higher intensities. This suggested that $C_{10}H_{16}O_4$ was formed through multiple pathways (such as the $RO_2$ + $RO_2$ reaction) (Peräkylä et al., 2023; Frandsen et al., 2025).".

From the figures it is unclear what species were detected and what are just assumed based on mechanistic principles. This is especially true for figure 5: Is the Figure 5a a measured spectrum or just a visualization of the identified peaks? Note that you will obtain the $C_{10}H_{15}O_x$ radical by hydrogen abstraction reaction too, and under such a high loading conditions $RO_2$ + $RO_2$ also surely occurs generating the odd oxygen product species through alkoxy radical isomerization reaction.

**The author's answer:** Thank you for your comments. Figure 5a was a visualization of the identified

peaks. These peaks were indeed from our actual measurements, and them to be valid. The peaks shown in Figure 5a correspond to the compounds generated by the mechanism illustrated in Figure 5c.

As for the OH radical generating the $C_{10}H_{15}O_x$ radical via H-abstraction, as noted by the reviewer, this mechanism was indeed valid. However, this pathway should be of minor importance in our system. After adding the OH scavenger, the peak intensity of $C_{10}H_{16}O_4$ in Figure 2(a) increased significantly, indicating that this peak did not primarily originate from reactions involving OH radicals. In studies of OH radical reactions with myrcene, the focus had predominantly been on addition mechanisms, while the importance of H-abstraction is likely to be minimal. Tan et al. only considered the OH-addition pathway in their study on the photo-oxidation of myrcene with OH radicals (https://doi.org/10.5194/acp-21-16067-2021) (Tan et al., 2021b). As the reviewer noted, $RO_2$ chemistry plays a critical role in aerosol formation, a point which we do not dispute. However, this is not the focus of our work.

**Comments about the methodology**

These are specialized techniques that must be explained carefully. The reader probably does not know that the signals are not exactly comparable across Ar matrix at 35K and particle-phase at room temperature. These are very different physical worlds, but now it sounds like it is just okay to equate chemical observations from the matrix to gas- and particle-phases. Why would you expect so? The minimum is that you explain to the reader why you think you can equate these worlds.

**The author's answer**:Thank you for your comments. Actually, matrix isolation and smog chamber experiments are not equivalent but rather complementary approaches that operate on different reaction timescales. Matrix isolation primarily targets the initial ozonolysis mechanisms of myrcene by enabling direct characterization of captured POZs and CIs. The reaction is unable to proceed further under the constraints imposed by the ultra-low temperature conditions. In contrast, smog chamber studies focus on the later stages of the reaction, revealing products contained CIs chain units that contribute to SOA formation and clarifying the specific mechanisms involved. The integration of these two methodologies provides a more coherent and complete picture of the overall reaction progression of CIs. The combination of these two methods aims to establish a connection between these gas-phase mechanisms and the formation of particle.

I am also a bit worried about the experimental conditions, but due to lack of documenta on it is hard to be

sure. So, CIs react with many of the present oxidation products with rapid rates, and if you really are observing CI related oligomerization, then it implies that you are using very high concentrations. Otherwise, it is very hard to see how you could see such reactive species oligomerizing in the gas-phase. However, from the Table 2 it seems that the highest primary oxidation rate is around 0.002 s$^{-1}$ (corresponding to k x [O$_3$]) which is rather low in comparison to atmospheric oxidation rates but still appears to rapidly result in the very high particle loads obtained without seed particles. Please explain what does not add up in these results.

**The author's answer:** Thank you for your comments. The VOCs concentrations in our experiments were reaching nearly 200 ppb, which was substantially higher than typical ambient levels. In the chamber experiment, myrcene was introduced first, followed by O$_3$ to initiate the reaction. The O$_3$ concentration (~ 200 ppb) listed in the table was the value recorded after the O$_3$ analyzer stabilized for 20 minutes, which corresponded to the 20-minute mark after the initiation of the experiment. This point has been addressed in the revised manuscript.

Furthermore, it was established in the literature that the ozonolysis of VOCs under seed-free conditions could also generate particles with a high number concentration.

Ditte Thomsen et al. reported that during the ozonolysis of α-pinene, the peak particle number concentration reached $1.18 \times 10^5$ cm$^{-3}$. The experiment was conducted with α-pinene at 126 ppb and ozone at 200 ppb. This number concentration was of the same order of magnitude as that obtained in our experiments (https://doi.org/10.1021/acs.est.2c04786) (Thomsen et al., 2022).

Yang et al. investigated the reaction of cyclooctene (195 ppb) with O$_3$ (839 ppb) at RH=25%, and observed that the particle number concentration could reach $5\times10^5$ cm$^{-3}$ within the initial 10 minutes of the reaction (https://doi.org/10.5194/acp-23-417-2023) (Yang et al., 2023).

Liu et al. used a flow tube to investigate the reactions of β-pinene (2–4 ppm) and limonene (1–3 ppm) with O$_3$ at concentrations of 50±10, 315±20, or 565±20 ppb, respectively. After a reaction time of 2.5 minutes in the flow tube, the maximum particle number concentrations reached ~$4\times10^6$ cm$^{-3}$ from β-pinene ozonolysis and ~$3.5\times10^6$ cm$^{-3}$ from limonene ozonolysis.

**Lines 250**, "Note: n-hexane is abbreviated as HA. The O$_3$ concentration listed in the table is the value recorded after the O$_3$ analyzer stabilized for 20 minutes, which corresponds to the 20-minute mark after the initiation of the experiment.".

Generally, it is very unrewarding to read "experiments were conducted under different conditions". From

the quite vague results given on the particles it seems that the experiment had very high oxidizing conditions, which seems surprising indeed in absence of seed particles. For example, it is said ". The dominant size range of SOA expanded from 50-250 nm during myrcene ozonolysis (Figure 3)." How can you reach so high particle loads without seed particles? What was the history of the used chamber setup? Could that have affected the results?

**The author's answer:** Thank you for your comments. **Lines 227-228**, we changed "the myrcene ozonolysis experiments were conducted under different conditions in a 1.2 m³ smog chamber." to "the myrcene ozonolysis experiments were conducted under different conditions (with or without an OH scavenger and at varying RH levels) in a 1.2 m³ smog chamber (Table 2).".

Numerous studies on VOC ozonolysis had been conducted in the absence of seed particles, and their reported particle size distribution ranged align with those observed in our work.

As shown in the Figure R5 below, Yang et al. investigated the particle size distribution from cyclooctene ozonolysis under different $SO_2$ concentrations without adding seed particles. While the study did not provide specific numerical values for the size distribution range, it could be seen from the Figure R5 that in the absence of $SO_2$, the particle size distribution exceeded 100 nm after 300 minutes of reaction. (https://doi.org/10.5194/acp-23-417-2023) (Yang et al., 2023).

[Figure]

**Figure R5 Size distributions of aerosol particles formed with various $SO_2$ concentrations at 10, 60, and 300 min after the initiation of cyclooctene ozonolysis. This figure comes from https://doi.org/10.5194/acp-23-417-2023**

The Figure R6 showed the particle size distributions reported by Liu et al. from the ozonolysis of

limonene and β-pinene in a flow tube study, where the residence time was only 2.5 min (Liu et al., 2023). Due to the use of higher VOC concentrations (2–4 ppm for β-pinene and 1–3 ppm for limonene), the particle size distribution obtained after 2.5 min in their system yielded a range comparable to that observed in our experiments after 30 min of reaction.

[Figure]

**Figure R6 Particle size and number concentration distributions of βpinene SOA (a) and limonene SOA (b) from SMPS measurement. This figure comes from https://doi.org/10.5194/acp-23-8383-2023.**

Collectively, the evidence presented above supported that a particle size distribution in the range of 50–250 nm was plausible for seed-free VOC ozonolysis.

The smog chamber used in this experiment had been only employed for ozonolysis studies prior to this work. Furthermore, before each experiment, it was thoroughly purged with clean zero air for at least 8 hours to ensure no carryover contamination could affect the results.

More about the chamber experiments:

• What is the timescale of the chamber experiments: It seems your growth rates are very high to obtain so many particles at so little time.

**The author's answer:** Thank you for your comments. As could be seen from Figure S2, our reaction

proceeded for 3 h. The phenomena we observed were in strong agreement with the existing literature, the relevant reports of which have been cited above.

• A representative figure of the experiment as a function of me showing the $O_3$, myrcene and some product me profiles would help to put the results in context.

**The author's answer:** Thank you for your comments. The objective of our chamber experiments was to investigate the key mechanisms of CIs in SOA formation by conducting offline analysis of myrcene ozonolysis SOA components using HPLC-ESI-MS. While such profiles could offer supplementary kinetic context, they were not essential for validating the core findings of our study, which centered on the offline molecular analysis of SOA composition to infer formation mechanisms. This approach was consistent with numerous prior chamber studies investigating aerosol formation mechanisms via detailed offline speciation (Yang et al., 2023; Thomsen et al., 2022; Liu et al., 2024), where the focus was predominantly on particle-phase composition rather than gas-phase temporal evolution. Therefore, we did not include these profiles in the manuscript.

We acknowledge that your suggestion was highlights an important aspect of chamber studies. We agree that systematic investigation of such time-resolved gas and particle-phase profiles will be valuable, and we plan to incorporate these measurements into our future, more kinetics-oriented work.

• How was the chamber experiments performed? Based on adding $O_2/O_3$ mixture with a syringe it sounds like you were doing batch-mode experiments, right? With the current documenta on it is unclear.

**The author's answer:** Thank you for your comments. The $O_2/O_3$ mixture was introduced in a single injection via a syringe and was not replenished during the experiment.

**Lines 116-117**, we added "The $O_2/O_3$ mixture was introduced in a single injection via a syringe and was not replenished during the experiment.".

• Spark generators do generally produce a lot of NOx. Did you measure how much NOx is in your reaction gas?

**The author's answer:** Thank you for your comments. Our $O_2/O_3$ mixture was generated from high-purity oxygen (≥99.999%) discharge, ensuring the absence of $N_2$ and consequently no NOx interference.

Furthermore, analysis of the particle phase in negative ion mode revealed that the major components consist solely of C, H, and O, with no N detectable. Below we presented the aerosol negative ion mode

spectrum (Figure R7), with the major molecular peaks correspondingly labeled.

[Figure]

**Figure R7 The SOA generated in Exp. 3 was analyzed by UHPLC/ESI-MS in negative ion mode.**

**Lines 114-115**, we added "An $O_2/O_3$ mixture was generated by passing high-purity $O_2$ (≥99.999%) at a flow rate of 200 mL/min through an ozone generator (Beijing Tonglin Technology Co., Ltd.).".

• The spectra given in Figure 4 are hard to compare. Especially the spectrum with hexane scavenger is very noisy and it is pretty much impossible to compare it to the peaks in the others. Also, it seems to contain more peaks than the other experiments, but the measured SOA yield is less. Does this mean that these spectra are not relevant to the SOA observations?

**The author's answer:** Thank you for your comments. Concerning the observation that the spectrum with added hexane (HA) showed more peaks but a lower SOA yield, we would like to clarify that the yield was not strongly correlated with the number of particle-phase components. The SOA yield depended on the total aerosol mass concentration, while the number of peaks reflected the diversity of compounds. After adding HA, although the number of major peaks increased, the overall abundance of these compounds decreased. The mass spectra presented in the manuscripts have been normalized, with the y-axis representing relative intensity. As shown in the unnormalized data (Figure R8), the absolute intensity of the dominant peaks decreased significantly (e.g., the highest peak dropped from $2.8 \times 10^7$ to $6.9 \times 10^6$). This indicates a reduction in the concentration of the major contributing components, which is consistent

with the lower overall SOA mass. The greater number of distinguishable peaks in the normalized spectrum is likely due to the suppression of dominant formation pathways, which makes a wider variety of minor products at relatively low concentrations more visible in the relative intensity plot.

[Figure]

**Figure R8 UHPLC/ESI-MS of SOA from myrcene ozonolysis in different conditions. (a) No scavenger was added. (b) 367 ppm n-hexane (HA) was added.**

• What is the cut-off size and the maximum size detected by the aerosol instrumentation?

**The author's answer:** Thank you for your comments. The measured particle size of the aerosol ranged from 14.3 to 723.4 nm.

  **Lines 129-130**, we added "The SMPS measured particles every 3 minutes across a size range of 14.3 to 723.4 nm. A sampling flow of 0.3 L min$^{-1}$ and a sheath flow of 3.0 L min$^{-1}$ were used.".

• How stabile was the $O_3$ syringe injection?

**The author's answer:** Thank you for your comments. We conducted a supplementary experiment to validate the stability of the $O_3$ syringe injection. In a 700 L chamber, an $O_2/O_3$ mixture of identical volume was injected four times sequentially. Following each injection, the $O_3$ concentration was monitored for 20 minutes. The resulting plot of $O_3$ concentration versus time was presented below. Each injection was designed to deliver approximately 100 ppb of $O_3$, with the observed variation in all cases remaining within

±20 ppb, as seen in the Figure R9.

[Figure]

**Figure R9 Time-dependent ozone concentration profile. Red text indicates the O₃ concentration measured after at least 20 min. Blue arrows denote the addition of the O₂/O₃ mixture.**

• What are the details of the LC-MS measurements? Please explain why you would get $Na^+$ and $NH_4^+$ clusters from a normal $H_3O^+$ source? Or what was the utilized ion source? What settings were used in the MS/MS analysis?

**The author's answer:** Thank you for your comments. $Na^+$ and $NH_4^+$ clusters are commonly observed ions in HPLC-ESI-MS analysis. Trace amounts of sodium ions ($Na^+$) and ammonium ions ($NH_4^+$) are virtually ubiquitous, originating from sources such as glassware and ambient air.

Zhao et al. observed $Na^+$ and $NH_4^+$ clusters when detecting SOA produced from α-pinene ozonolysis using UPLC-QToF-MS (Waters Acquity Xevo G2-XS) (https://doi.org/10.1021/acs.est.2c02090) (Zhao et al., 2022).

Wang et al., when measuring SOA from the ozonolysis of an isoprene and α-pinene mixture using UPLC-QToF-MS (Waters Acquity Xevo G2-XS), likewise detected $Na^+$ and $NH_4^+$ clusters (Wang et al., 2021).

Zhang et al. likewise observed that highly oxygenated molecules containing multiple peroxide groups are readily cationized through $Na^+$ adduction in their study of α-Pinene secondary organic aerosol using ESI-TOF (https://pubs.acs.org/doi/10.1021/acs.est.6b06588) (Zhang et al., 2017).

Why did you study the system by computations? Were they performed only to get the corresponding IR absorptions? If yes, then this should be explained clearly. Also, the accuracy of the predictions should be discussed. Currently the details of the computations are apparently missing.

**The author's answer:** Thank you for your comments. The primary purpose of conducting quantum chemical calculations is to compare with the experimentally obtained infrared spectra. Since no prior studies have reported the infrared characteristic peaks of POZs and CIs from myrcene ozonolysis, we relied on quantum chemical calculations to predict their vibrational signatures, which were then compared with our experimental spectra. A comparison was made among the widely used computational levels for predicting IR spectra of CIs, and the approach with minimal error was chosen for our study. The accuracy of the methodology and further details have been revised accordingly in the manuscript and the Supporting Information (SI).

**Lines 152**, "Additional quantum chemical calculations were performed to compare with the experimentally obtained infrared spectra."

**Lines 154-156**, we added "Harmonic vibrational frequencies were calculated at B3LYP/6-311G++(d,2p) calculation level for the comparison with the experimental infrared peak. Various computational levels for CIs were compared, and the one with superior performance was selected accordingly. Please refer to Table S1 for the specific comparison.".

In the SI, we have added the following contents.

**Table S1 The characteristic infrared peaks of CIs were obtained using different computational methods.**

| CIs | Calculated band/(cm$^{-1}$) | | | | Experimental band/ (cm$^{-1}$) [a] | Assignment |
|---|---|---|---|---|---|---|
| | B3LYP-D3(BJ)/ 6−311++G(d,p) | B3LYP-D3(BJ)/ aug-cc-pVTZ | **B3LYP/6-311G++(d,2p)** | B3LYP/ 6-311++G( 2d,2p) | | |
| Anti-CH$_3$CHOO | 961 | 968 | **961** | 971 | **884** | O-O stretch |
| Syn-CH$_3$CHOO | 884 | 897 | **882** | 908.17 | **871.2** | O-O stretch |
| (CH$_3$)$_2$COO | 912 | 925 | **909** | 928 | **887.4** | O-O stretch |
| CH$_2$OO | 899 | 912 | **901** | 923 | **909** | O-O stretch |
| Syn-C7-CIs | 883 | 893 | **882** | 900 | \ | O-O stretch |
| Anti-C7-CIs | 912 | 923 | **910** | 928 | \ | O-O stretch |

Note: [a] Data from references (Chhantyal-Pun et al., 2020a).

As shown in Table R1, the table correspondingly list only the peak positions associated with this vibration because of the manuscript focused solely on the most intense O-O stretching vibration of CIs. Among the four computational methods, the B3LYP/6-311G++(d,2p) level demonstrated superior

performance in predicting the O-O stretching vibrational frequencies of CIs. The results from this method were also in close agreement with the computational values reported previously by Li et al (Lin et al., 2015; Su et al., 2013; Wang et al., 2016). Thus, we have recalculated the infrared spectra of all relevant configurations in our manuscript using the B3LYP/6-311G++(d,2p) method and incorporated the necessary revisions. It could be observed that the calculated peak positions for alkyl-substituted CIs were systematically overestimated compared to the available experimental values. For example, a significant discrepancy of 77 cm$^{-1}$ was noted for anti-CH$_3$CHOO. In contrast, the deviation was much smaller for syn-CH$_3$CHOO being only 10.8 cm$^{-1}$. This was consistent with the trend previously observed by Li et al (Lin et al., 2015). Therefore, the computational results provided strong support for the identification of syn-C7-CIs.

What was the temperature in the twin-jet mixing stage. I'm trying to understand at what conditions the MT+O$_3$ reaction occurred as it is difficult to see how you could form the Criegee in the cold matrix through POZ isomerization.

**The author's answer:** Thank you for your comments. Matrix isolation coupled with Fourier transform infrared spectroscopy is currently a common technique for the measurement of POZs and CIs (Hoops and Ault, 2009; Li et al., 2019; Deng et al., 2012; Yang et al., 2020).

The mixing temperature was provided in the Methods section. **Lines 103-109**, "The deposition of myrcene/Ar and O$_3$/Ar onto the 6±1 K cold window was facilitated by two angled and independent tubes at a rate of 5 ml/min. This deposition was known as the twin-jet co-deposition mode. The deposition duration was approximately 120 min. To allow limit the diffusion and/or reaction of reactants, these matrices were heated or annealed to 35 K and held for 0.5 h, and then cooled to 6±1 K after which the spectra were recorded. To promote the further occurrence of the reaction and further soften and diffuse the matrix, the matrix was further heated to 45 and 55 K (Yu et al., 2025; Yang et al., 2020). To prevent matrix loss, it was imperative to immediately cool down to 6±1 K after reaching the target temperature and to record the spectra.".

In the solid argon matrix (6±1 K), reactant molecules are "frozen" and isolated in rigid lattice sites. POZs isomerization requires energy. The subsequent programmed temperature annealing steps (to 35 K, 45 K, and 55 K) are crucial. These steps provide the trapped molecules with sufficient kinetic energy to undergo very limited movement within the lattice, thereby overcoming the relatively low energy barrier to allow POZs formation and its subsequent unimolecular decomposition to generate CIs.

Some of the new peaks resemble more like noise, for example the 880 and 1074 cm$^{-1}$. How do you define a new peak exactly?

**The author's answer:** Thank you for your comments. The repeated experiments were carried out multiple times to eliminate the interference of noise. Infrared spectra obtained from two separate twin-jet experiments after annealing to 55 K were presented below. As shown in the Figure R10, the orange spectrum corresponds to the one used in the main text, while the green spectrum represents an additional replicate experiment. Both spectra exhibit peaks at 880 and 1074 cm$^{-1}$.

[Figure]

**Figure R10 Infrared spectra obtained from two separate twin-jet experiments after annealing to 55 K.**

You said that "(The peaks with relative abundance of less than 1% were ignored)." But the Spectra shown in Figure 4 contain 100s (or 1000s?) of peaks and you have only labelled 5 and given a handful of others in the text. Thus, how many of the peaks were neglected? You should ideally provide the peak list for signals (above some threshold) observed during the experiment.

**The author's answer:** Thank you for your comments. As shown in the Table R3, for each mass spectrum in Figure 4, we present the total number of peaks, the number of peaks with relative intensity >1%, and the ratio of the summed intensity of those >1% peaks to the total intensity of all peaks. The table indicates that the peaks above 1% relative intensity largely capture the composition of SOA. This observation suggests that omitting peaks below 1% relative abundance is an appropriate data treatment.

**Table R3 The total number of peaks, the number of peaks with relative intensity >1%, and the ratio of the summed intensity of those >1% peaks to the total intensity of all peaks in Figure 4.**

| Exp. | The total number | Number of peaks with RA >1% | Ratio of the summed RA of peaks >1% to the total RA |
|---|---|---|---|
| 1 | 3753 | 903 | 88 % |
| 2 | 4267 | 1969 | 95 % |
| 3 | 3799 | 1169 | 91% |
| 4 | 3799 | 1225 | 91% |

Regarding the limited labeling in Figure 4, Figure 4 labeled only a select few peaks for clarity of visual presentation. The primary purpose of this figure was to provide a qualitative overview showing the distinct spectral patterns between different experimental conditions (e.g., the emergence of new product families upon scavenger addition). The species we have annotated, including the C10H18Ox, C7H10Ox, and C10H16Ox families, have been identified as the important constituents within the SOA generated from myrcene ozonolysis. This approach was intended to intuitively convey to readers the product peaks corresponding to the key mechanisms investigated in our study. Moreover, this limited labeling approach was frequently adopted in the analysis of SOA sample composition. Examples of such limited labeling could be found in previously published mass spectra in the literature, as presented in the following section.

Figure R11 was derived from the compositional analysis mass spectra of SOA generated during the ozonolysis of *trans*-3-hexene. That study also focused on investigating the CIs oligomerization mechanism, and consequently, only the product peaks attributable to this specific mechanism were labeled.

[Figure]

**Figure R11 Normalized ESI mass spectra of SOA from ozonolysis of trans-3-hexene in the flow reactor in the absence and presence of an OH scavenger. This figure came from (Zhao et al., 2015).**

Figure 12 presented the compositional analysis mass spectra, with limited labeling, of SOA formed from the OH-oxidation of a myrcene and D-limonene mixed system.

[Figure]

**Figure R12 High-resolution mass spectra (100−700 Da) of SOA produced from mixtures of myrcene and D-limonene. This figure came from**

In this study, we specifically targeted the product peaks corresponding to the CI oligomerization pathway. Our focus did not encompass all SOA formation pathways. Therefore, a complete peak list of all signals observed (above a certain threshold) during the experiment was not provided.

Clearly more details are needed to understand the experiments done, and thereby also the proposed chemistry, which is discussed next.

**Comments about the proposed mechanisms**

The manuscript makes several claims about the potential mechanism of oligomer forma on by Criegee intermediate reactions and their relevance for SOA formation from myrcene.

First of all, I would like to take the me to explain that in chemistry the word "mechanism" has a very strong meaning and is reserved to explain how the molecules transform. What is meant here by the same word in several instances is hardly a mechanism. Thus, mentions like "The mechanisms may also exist in other monoterpenes ozonolysis, which offering new insights into the contribution of CIs to SOA formation." do not make much sense without detailing the molecular steps.

**The author's answer:** Thank you for your comments. In the revised manuscript, we have rephrased the description of the synergistic mechanism between CIs oligomerization and $RO_2$ autoxidation as the

**synergistic effect**. **Lines 18 and 388**, we have revised the term "synergistic mechanism" to "synergistic effect" in the manuscript. **Lines 342**, we changed "proposed mechanism" to "proposed pathway".

Related: "The coexistence of these CIs of different molecular sizes led to a distinctly different ozonolysis mechanism for myrcene compared to that of cyclic monoterpenes (e.g., α-pinene, limonene)". No. It is the same mechanism for both but with acyclic species the bond breaking leads to two species, whereas with cyclic species only one product is generated. And further: I can't seem to make sense of the following statement: "Our findings further demonstrated that when evaluating SCIs contribution pathways to SOA, the molecular size of SCIs must be prioritized especially during monoterpenes ozonolysis."

**The author's answer:** Thank you for your comments. We have made the corresponding revisions in the manuscript. **Lines 391-393**, we changed "The coexistence of these CIs of different molecular sizes led to a distinctly different ozonolysis mechanism for myrcene compared to that of cyclic monoterpenes (e.g., α-pinene, limonene)" to "The coexistence of carbon atoms with varying molecular sizes leads to a different role of CIs in secondary organic aerosol SOA formation during the ozonolysis of myrcene, compared to that observed in cyclic monoterpenes (e.g., α-pinene and limonene).".

Our conclusion was based on the specific experimental observation from this study. During myrcene ozonolysis, which produced both larger (C7-CIs) and smaller (C3-CIs) CIs, our analysis detected oligomeric products attributable to reactions by the smaller C3-SCIs. The larger C7-SCIs appeared to react predominantly via unimolecular decomposition, yielding monomeric $C_7H_{10}O_x$ compounds into the particle phase. Furthermore, a review of the literature on monoterpene ozonolysis (e.g., α-pinene, limonene) revealed that the proposed SCIs reactions typically did not involve an CIs oligomerization pathway for the standard C10-SCIs. Therefore, our findings suggested that the oligomerization mechanism might be a unique contribution pathway specifically accessible to systems that could generate smaller CIs (like C3-CIs from myrcene). Hence, when evaluating the CIs contribution pathways to SOA, especially to identify if oligomerization was possible, the molecular sizes became a factor to consider. We have refined the relevant sentence in the manuscript to express this more precisely.

**Lines 373-375**, we changed "Our findings further demonstrated that when evaluating SCIs contribution pathways to SOA, the molecular size of SCIs must be prioritized especially during monoterpenes ozonolysis." to "Our findings further demonstrated that when evaluating SCIs contribution pathways to SOA, the molecular size of SCIs might be prioritized especially during monoterpenes

ozonolysis. The molecular size of SCIs might lead to differences in their primary mechanisms of contributing to SOA formation.".

And further: You say that "To our knowledge, this was the first me this synergistic mechanism has been proposed." – this seems like an awkard statement as there is no actual mechanism presented.

**The author's answer**:Thank you for your comments. We have made the corresponding revisions in the manuscript. **Lines 388**, "To our knowledge, this was the first me this synergistic effect has been proposed.".

And further: "MI-FTIR experiments unequivocally verified that myrcene ozonolysis proceeded via the Criegee mechanism." – this is confusing as it is completely unclear what would be the "other mechanism" the Authors are referring to? Ozonolysis is commonly expected to proceed through the Criegee mechanism.

**The author's answer**:Thank you for your comments. You are correct in noting that ozonolysis is generally expected to proceed via the Criegee mechanism. Our purpose in stating this was not to imply an alternative pthway, but rather to provide direct, spectroscopic confirmation for this specific system. By using MI-FTIR to detect key transient intermediates (POZs and CIs) unique to the Criegee pathway, we offer definitive experimental evidence that the ozonolysis of myrcene follows this established mechanism.

The detected compounds are C7, C10, C13 and C16 species, which all appear to have also alternative production paths, especially through $RO_2$ chemistry. What is noteworthy is that in the current chamber experiments very high growth rates are obtained even in apparent absence of seed particles which testifies the very high oxidation conditions used in the experiments. Under such conditions many sorts of radical recombination can occur – potentially even the $sCI + RO_2$, which I still find much more unlikely than the $RO_2 + R'O_2$ processes.

**The author's answer**:Thank you for your comments. Following the addition of an OH radical scavenger, the formation of $C_7H_{10}O_x$ can be attributed exclusively to the unimolecular reaction of C7-CIs according to currently established mechanisms, with no alternative gas-phase pathway available to explain it. To elucidate the formation mechanisms of C10, C13, and C16, we present additional evidence from their MS/MS spectra.

Figure S4 presented the MS/MS spectra of $C_{10}H_{16}O_4$, $C_{13}H_{22}O_6$, and $C_{16}H_{28}O_8$. The major fragment

peaks of $C_{13}H_{22}O_6$ and $C_{16}H_{28}O_8$ were highly similar. The presence of fragments such as $C_5H_7^+$ (m/z 67.06), $C_7H_7O^+$ (m/z 95.05), and $C_6H_9O_2^+$ (m/z 113.06) indicated that the oligomers in Sequence 2 possessed a conjugated double-bond skeleton. This suggested that this sequence was initiated by C7 compounds. The fragment at m/z 71.05 ($C_4H_7O^+$) was generated via α-cleavage adjacent to a carbonyl group in the oligomer backbone. This fragmentation was typical of compounds containing conjugated carbonyl motifs and further supported the presence of a C7-based conjugated skeleton in the sequence. The $C_3H_7O^+$ (m/z 59.05) fragment ions originated from cleavage of terminal -OH functionalized tertiary carbon moieties in the oligomers. The formation of this ion fragment was similar to that in Sequence 1. In the MS/MS spectra of $C_{10}H_{16}O_4$, we observed that besides the fragment peaks annotated in Figure S4, there existed fragment ion peaks with higher intensities. This suggested that $C_{10}H_{16}O_4$ was formed through multiple pathways (such as the $RO_2 + RO_2$ reaction mentioned by the reviewer).

[Figure]

**Figure S4 The MS/MS of $C_7H_{10}O_2$ + n-C3-SCIs + $RO_2$ sequence (a) and chemical structures of ions corresponding to major fragment peaks in MS/MS spectra (b).**

We have added Figure S4 to the SI.

**Lines 302-312**, we added "Figure S4 presented the MS/MS spectrum of Sequence 2 along with the proposed structures corresponding to the major fragment peaks. The presence of fragments such as $C_5H_7^+$ (m/z 67.06), $C_7H_7O^+$ (m/z 95.05), and $C_6H_9O_2^+$ (m/z 113.06) indicated that the oligomers in Sequence 2 possessed a conjugated double-bond skeleton. This suggested that this sequence was initiated by C7 compounds. The fragment at m/z 71.05 ($C_4H_7O^+$) was generated via α-cleavage adjacent to a carbonyl group in the oligomer backbone. This fragmentation was typical of compounds containing conjugated carbonyl motifs and further supported the presence of a C7-based conjugated skeleton in the sequence.

The $C_3H_7O^+$ (m/z 59.05) fragment ions originated from cleavage of terminal -OH functionalized tertiary carbon moieties in the oligomers. The formation of this ion fragment was similar to that in Sequence 1. In the MS/MS spectra of $C_{10}H_{16}O_4$, we observed that besides the fragment peaks annotated in Figure S4, there existed fragment ion peaks with higher intensities. This suggested that $C_{10}H_{16}O_4$ was formed through multiple pathways (such as the $RO_2 + RO_2$ reaction).".

Indeed, the experiment was conducted under a highly oxidized state. Comparing this with the current literature, I see no issues with our experimental conditions. Many current studies on ozonolysis reactions employ relatively high oxidant concentrations. The Table R4 compiles the precursor concentrations used in ozonolysis experiments as reported in the current literature.

**Table R4 The precursor concentrations used in current smog chamber studies investigating VOC ozonolysis reactions.**

| No. | precursor concentrations | Reference |
|-----|--------------------------|-----------|
| 1 | α-pinene 67.7-323 ppb, $O_3$ 640-698 ppb (298 K) | https://doi.org/10.5194/acp-21-5983-2021 |
| 2 | 2,3- dihydrofurans 0.2-3 ppm, $O_3$ 0.2-4 ppm | https://doi.org/10.5194/acp-17-2347-2017 |
| 3 | $\Delta^3$-carene ~1100 ppb, $O_3$ ~900 ppb | https://doi.org/10.5194/acp-24-9459-2024 |
| 4 | Limonene 321±39 ppb, $O_3$ 5.5-6.0 ppm | https://doi.org/10.5194/acp-23-10809-2023 |
|   | $\Delta^3$-carene 341±28 ppb, $O_3$ 6.0-6.4 ppm | |
| 5 | Trans-β-methylstyrene ~926 ppb, $O_3$ ~ 1 ppm | https://doi.org/10.1021/acs.est.5c00769 |
|   | Styrene 1047 ppb, O3 ~ 1 ppm | |

Given that $CH_3OO + CH_3OO$ reacts more slowly than $CH_2OO + CH_3OO$ (as shown in the kinetic data above) ((https://doi.org/10.1021/acs.jpclett.4c00159) (Chao et al., 2024), the occurrence of SCI-$RO_2$ reactions is kinetically plausible.

"Current studies have not confirmed that the C10-CIs generated from monoterpene ozonolysis can contribute to SOA forma on through oligomerization." This makes sense in considering the reactivities of the CIs and sCIs discussed in the above comments and again appears to point out that it is more likely you are observing $RO_2$ chemistry. Perhaps this is possible in high concentrations in a laboratory setting, but even then, it is not so easy to make sCIs oligomerize due to abundance of other potential sCI reaction partners generated during oxidation (e.g., any species containing carbonyl functionality).

**The author's answer:** Thank you for your comments. As noted in the preceding response, oligomers containing CIs structural units have indeed been detected in the atmosphere. Therefore, the possibility of CIs oligomerization is beyond doubt. The specific evidence has been presented in the preceding response.

We agreed that $RO_2$ chemistry was a dominant and crucial mechanism for aerosol formation. Our

experimental design and discussion intentionally focused on products attributable to CIs-driven reactions, particularly oligomerization, to evaluate its distinct contribution.

Your point about the competition from abundant carbonyl species was well-taken and was a key factor in evaluating the likelihood of SCIs oligomerization. Indeed, established kinetic data showed that the reaction of the simplest SCIs, $CH_2OO$, with carbonyls like acetaldehyde ($\sim 9.4 \times 10^{-13}$ $cm^3$ $molecule^{-1}$ $s^{-1}$) or acetone ($\sim 2.3 \times 10^{-13}$ $cm^3$ $molecule^{-1}$ $s^{-1}$) was relatively slow (https://pubs.rsc.org/en/content/articlelanding/2012/cp/c2cp40294g) (Taatjes et al., 2012). Crucially, these rates were approximately two orders of magnitude slower than the reaction between $CH_2OO$ and the $CH_3OO$ (($1.7 \pm 0.5) \times 10^{-11}$ $cm^3$ $s^{-1}$ (https://doi.org/10.1021/acs.jpclett.4c00159) or ($2.4 \pm 1.2) \times 10^{-11}$ $cm^3$ $molecule^{-1}$ $s^{-1}$ (https://dx.doi.org/10.1021/acsearthspacechem.0c00147)) (Chao et al., 2024; Chhantyal-Pun et al., 2020b). Therefore, the significant kinetic preference of CIs for reaction with $RO_2$ radicals over carbonyl compounds provides a mechanistic basis for this process to compete effectively, even in a complex mixture of oxidation products.

**Comments**

I would strongly recommend language editing by a native speaker as the text contains several apparent ambiguities that hinder understanding the work. Some examples below:

When you talk about unimolecular degradation of sCI contributing to SOA it seems odd. Note that "degradation" seems to imply the molecule breaking into small pieces whereas you probably just mean the OH loss through a VHP that kickstarts the autoxidation process. I can only assume you mean this as nothing like that has been explained in the article (e.g., this is omitted from Figure 1, for example).

**The author's answer:** Thank you for your comments. **Lines 16, 38, 39, 40, 174, 263, 267, 268 and 389**, we changed "degradation" to "decomposition". "Decomposition" was widely used to describe the unimolecular reactions of CIs (https://pubs.rsc.org/en/content/articlelanding/2018/em/c7em00585g, https://pubs.acs.org/doi/10.1021/acs.jpca.5b12124) (Khan et al., 2018; Smith et al., 2016).

Figure captions should be expanded to explain clearly what is shown in the figures. Some examples: It took me a while to realize that the myrcene + $O_3$ spectrum is included in Figure 2 as it is currently poorly labelled and not mentioned in the caption. Figure 1 is messy and hard to follow and will require a long explanation of the steps shown (e.g., the all-important VHP decomposition is not marked).

**The author's answer:** Thank you for your comments. **Lines 220**, we modified the caption of Figure 2,

"The twin-jet IR spectra of myrcene ozonolysis reaction in a low temperature and Ar matrix after annealing to 35 K, 45 K and 55 K. The blank spectra of myrcene/Ar and $O_3$/Ar were also given.". in the Figure 2 caption.

We have also revised Figure 1. The VHP channel was not the sole pathway for unimolecular degradation of CIs. While syn-CIs readily underwent unimolecular reactions via the VHP route, anti-CIs primarily proceeded through an initial rearrangement (1,3 ring-closure) to form a dioxirane intermediate. Therefore, in the Figure 1, no distinction was made, and both were uniformly labeled as "Uni." Additionally, an explanatory note had been added to the Figure 1 caption.

**Lines 178**,

**Figure 1 Proposed the key pathway in the initial ozonolysis of myrcene. The values in parentheses represent the yields of the corresponding products. The blue boxes indicate OH-derived products. Green-shaded boxes represent products formed via unimolecular reaction of *syn*-CIs through the vinylhydroperoxide (VHP) channel followed by $O_2$ addition. Gray-shaded boxes correspond to products generated from unimolecular reaction of *anti*-CIs via 1,3-ring-closure.**

I can't understand the following: "The OH radical yield from myrcene ozonolysis was generally high, which also confirmed that the larger CIs generated during this process tend to react via unimolecular decay pathways (Cox et al., 2020)". Why "generally"? Why "confirm" here?

**The author's answer:** Thank you for your comments. As presented in Table R5, which summarizes the OH radical yields from the ozonolysis of various VOCs, it is evident that the yield from myrcene

ozonolysis is higher than those from most small alkenes, including isoprene. To prevent potential misinterpretation, we have revised the corresponding description in the text. **Lines 271**, we deleted "generally".

**Table R5 Summary of recommended HO yields for reactions of $O_3$ with alkenes at 298 K and 1 bar. This table came from https://doi.org/10.5194/acp-2020-472.**

| Reaction ID [a] | alkene | HO yield | comments |
|---|---|---|---|
| **Small alkene reactions** | | | |
| Ox_VOC5 | ethene | $0.17 \pm 0.05$ | (b) |
| Ox_VOC6 | propene | $0.36 \pm 0.04$ | (c) |
| Ox_VOC16 | but-1-ene | $0.38 \pm 0.18$ | (d) |
| Ox_VOC17 | *cis*-but-2-ene | $0.33 \pm 0.07$ | (e) |
| Ox_VOC18 | *trans*-but-2-ene | $0.60 \pm 0.06$ | (f) |
| Ox_VOC15 | 2-methylpropene | $0.69 \pm 0.15$ | (g) |
| Ox_VOC41 | 2,3-dimethylbut-2-ene | $0.93 \pm 0.14$ | (h) |
| Ox_VOC7 | isoprene | $0.26 \pm 0.04$ | (i) |
| **Monoterpene reactions** | | | |
| Ox_VOC8 | α-pinene | $0.80 \pm 0.10$ | (j) |
| Ox_VOC19 | β-pinene | $0.30 \pm 0.06$ | (k) |
| Ox_VOC20 | limonene | $0.66 \pm 0.04$ | (l) |
| Ox_VOC21 | camphene | $\leq 0.18$ | (m) |
| Ox_VOC22 | 2-carene | $0.81 \pm 0.11$ | (n) |
| Ox_VOC23 | 3-carene | $0.86 \pm 0.11$ | (n) |
| Ox_VOC24 | β-myrcene | $0.63 \pm 0.09$ | (n) |
| Ox_VOC25 | β-ocimene | $0.55 \pm 0.09$ | (n) |
| Ox_VOC26 | α-phellandrene | $0.29 \pm 0.05$ | (o) |
| Ox_VOC27 | β-phellandrene | $0.14^{+0.07}_{-0.05}$ | (m) |
| Ox_VOC28 | sabinene | $0.33 \pm 0.05$ | (n) |
| Ox_VOC29 | α-terpinene | $0.32 \pm 0.06$ | (p) |
| Ox_VOC30 | γ-terpinene | $0.81 \pm 0.11$ | (n) |
| Ox_VOC31 | terpinolene | $0.70 \pm 0.08$ | (q) |
| **Sesquiterpene reactions** | | | |
| Ox_VOC32 | β-caryophyllene | $0.08 \pm 0.03$ | (r) |
| Ox_VOC33 | α-cedrene | $0.65 \pm 0.05$ | (s) |
| Ox_VOC34 | α-copaene | $0.35^{+0.18}_{-0.12}$ | (t) |
| Ox_VOC37 | α-humulene | $0.16 \pm 0.06$ | (u) |

During the myrcene ozonolysis, OH radicals were predominantly generated from the unimolecular degradation of CIs. Combined with mass spectrometry analysis, the detection of oligomers featuring C3-CIs chain units demonstrated that a substantial portion of C3-CIs was removed via bimolecular reaction pathways, thereby unable to fully contribute to OH radical formation. In the mass spectra, C7-$RO_2$ radicals derived from the unimolecular degradation of C7-CIs are observed, whereas no distinct product peaks corresponding to bimolecular reactions of C7-CIs were detected. Thus, it was reasonable to propose that the larger CIs (C7-CIs) predominantly underwent reaction via unimolecular degradation pathways.

The following seems to contradict itself: "The initial ozonolysis mechanism of myrcene had been

established as shown in Figure 1 based on the current studies."

**The author's answer:** Thank you for your comments. There is no contradiction. The mechanism presented in Figure 1 was previously inferred based on established research conclusions, as direct measurements of CIs and POZs were not available. Our application of the MI-FTIR method, which enabled the direct detection of key intermediates (POZs and CIs), provided the most direct evidence that the initial ozonolysis of myrcene followed the Criegee mechanism.

"As shown in Figure 4(b), the peaks corresponding to $C_{10}H_{18}O_4$ and $C_{10}H_{18}O_5$ disappeared after the addition of the OH radical scavenger, as expected. The $C_7H_{10}O_2$ peak remained, which further demonstrated that $C_7H_{10}O_2$ originated from SCIs-derived products. Compared with the mass spectrum without the scavenger, the contribution of the oligomers in Sequence 2 markedly increased." With this low-resolution figure, it is hard to say what peaks decreased and what increased. To me it looks like the sequence 2 peaks actually decreased. Also, the next claim "Correspondingly, the contribution of Sequence 2 to SOA forma on decreased progressively with increasing RH as shown in Figure 4(c) and (d)." I really can't read from the current figure.

**The author's answer:** Thank you for your comments regarding the clarity of Figure 4 and the interpretation of the spectral changes. We appreciate your careful examination. You raised a valid point about the current rendering of the figures potentially affecting the ease of visual comparison. While the embedded images were prepared according to the journal's formatting guidelines, which can sometimes limit the display resolution, we assure you that all underlying mass spectrometry data were acquired at high resolution. The type and model of the mass spectrometer employed for offline SOA analysis are provided in the manuscript. Additional clarification regarding the changes in product peak intensities has been provided in the manuscript.

**Lines 137-141**, "The SOA particles generated within the chamber were captured on a 25 mm PTFE filter (Sartorius, 0.45 μm pore size) and subsequently analyzed using an ultra-high performance liquid chromatography with a Quadrupole-Orbitrap mass spectrometer equipped with an electrospray ionization source (UHPLC/ESI-MS, UPLC, UltiMate 3000, Thermo Scientific, ESI-MS, Q-Exactive, Thermo Scientific) equipped with a Hypersil GOLD C18 column (2.1 × 100 mm, 1.9 μm packing size).

**Lines 148-149**, we added "Both scans were performed at a resolution setting defined at m/z 200, with values of R = 70,000 for the MS scan and R = 17,500 for the MS/MS scan.".

**Lines 315-324**, As shown in Figure 4(b), the peaks corresponding to $C_{10}H_{18}O_4$ and $C_{10}H_{18}O_5$

disappeared after the addition of the OH radical scavenger, as expected. The $C_7H_{10}O_2$ peak remained, which further demonstrated that $C_7H_{10}O_2$ originated from SCIs-derived products. Compared with the mass spectrum without the scavenger, the contribution of the oligomers in Sequence 2 markedly increased. Following the introduction of the scavenger, the peak intensities corresponding to compounds with higher degrees of oligomerization ($C_{13}H_{22}O_6$ and $C_{16}H_{28}O_8$) within the sequence were significantly enhanced. Moreover, compared to the spectrum acquired in the absence of the scavenger, a more highly oligomerized compound ($C_{19}H_{34}O_{10}$) was detected. Water served as the dominant removal pathway for SCIs in the atmosphere. Correspondingly, the contribution of Sequence 2 to SOA formation decreased progressively with increasing RH as shown in Figure 4(c) and (d), especially substances with a higher degree of oligomerization in the sequence ($C_{13}H_{22}O_6$ and $C_{16}H_{28}O_8$). So, the Sequence 2 was primarily contributed by SCIs-derived.

As this is ACP the page content is not limited. Thus, you should put the SI material directly into the main text to improve readability.

**More minor comments:**

• Please refrain from using "extreme" when it is not needed. There is definitely no "extremely high abundance of water vapor" and Criegee does not react at extremely fast rates.

**The author's answer:** Thank you for your comments. **Lines 28 and 32**, we deleted "extreme".

• What is a "a heated three-way U-shaped tube."?

**The author's answer:** Thank you for your comments. A schematic diagram of the heated three-way U-shaped tube is presented below. The two ends of the U-shaped tube are connected to zero air and the smog chamber, respectively, while the VOC is injected through the middle using a micro-syringe. When VOCs are added into the U-shaped tube and zero air flows through it, heating the bottom of the U-shaped tube causes the zero air to carry the VOC vapor into the smog chamber.

[Figure]

**Figure R13 Schematic diagram of the heated three-way U-shaped tube.**

• There is no Table 1.

**The author's answer:** Thank you for your comments. **Lines 221**, we changed "Table 2" to "Table 1".

• "Zhang et al. found that limonene yield gradually increased with increasing RH" – I don't think you mean limonene concentrations increasing.

**The author's answer:** Thank you for your comments. **Lines 50-51**, "Zhang et al. found that SOA yield form limonene ozonolysis gradually increased with increasing RH.".

• Please explain in the text why you added formic acid to the mixture?

**The author's answer:** Thank you for your comments. **Lines 112**, we changed "The addition of formic acid and n-hexane was conducted in a consistent manner." to "The n-hexane was conducted in a consistent manner.".

• Mark the peaks with the corresponding assignments in Figure S1.

**The author's answer:** Thank you for your comments. We have revised Figure S1 as shown below.

[Figure]

**Figure S1 The IR spectra of precursor (myrcene/Ar and O$_3$/Ar) in a low temperature after annealing to 35 K, 45 K and 55 K.**

• What is the difference in sequence 2 and sequence 2N?

**The author's answer:** Thank you for your comments. Sequence 2N contained Sequence 2. The formation of Sequence 2N was driven by the oligomerization of SCIs and the autoxidation of RO$_2$. Specifically, Sequence 2 included C$_7$H$_{10}$O$_2$, C$_{10}$H$_{16}$O$_4$, C$_{13}$H$_{22}$O$_6$, and C$_{16}$H$_{28}$O$_8$, while Sequence 2N included C$_7$H$_{10}$O$_{2-5}$, C$_{10}$H$_{16}$O$_{4-7}$, C$_{13}$H$_{22}$O$_{6-9}$, and C$_{16}$H$_{28}$O$_{8-11}$.

• Note that it is generally not possible to label the products to -OOH and -OH species simply by their measured composition. It is plausible these could be the products but with the current techniques and experimental conditions it appears that you have no way to be sure about them. Reword accordingly.

**The author's answer:** Thank you for your comments. **Lines 263**, we deleted "C10-RO$_2$". **Lines 265**, we deleted "C10-R'O$_2$" and "(C10-R'OOH)". **Lines 267**, we deleted "C10-R'OH". **Lines 269**, we deleted "C7-RO$_2$". **Lines 269**, we deleted "C7-OH". **Lines 293**, we changed "C$_{10}$H$_{17}$O$_5$ + C7-RO$_2$ (C$_7$H$_{11}$O$_4$, the formation pathway as shown in Scheme S1)" to "C$_{10}$H$_{17}$O$_5$ + C$_7$H$_{11}$O$_4$ (the formation pathway as shown in Scheme S1)"

• The following is misleading "Quantum chemical calculations have revealed that multiple SCIs may undergo oligomerization reactions with water vapor to form oligomers with lower volatility (Chen et al., 2019)." – this is not what the cited article says or what you mean here.

**The author's answer:** Thank you for your comments. This reference was "Oligomer formation from the gas-phase reactions of Criegee intermediates with hydroperoxide esters: mechanism and kinetics". The hydroperoxide esters was the product from the reaction of $CH_2OO$ with water. To prevent any further misunderstandings, we made some changes.

**Lines 55-56**, "Quantum chemical calculations have revealed that multiple SCIs may undergo oligomerization reactions with hydroperoxide esters (the products of the reaction of $CH_2OO$ with water) to form oligomers with lower volatility (Chen et al., 2019).".

• This is ambiguous "Both in the α-pinene and limonene ozonolysis, SCIs-derived products contribute to both monomers and dimers forma on of SOA" – of course the reaction initiating the whole oxidation systems contributes. Please be clearer what you mean.

**The author's answer:** Thank you for your comments. **Lines 65-66**, we changed "Both in the α-pinene and limonene ozonolysis, SCIs-derived products contribute to both monomers and dimers forma on of SOA" to "specific mechanisms have been proposed in which both the unimolecular channels and the bimolecular reactions of SCIs contribute to the formation of SOA.".

• Open all abbreviations. For example, POZ does not appear to be explained.

**The author's answer:** Thank you for your comments. **Lines 152**, The geometries of the myrcene, primary ozonides (POZs) and SCIs were optimized using the hybrid density functional theory B3LYP-D3(BJ) with the aug-cc-pVTZ basis set.

**References**

Atkinson, R.: Kinetics of the gas-phase reactions of OH radicals with alkanes and cycloalkanes, Atmos. Chem. Phys., 3, 2233-2307, 10.5194/acp-3-2233-2003, 2003.

Benoit, R., Belhadj, N., Dbouk, Z., Lailliau, M., and Dagaut, P.: On the formation of highly oxidized pollutants by autoxidation of terpenes under low-temperature-combustion conditions: the case of limonene and α-pinene, Atmos. Chem. Phys., 23, 5715-5733, 10.5194/acp-23-5715-2023, 2023.

Caravan, R. L., Bannan, T. J., Winiberg, F. A. F., Khan, M. A. H., Rousso, A. C., Jasper, A. W., Worrall,

S. D., Bacak, A., Artaxo, P., Brito, J., Priestley, M., Allan, J. D., Coe, H., Ju, Y., Osborn, D. L., Hansen, N., Klippenstein, S. J., Shallcross, D. E., Taatjes, C. A., and Percival, C. J.: Observational evidence for Criegee intermediate oligomerization reactions relevant to aerosol formation in the troposphere, Nat. Geosci., 17, 219–226, 10.1038/s41561-023-01361-6, 2024.

Chao, W., Markus, C. R., Okumura, M., Winiberg, F. A. F., and Percival, C. J.: Chemical Kinetic Study of the Reaction of $CH_2OO$ with $CH_3O_2$, J. Phys. Chem. Lett., 15, 3690-3697, 10.1021/acs.jpclett.4c00159, 2024.

Chen, L., Huang, Y., Xue, Y., Shen, Z., Cao, J., and Wang, W.: Mechanistic and kinetics investigations of oligomer formation from Criegee intermediate reactions with hydroxyalkyl hydroperoxides, Atmos. Chem. Phys., 19, 4075-4091, 10.5194/acp-19-4075-2019, 2019.

Chhantyal-Pun, R., Khan, M. A. H., Taatjes, C. A., Percival, C. J., Orr-Ewing, A. J., and Shallcross, D. E.: Criegee intermediates: production, detection and reactivity, Int. Rev. Phys. Chem., 39, 383-422, 10.1080/0144235x.2020.1792104, 2020a.

Chhantyal-Pun, R., Khan, M. A. H., Zachhuber, N., Percival, C. J., Shallcross, D. E., and Orr-Ewing, A. J.: Impact of Criegee Intermediate Reactions with Peroxy Radicals on Tropospheric Organic Aerosol, ACS Earth Space Chem., 4, 1743-1755, 10.1021/acsearthspacechem.0c00147, 2020b.

Chhantyal-Pun, R., Welz, O., Savee, J. D., Eskola, A. J., Lee, E. P. F., Blacker, L., Hill, H. R., Ashcroft, M., Khan, M. A. H., Lloyd-Jones, G. C., Evans, L., Rotavera, B., Huang, H., Osborn, D. L., Mok, D. K. W., Dyke, J. M., Shallcross, D. E., Percival, C. J., Orr-Ewing, A. J., and Taatjes, C. A.: Direct Measurements of Unimolecular and Bimolecular Reaction Kinetics of the Criegee Intermediate $(CH_3)_2COO$, J. Phys. Chem. A, 121, 4-15, 10.1021/acs.jpca.6b07810, 2017.

Coleman, B. E. and Ault, B. S.: Matrix isolation investigation of the ozonolysis of propene, J. Mol. Struct., 976, 249-254, https://doi.org/10.1016/j.molstruc.2010.03.050, 2010.

Coleman, B. E. and Ault, B. S.: Investigation of the mechanism of ozonolysis of (Z)-3-methyl-2-pentene using matrix isolation infrared spectroscopy, J. Mol. Struct., 1031, 138-143, 10.1016/j.molstruc.2012.07.046, 2013.

Deng, J., Chen, J., Liu, H., Wang, W., and Wang, X.: Matrix Isolation FT-IR Study on the Reaction Mechanisms between Ozone and Ethene, R. Environ. Sci., 25, 1-9, 2012.

Deng, Y. G., Inomata, S., Sato, K., Ramasamy, S., Morino, Y., Enami, S., and Tanimoto, H.: Temperature and acidity dependence of secondary organic aerosol formation from α-pinene ozonolysis with a compact chamber system, Atmos. Chem. Phys., 21, 5983-6003, 10.5194/acp-21-5983-2021, 2021.

Docherty, K. S. and Ziemann, P. J.: Effects of stabilized Criegee intermediate and OH radical scavengers

on aerosol formation from reactions of beta-pinene with $O_3$, Aerosol Sci. Technol., 37, 877-891, 10.1080/02786820300930, 2003.

Gong, Y. W., Jiang, F., Li, Y. X., Leisner, T., and Saathoff, H.: Impact of temperature on the role of Criegee intermediates and peroxy radicals in dimer formation from β-pinene ozonolysis, Atmos. Chem. Phys., 24, 167-184, 10.5194/acp-24-167-2024, 2024.

Hearne, T. S., Karakyriakos, E., Dunford, C. L., Kettner, M., Wild, D. A., and McKinley, A. J.: A matrix isolation ESR investigation of the MgCH radical, J. Chem. Phys., 151, 10.1063/1.5119146, 2019.

Hoops, M. D. and Ault, B. S.: Matrix Isolation Study of the Early Intermediates in the Ozonolysis of Cyclopentene and Cyclopentadiene: Observation of Two Criegee Intermediates, J. Am. Chem. Soc., 131, 2853-2863, 10.1021/ja8065286, 2009.

Huang, H.-L., Chao, W., and Lin, J. J.-M.: Kinetics of a Criegee intermediate that would survive high humidity and may oxidize atmospheric $SO_2$, Proc. Natl. Acad. Sci. U. S. A., 112, 10857-10862, 10.1073/pnas.1513149112, 2015.

Jokinen, T., Sipila, M., Richters, S., Kerminen, V. M., Paasonen, P., Stratmann, F., Worsnop, D., Kulmala, M., Ehn, M., Herrmann, H., and Berndt, T.: Rapid Autoxidation Forms Highly Oxidized $RO_2$ Radicals in the Atmosphere, Angew. Chem. Int. Ed., 53, 14596-14600, 10.1002/anie.201408566, 2014.

Jr-Min Lin, J. and Chao, W.: Structure-dependent reactivity of Criegee intermediates studied with spectroscopic methods, Chem. Soc. Rev., 46, 7483-7497, 10.1039/C7CS00336F, 2017.

Khan, M. A. H., Percival, C. J., Caravan, R. L., Taatjes, C. A., and Shallcross, D. E.: Criegee intermediates and their impacts on the troposphere, Environ. Sci.-Proc. Imp., 20, 437-453, 10.1039/c7em00585g, 2018.

Kugel, R. W. and Ault, B. S.: Infrared Matrix Isolation and Theoretical Studies of Reactions of Ozone with Bicyclic Alkenes: alpha-Pinene, Norbornene, and Norbornadiene, J. Phys. Chem. A, 119, 312-322, 10.1021/jp510883k, 2015.

Kugel, R. W. and Ault, B. S.: Infrared Matrix-Isolation and Theoretical Study of the Reactions of Ruthenocene with Ozone, J. Phys. Chem. A, 123, 5768-5780, 10.1021/acs.jpca.9b02374, 2019.

Li, S. S., Yang, X. Y., Xu, Y. S., and Jiang, L.: Identification of the early intermediates formed in ozonolysis of cis-2-butene and limonene: a theoretical and matrix isolation study, Rsc Adv., 9, 20100-20106, 10.1039/c9ra04176a, 2019.

Liu, D. D., Zhang, Y., Zhong, S. J., Chen, S., Xie, Q. R., Zhang, D. H., Zhang, Q., Hu, W., Deng, J. J., Wu, L. B., Ma, C., Tong, H. J., and Fu, P. Q.: Large differences of highly oxygenated organic molecules (HOMs) and low-volatile species in secondary organic aerosols (SOAs) formed from ozonolysis of β-pinene and limonene, Atmos. Chem. Phys., 23, 8383-8402, 10.5194/acp-23-8383-2023, 2023.

Liu, S., Galeazzo, T., Valorso, R., Shiraiwa, M., Faiola, C. L., and Nizkorodov, S. A.: Secondary Organic Aerosol from OH-Initiated Oxidation of Mixtures of d-Limonene and beta-Myrcene, Environ. Sci. Technol., 10.1021/acs.est.4c04870, 2024.

Luo, Y., Franzon, L., Zhang, J., Sarnela, N., Donahue, N. M., Kurten, T., and Ehn, M.: Gas-phase observations of accretion products from stabilized Criegee intermediates in terpene ozonolysis with two dicarboxylic acids, Atmos. Chem. Phys., 25, 4655-4664, 10.5194/acp-25-4655-2025, 2025.

Lv, C., Du, L., Tang, S. S., Tsona, N. T., Liu, S. J., Zhao, H. L., and Wang, W. X.: Matrix isolation study of the early intermediates in the ozonolysis of selected vinyl ethers, Rsc Advances, 7, 19162-19168, 10.1039/c7ra01011g, 2017.

Pinelo, L., Gudmundsdottir, A. D., and Ault, B. S.: Matrix Isolation Study of the Ozonolysis of 1,3-and 1,4-Cyclohexadiene: Identification of Novel Reaction Pathways, J. Phys. Chem. A, 117, 4174-4182, 10.1021/jp402981n, 2013.

Prenni, A. J., Petters, M. D., Kreidenweis, S. M., DeMott, P. J., and Ziemann, P. J.: Cloud droplet activation of secondary organic aerosol,. J. Geophys. Res.-Atmos., 112, D10223, 10.1029/2006JD007963, 2007.

Rousso, A. C., Hansen, N., Jasper, A. W., and Ju, Y. G.: Identification of the Criegee intermediate reaction network in ethylene ozonolysis: impact on energy conversion strategies and atmospheric chemistry, Phys. Chem. Chem. Phys., 21, 7341-7357, 10.1039/c9cp00473d, 2019.

Ryazantsev, S. V., Tyurin, D. A., Feldman, V. I., and Khriachtchev, L.: Spectroscopic characterization of the complex of vinyl radical and carbon dioxide: Matrix isolation and ab initio study, J. Chem. Phys., 147, 10.1063/1.5000578, 2017.

Sakamoto, Y., Inomata, S., and Hirokawa, J.: Oligomerization Reaction of the Criegee Intermediate Leads to Secondary Organic Aerosol Formation in Ethylene Ozonolysis, J. Phys. Chem. A, 117, 12912-12921, 10.1021/jp408672m, 2013.

Saraswat, M., Portela-González, A., Wulff, K., and Eckhardt, A. K.: P-Centered Dibenzophospholyl Radical: A Matrix Isolation IR, UV/Vis and ESR Spectroscopic Study, J. Org. Chem., 90, 10616-10624, 10.1021/acs.joc.5c00840, 2025.

Smith, M. C., Chao, W., Takahashi, K., Boering, K. A., and Lin, J. J.-M.: Unimolecular Decomposition Rate of the Criegee Intermediate $(CH_3)_2COO$ Measured Directly with UV Absorption Spectroscopy, J. Phys. Chem. A, 120, 4789-4798, 10.1021/acs.jpca.5b12124, 2016.

Taatjes, C. A., Welz, O., Eskola, A. J., Savee, J. D., Osborn, D. L., Lee, E. P. F., Dyke, J. M., Mok, D. W. K., Shallcross, D. E., and Percival, C. J.: Direct measurement of Criegee intermediate $(CH_2OO)$ reactions

with acetone, acetaldehyde, and hexafluoroacetone, Phys. Chem. Chem. Phys., 14, 10391-10400, 10.1039/C2CP40294G, 2012.

Tan, Z., Hantschke, L., Kaminski, M., Acir, I. H., Bohn, B., Cho, C., Dorn, H. P., Li, X., Novelli, A., Nehr, S., Rohrer, F., Tillmann, R., Wegener, R., Hofzumahaus, A., Kiendler-Scharr, A., Wahner, A., and Fuchs, H.: Atmospheric photo-oxidation of myrcene: OH reaction rate constant, gas-phase oxidation products and radical budgets, Atmos. Chem. Phys., 21, 16067-16091, 10.5194/acp-21-16067-2021, 2021a.

Tan, Z. F., Hantschke, L., Kaminski, M., Acir, I. H., Bohn, B., Cho, C. M., Dorn, H. P., Li, X., Novelli, A., Nehr, S., Rohrer, F., Tillmann, R., Wegener, R., Hofzumahaus, A., Kiendler-Scharr, A., Wahner, A., and Fuchs, H.: Atmospheric photo-oxidation of myrcene: OH reaction rate constant, gas-phase oxidation products and radical budgets, Atmos. Chem. Phys., 21, 16067-16091, 10.5194/acp-21-16067-2021, 2021b.

Tang, S. S., Du, L., Tsona, N. T., Zhao, H. L., and Wang, W. X.: A new reaction pathway other than the Criegee mechanism for the ozonolysis of a cyclic unsaturated ether, Atmos. Environ., 162, 23-30, 10.1016/j.atmosenv.2017.05.011, 2017.

Thomsen, D., Thomsen, L. D., Iversen, E. M., Bjorgvinsdottir, T. N., Vinther, S. F., Skonager, J. T., Hoffmann, T., Elm, J., Bilde, M., and Glasius, M.: Ozonolysis of $\alpha$-Pinene and $\Delta^3$-Carene Mixtures: Formation of Dimers with Two Precursors, Environ. Sci. Technol., 56, 16643-16651, 10.1021/acs.est.2c04786, 2022.

Virkkula, A., Van Dingenen, R., Raes, F., and Hjorth, J.: Hygroscopic properties of aerosol formed by oxidation of limonene, $\alpha$-pinene, and $\beta$-pinene, J. Geophys. Res.-Atmos, 104, 3569-3579, 10.1029/1998JD100017, 1999.

Wang, Y. Q., Zhao, Y., Li, Z. Y., Li, C. X., Yan, N. Q., and Xiao, H. Y.: Importance of Hydroxyl Radical Chemistry in Isoprene Suppression of Particle Formation from $\alpha$-Pinene Ozonolysis, Acs Earth Space Chem., 5, 487-499, 10.1021/acsearthspacechem.0c00294, 2021.

Wang, Z., Tong, S. R., Chen M. F., Jing, B., Li, W. R., Guo, Y. C., Ge, M. F., Wang, S. F.: Study on ozonolysis of asymmetric alkenes with matrix isolation and FT-IR spectroscopy, Chemosphere, 252, 126413, 10.1016/j.chemosphere.2020.126413, 2020.

Yang, X., Deng, J., Li, D., Chen, J., Xu, Y., Zhang, K., Shang, X., and Cao, Q.: Transient species in the ozonolysis of tetramethylethene, J. Environ. Sci., 95, 210-216, 10.1016/j.jes.2020.03.027, 2020.

Yang, Z., Li, K., Tsona, N. T., Luo, X., and Du, L.: $SO_2$ enhances aerosol formation from anthropogenic volatile organic compound ozonolysis by producing sulfur-containing compounds, Atmos. Chem. Phys., 23, 417-430, 10.5194/acp-23-417-2023, 2023.

Zasimov, P. V., Tyurin, D. A., Ryazantsev, S., and Feldman, V.: Formation and Evolution of $H_2C_3O^+\bullet$

Radical Cations: A Computational and Matrix Isolation Study, J. Am. Chem. Soc., 144, 8115-8128, 10.1021/jacs.2c00295, 2022.

Zhang, X., Lambe, A. T., Upshur, M. A., Brooks, W. A., Be, A. G., Thomson, R. J., Geiger, F. M., Surratt, J. D., Zhang, Z., Gold, A., Graf, S., Cubison, M. J., Groessl, M., Jayne, J. T., Worsnop, D. R., and Canagaratna, M. R.: Highly Oxygenated Multifunctional Compounds in α-Pinene Secondary Organic Aerosol, Environ. Sci. Technol., 51, 5932-5940, 10.1021/acs.est.6b06588, 2017.

Zhao, J., Häkkinen, E., Graeffe, F., Krechmer, J. E., Canagaratna, M. R., Worsnop, D. R., Kangasluoma, J., and Ehn, M.: A combined gas- and particle-phase analysis of highly oxygenated organic molecules (HOMs) from α-pinene ozonolysis, Atmos. Chem. Phys., 23, 3707-3730, 10.5194/acp-23-3707-2023, 2023.

Zhao, Y., Wingen, L. M., Perraud, V., Greaves, J., and Finlayson-Pitts, B. J.: Role of the reaction of stabilized Criegee intermediates with peroxy radicals in particle formation and growth in air, Phys. Chem. Chem. Phys., 17, 12500-12514, 10.1039/c5cp01171j, 2015.

Zhao, Y., Yao, M., Wang, Y., Li, Z., Wang, S., Li, C., and Xiao, H.: Acylperoxy Radicals as Key Intermediates in the Formation of Dimeric Compounds in α-Pinene Secondary Organic Aerosol, Environ. Sci. Technol., 56, 14249-14261, 10.1021/acs.est.2c02090, 2022.

Zhu, C., Duarte, L., and Khriachtchev, L.: Matrix-isolation and computational study of $H_2CCCl$ and $H_2CCBr$ radicals, J. Chem. Phys., 145, 10.1063/1.4961155, 2016.